# Domain-specific schema reuse supports flexible learning to learn in the primate brain

Kaixi Tian[1,2,3,7], Zhiping Zhao[4,7], Yang Chen[1,3,7], Ningling Ge[1,3,5], Shenghao Cao[1,3,5], Xinyong Han[1], Jianwen Gu[6] ✉ & Shan Yu ⓞ[1,2,3] ✉

Prior knowledge accelerates subsequent learning of similarly structured problems – a phenomenon termed learning to learn – by forming generalizable neural representations called neural correlates of schema (NCS). However, how the brain exploits stable NCS while remaining flexible towards changes (the stability-plasticity dilemma) remains unclear. Here, we show that the primate brain addresses this dilemma by representing the stable NCS and task-unique changes in a near-orthogonal manner. We analyzed neural activities in the dorsolateral premotor cortex of three male macaques trained to perform a series of visuomotor mapping tasks. By delineating decision and stimulus-related subspaces, we identified NCS within the decision subspace, whose reuse facilitated subsequent learning. In addition, the decision subspace exhibited a near-orthogonal relationship with the stimulus-related subspace, minimizing cross-domain interference. Our results reveal that restricting NCS to specific functional domains can preserve useful knowledge while maintaining near-orthogonality with other subspaces, enabling flexible adaptation to new environments, thereby resolving the stability-plasticity dilemma.

Prior knowledge can expedite the learning process, enabling faster learning of new problems if they are similar to the ones already known. This process is known as learning to learn[1,2]. In cognitive psychology, such prior knowledge is often referred to as a schema – a structured mental representation that organizes and guides behavior[3–5]. It is hypothesized that acquired knowledge is represented by specific neural activity patterns, which we refer to as neural correlates of schema (NCS). These patterns can be reused to facilitate subsequent learning. NCS have been observed in rodents[6,7], non-human primates[8,9] and humans[10,11], manifesting as stable activity patterns across similarly structured tasks. In addition, schema formation has been observed in parallel with improved learning efficiency when mice performed sequence tasks[12]. However, regarding the schema-based hypothesis of learning to learn, the stability-plasticity dilemma remains unsolved. That is, how can the brain maintain stable representations so that

essential knowledge can be preserved while still leaving enough plasticity to adapt to possible changes in subsequent learning (Fig. 1a).

The stability-plasticity dilemma presents a significant challenge in the field of machine learning, which concerns how intelligent systems can leverage prior tasks or experiences to inform future learning. Various approaches have been employed to address this issue: continual learning explores the problem of catastrophic forgetting in artificial neural networks[13–15], meta-learning focuses on how to rapidly adapt to new tasks through a few trials[16,17], and online learning emphasizes strategies for tackling unknown tasks in dynamic environments[18]. Despite significant advances in these fields, it is still challenging for artificial intelligence systems to balance stability – retaining past knowledge while minimizing catastrophic forgetting – and plasticity, which involves the ability to learn quickly from new experiences and generalize effectively. In contrast, biological systems,

[1]Laboratory of Brain Atlas and Brain-inspired Intelligence, Institute of Automation, Chinese Academy of Sciences, Beijing, China. [2]School of Future Technology, University of Chinese Academy of Sciences, Beijing, China. [3]State Key Laboratory of Brain Cognition and Brain-inspired Intelligence Technology, Institute of Automation, Chinese Academy of Sciences, Beijing, China. [4]Department of Ophthalmology, First Hospital of Jilin University, Changchun, China. [5]School of Artificial Intelligence, University of Chinese Academy of Sciences, Beijing, China. [6]The Ninth Medical Center of PLA General Hospital, Beijing, China. [7]These authors contributed equally: Kaixi Tian, Zhiping Zhao, Yang Chen. ✉e-mail: jiuwufushou@qq.com; shan.yu@nlpr.ia.ac.cn

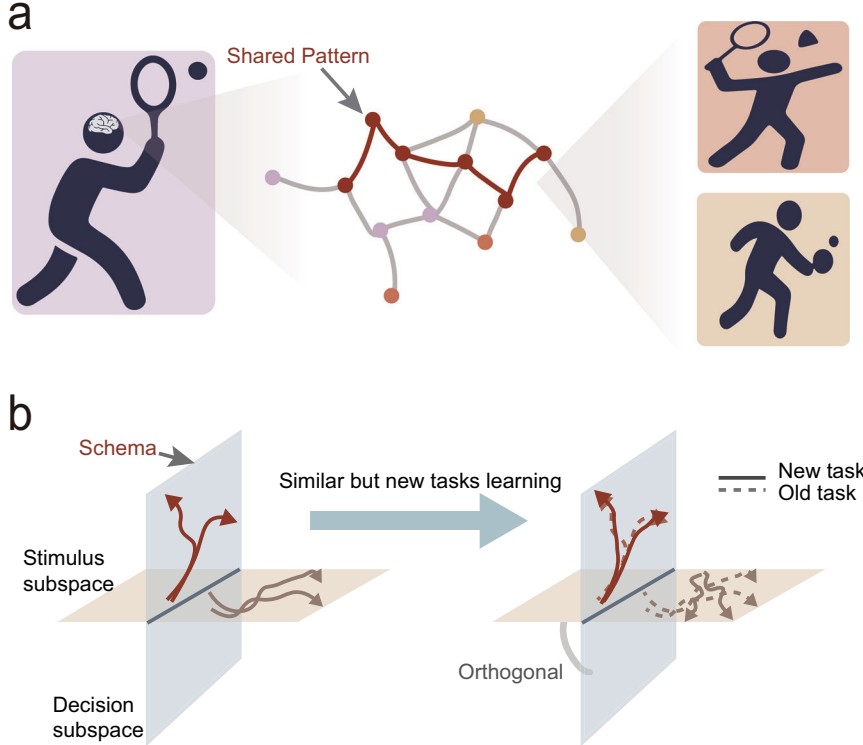

**Fig. 1 | Stability-plasticity dilemma and proposed neural solution. a** Illustration of the stability–plasticity dilemma, highlighting the balance between reusing existing knowledge (schemas) and the need for flexible adaptation. For example, individuals trained in one racquet sport, such as tennis, can often learn related sports like badminton or table tennis more efficiently, suggesting the transfer of stable motor schemas across contexts with variable demands. **b** Hypothesized neural solution: domain-specific NCS are preserved to maintain core task-relevant knowledge, while near-orthogonal neural subspaces allow flexible adaptation to task changes without disrupting the stable representations.

particularly mammals, exhibit remarkable balance between stability and plasticity: they preserve and effectively capitalize on prior knowledge, while still being highly flexible in dealing with unforeseeable changes[19,20].

What neural mechanisms allow the brain to balance stability and plasticity during schema-based learning? Recently, Goudar et al. used an artificial neural network model to simulate the learning of a series of visuomotor mapping tasks[21]. The model exhibited a schema-like, low-dimensional activity manifold, accompanied by increased learning efficiency. The authors demonstrated that the schema and its reuse were observed in the decision subspace, but not in the stimulus-related subspace. Similarly, Driscoll et al. achieved flexible multi-task computation in artificial neural networks through reuse of local dynamical motifs[22]. These studies inspired us to hypothesize that restricting the schema to specific functional domains, e.g., decision, and making it near-orthogonal to other subspaces allows the brain to preserve essential knowledge while enabling flexibility to handle the variability of future problems, e.g., with different stimuli, thereby providing a solution to the stability-plasticity dilemma (Fig. 1b). Here, we show that the macaque brain addresses the stability-plasticity dilemma through domain-specific schema representation and near-orthogonal subspace organization. We train three macaques on a series of visuomotor mapping tasks and record neural population activity in the dorsolateral premotor cortex (PMd). We find that NCS are embedded within the decision-related subspace, where their reuse facilitates subsequent learning of similar tasks, while the stimulus-related subspace remains unconstrained for representing novel sensory inputs. Critically, we find that these two subspaces maintain a near-orthogonal relationship, enabling the brain to achieve efficient learning while remaining flexible to solve new problems.

## Results

We trained three macaques (Monkey AB, ZZ, and XW) on a series of visuomotor mapping tasks and recorded neural population activity in the dorsolateral premotor cortex (PMd) (Supplementary Fig. 1a), a region involved in decision-making and action planning[23,24]. The tasks required the monkeys to press corresponding buttons based on the presented visual stimuli (Fig. 2a). Behavioral data were collected across multiple sessions (Monkey AB: 5 sessions, 383.8 ± 106.2 trials per session; Monkey XW: 5 sessions, 225.4 ± 33.2 trials per session; Monkey ZZ: 4 sessions, 390.3 ± 105.8 trials per session, mean ± SD). Each session, the monkeys needed to learn two (task A and B for monkeys AB and ZZ) or three (task A, B, and C for monkey XW) new stimulus-action pairs, followed by a task of revisiting pair A (Revisit-A). For monkeys AB and ZZ, an extra reverse task was added at the end, in which the animals needed to learn the reverse mapping of pair A (Reverse-A) (Fig. 2b). New tasks were introduced sequentially, and only after the monkey reached the learning criterion for the preceding one (see Methods). All task conditions were completed once per session and presented in a fixed order on the same session. The full set of visual stimuli is shown in Supplementary Fig. 1c, with monkeys encountering novel visual stimuli each session that they had never seen before. Neural data analysis focused on three recording sessions for each monkey (Monkey AB/ZZ/XW completed 420.7 ± 129.1 / 357.0 ± 109.3 / 209.0 ± 6.6 trials per session, mean ± SD). In total, we recorded activities from 728 multi-units (MUA) and single-units (SUA). The mean (± SD) unit counts per session were: Monkey AB: 42.3 ± 3.8 units, Monkey ZZ: 43.3 ± 1.5 units, and Monkey XW: 157.0 ± 21.7 units (see Methods for details). Only the activities during the 1-second visual cue presentation time, before the onset of the action, were analyzed in the present study (see Methods for details).

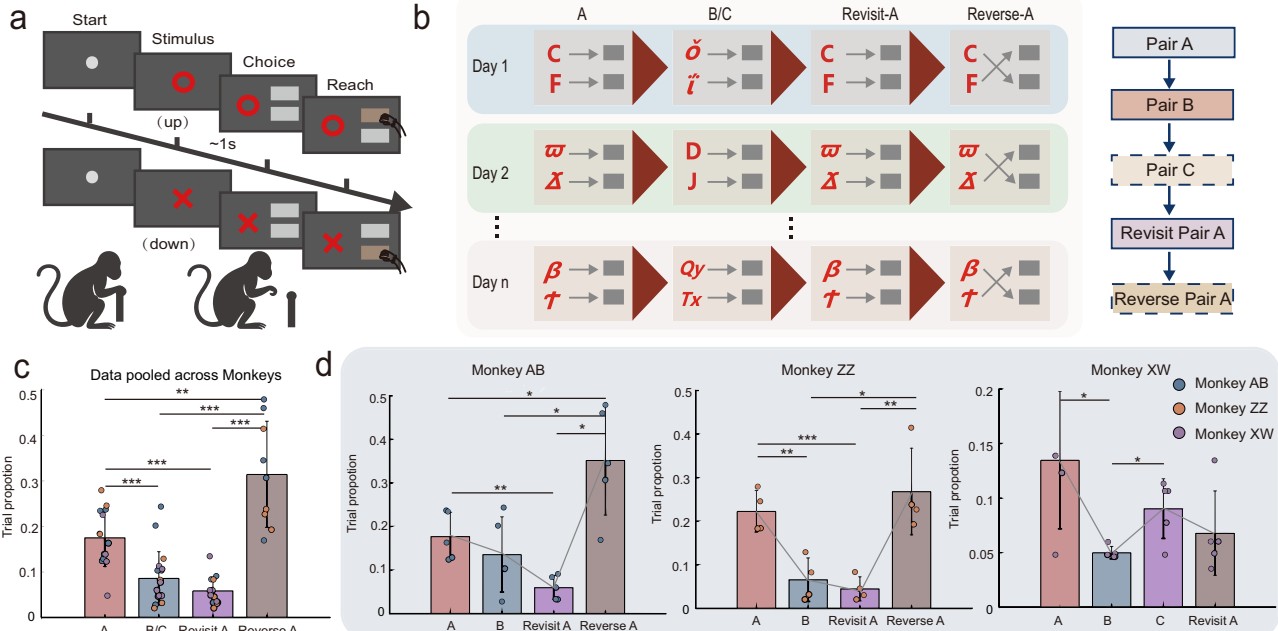

**Fig. 2 | Experimental paradigm and behavioral analysis. a** Schematic of the visuomotor mapping tasks. **b** Workflow for learning a series of visuomotor mappings within a single session. Left: Monkeys were trained on new visuomotor mappings each session, with visual stimulus pairs not seen before. Monkeys progressed to the next task only after successfully learning the previous one. Right: Task sequences completed within a session. Monkeys AB and ZZ followed an A–B–Revisit A–Reverse A training sequence; monkey XW followed an A–B–C–Revisit A training sequence. **c** Proportion of trials required to reach performance criterion (90% correct over 15 consecutive trials) relative to the total number of trials in a session. Data pooled across three monkeys. Bar plots show the mean across 14 sessions (Monkey AB: 5 sessions; Monkey ZZ: 4 sessions; Monkey XW: 5 sessions), with error bars indicating standard deviation across sessions. Data analyzed by two-tailed paired t-tests (A vs. B/C: $P = 2.30E\text{-}04$, $g = 1.47$; A vs.

$P = 1.52E\text{-}06$, $g = 2.34$; A vs. Reverse-A: $P = 1.20E\text{-}03$, $g = \text{-}1.59$; B/C vs. Reverse-A: $P = 2.07E\text{-}07$, $g = \text{-}2.82$; Revisit-A vs. Reverse-A: $P = 1.02E\text{-}07$, $g = \text{-}3.37$). **d** Comparison of task learning efficiency across monkeys within-session. For each monkey, the proportion of trials needed to reach criterion is plotted, with error bars representing standard deviation across recording sessions (Monkey AB: $n = 5$ sessions. A vs. Revisit-A: $P = 0.006$, $g = 2.36$, A vs. Reverse-A: $n = 5$ sessions, $P = 0.022$, $g = \text{-}1.80$; B vs. Reverse-A: $P = 0.013$, $g = \text{-}2.00$; Revisit-A vs. Reverse-A: $P = 0.001$, $g = \text{-}3.05$; Monkey ZZ: $n = 4$ sessions, A vs. B: $P = 0.004$, $g = 3.22$, A vs. Revisit-A: $P < 0.001$, $g = 4.57$, B vs. Reverse-A: $n = 4$ sessions, $P = 0.011$, $g = \text{-}2.57$; Revisit-A vs. Reverse-A: $P = 0.005$, $g = \text{-}3.06$; Monkey XW: $n = 5$ sessions, A vs. B: $P = 0.017$, $g = 1.90$, B vs. C: $P = 0.01$, $g = \text{-}2.04$). Data are presented as mean ± SD. Colored scatter dots represent individual sessions, with each color corresponding to one monkey. *, $P < 0.05$, **, $P < 0.01$, ***, $P < 0.001$, Data analyzed by two-tailed paired t-tests.

## Similar tasks learned faster but reversals delayed

The behavioral results are shown in Fig. 2c, d. Similar to previous findings[2,12], we found that learning efficiency (measured by the number of trials needed to reach a 0.9 correct rate in 15 consecutive trials) increased for later-learned pairs. To account for cross-session variability in task engagement, we normalized this value by the number of trials completed on each session. Within-subject analysis across sessions (Fig. 2d) showed that the learning speeds for pair B were significantly faster compared to pair A in monkeys ZZ and XW (Monkey ZZ: $n = 4$ sessions, $P = 0.004$, Hedges' $g = 3.22$; Monkey XW: $n = 5$ sessions, $P = 0.017$, $g = 1.90$). Similarly, learning speed for the Revisit-A was significantly faster compared to pair A in monkeys AB and ZZ (Monkey ZZ: $n = 4$ sessions, $P < 0.001$, $g = 4.57$; Monkey AB: $n = 5$ sessions, $P = 0.006$, $g = 2.36$). In contrast, the reverse task was more challenging, with the learning speed of Reverse-A significantly slower than that of A, B, and Revisit-A (Monkey ZZ: B vs. Reverse-A: $n = 4$ sessions, $P = 0.011$, $g = \text{-}2.57$; Revisit-A vs. Reverse-A: $P = 0.005$, $g = \text{-}3.06$; Monkey AB: A vs. Reverse-A: $n = 5$ sessions, $P = 0.022$, $g = \text{-}1.80$; B vs. Reverse-A: $P = 0.013$, $g = \text{-}2.00$; Revisit-A vs. Reverse-A: $P = 0.001$, $g = \text{-}3.05$; Fig. 2d). Pooling the data from three monkeys, where tasks B and C were grouped together since only Monkey XW performed task C and both represent subsequent learning of new, similar visual–motor mappings, reconfirmed the trends we saw in individual animals, that is, later learned pairs exhibited significantly increased efficiency, while the reverse learning task was significantly slowed (pooled $n = 14$ sessions, i.e., 5 from Monkey AB, 4 from Monkey ZZ and 5 from Monkey XW; A vs. B/C: $P = 2.30E\text{-}04$, $g = 1.47$; A vs.

Revisit-A: $P = 1.52E\text{-}06$, $g = 2.34$; A vs. Reverse-A: $P = 1.20E\text{-}03$, $g = \text{-}1.59$; B/C vs. Reverse-A: $P = 2.07E\text{-}07$, $g = \text{-}2.82$; Revisit-A vs. Reverse-A: $P = 1.02E\text{-}07$, $g = \text{-}3.37$; Fig. 2c). The analysis of the reaction time (RT) also corroborated that of the accuracy (Supplementary Fig. 1b). Overall, these behavioral results reconfirm that similar problems encountered later can be learned faster, but learning new problems may be delayed if they contradict previously acquired knowledge (see Supplementary Notes 1.1).

## NCS identified in decision subspace

Next, we searched for NCS in population activities recorded from the PMd, by decomposing to the stimulus- and decision-related domains. First, we examined whether the population activity indeed encoded task-related parameters in the stimulus-related and decision domains. To this end, we constructed two classifiers using convolutional neural networks (CNNs) to decode visual stimuli and motor decisions for each monkey on each of the three recording sessions ($n = 3$ sessions per monkey; Fig. 3a). We employed a nonlinear dimensionality reduction method (LFADS)[25] to reduce the high-dimensional neural firing rate data to a low-dimensional space ($n = 16$ dimensions, see Supplementary Notes 1.2 and Supplementary Notes 1.3 for the reasons for selecting methods and dimensions), which mitigates trial-to-trial variability in neural recordings and enhances decoding performance by extracting smooth underlying dynamics. The data recorded during the learning of new pairs were used to train the classifiers. For each session's recordings, 80% of trials were randomly selected for training and the remaining 20% for validation. We found that PMd activities

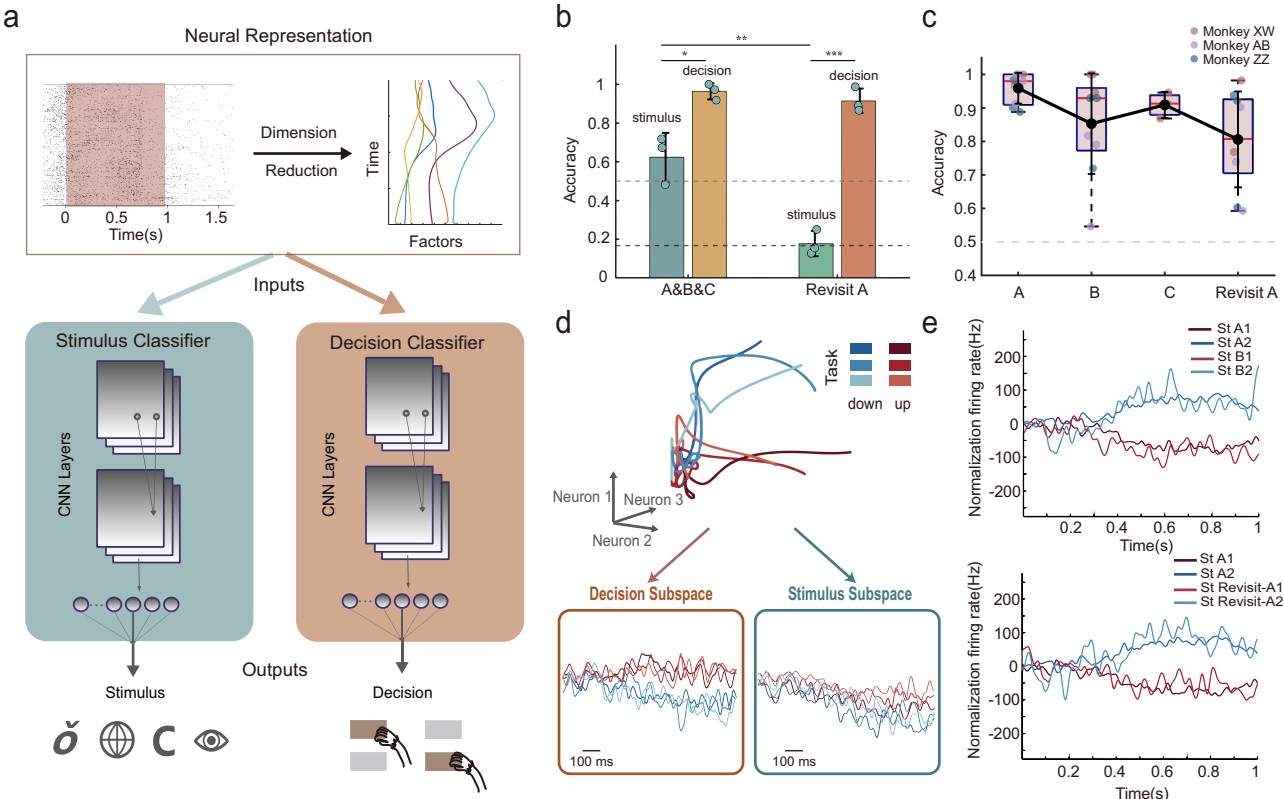

**Fig. 3 | Identifying shared patterns from neural population activity. a** Schematic of the decoding shared patterns across tasks: neural population activity from task A was reduced to a lower dimension ($n = 16$ dimensions) using a nonlinear method, and CNNs were trained separately to classify different visual stimuli and motor decisions. The reddish shadow area highlights neural activity used from the onset of visual stimulus presentation to the appearance of the choice buttons. **b** Example from monkey XW: classification accuracy of CNN decoders trained to decode visual stimuli and motor decisions from Task A/B/C on each session ($n = 3$ sessions). Trained decoders were then tested on Revisit-A task; Bars show mean accuracy, and error bars represent standard deviation across three recording sessions (A&B&C vs. chance level: Stimulus: $P = 0.02$, $g = 2.27$; Decision: $P = 0.003$, $g = 6.36$; Stimulus: Revisit-A vs. chance level: $P = 0.30$, $g = 0.46$; ABC vs. Revisit-A: $P = 0.005$, $g = 4.48$; Decision: Revisit-A vs. chance level: $P = 0.008$, $g = 3.69$, ABC vs. Revisit-A: $P = 0.33$, $g = 0.91$). The dark grey dashed line indicates chance level for stimulus classification (0.17), and the light grey dashed line indicates chance level for decision classification (0.5). Colored scatter dots represent individual sessions. **c** Generalization accuracy of decision classifiers trained on Task A when applied to Task B, Task C, and Task Revisit-A across three monkeys. Bars show mean accuracy across all

sessions (9 sessions total: 3 from monkey AB, 3 from monkey ZZ, 3 from monkey XW); Box plots show the median (centre line), 25th and 75th percentiles (box bounds), and whiskers extend to minimum and maximum values. Each dot indicates performance from one session, with different colors representing individual monkeys. The grey line denotes the decision classification chance level (0.5). Data analyzed by one-way ANOVA ($P = 0.07$, Cohen's $f = 0.56$). **d** Decompose neural population activity from an example session of monkey AB into the decision- and the stimulus-related subspace, showing principal components explaining the highest proportion of variance in each subspace. Color intensity reflects different tasks (A-B-Revisit A) within a session, ranging from darker to lighter shades. Red and blue represent the trial average activity for choices toward the upper and lower buttons, respectively. **e** Neural activity during subsequent learning tasks from one session of monkey XW was projected into the primary principal component of the decision subspace using the projection matrix from task A. The top panel shows task B, and the bottom panel shows Revisit-A. St-A1: Stimulus 1 in Task A; St-A2: Stimulus 2 in Task A; St-Revisit-A1: Stimulus 1 in Task Revisit-A; St-Revisit-A2: Stimulus 2 in Task Revisit-A; *, $P < 0.05$, **, $P < 0.01$, ***, $P < 0.001$; two-tailed t-tests.

encoded both visual stimuli and motor decisions, with the corresponding classifiers exhibiting significantly higher performance compared to the chance level (A&B&C: Stimulus: $P = 0.021$, $g = 2.27$; Decision: $P = 0.003$, $g = 6.36$; Fig. 3b). We note that the encoding of visual stimuli was independent of the motor decisions, as the visual classifier could discriminate different stimuli associated with the same motor response (10-fold cross-validation, stimuli classification accuracy, mean ± SD: $0.67 \pm 0.03$, $0.48 \pm 0.01$, $0.72 \pm 0.03$ for sessions 1-3; decision classification accuracy, mean ± SD: $1.00 \pm 0.00$, $0.92 \pm 0.01$, $0.97 \pm 0.002$ for sessions 1-3; Fig. 3b). Results for monkey XW are shown on the left side of Fig. 3b, and similar results for monkeys AB and ZZ are shown in Supplementary Fig. 4b. To test whether these information-bearing neural activities were stable across tasks, we fixed the parameters of the two classifiers trained on each session's data and calculated their performance on the same session's unseen data in the Revisit-A task. Surprisingly, even with the same stimuli as in task A, the visual classifier failed to discriminate the visual stimuli (compared with

chance level: $P = 0.300$, $g = 0.46$; see the right side of Fig. 3b; monkey AB, ZZ's results in Supplementary Fig. 4b), suggesting that the visual representation in the PMd was not stable across tasks ($n = 3$ sessions; ABC vs. Revisit-A: $P = 0.005$, $g = 4.48$). In contrast, the motor decision classifier generalized well when applied to the unseen data from Revisit-A, compared to the data on which it was trained (compared with chance level: $P = 0.008$, $g = 3.69$; the right side of Fig. 3b; monkey AB, ZZ's results in Supplementary Fig. 4b), suggesting the decision-related information was stable across tasks ($n = 3$ sessions; ABC vs. Revisit-A: $P = 0.33$, $g = 0.91$).

To further explore the across-task stability of decision-related representations, we trained another motor decision classifier for each session using data only from task A and tested its generalization ability on tasks B/C from the same session. We found that the classifier trained on task A data successfully generalized to the unseen data from tasks B/C and Revisit-A in all monkeys without significant differences ($n = 3$ sessions per monkey; one-way ANOVA, Monkey XW: $P = 0.09$, Cohen's

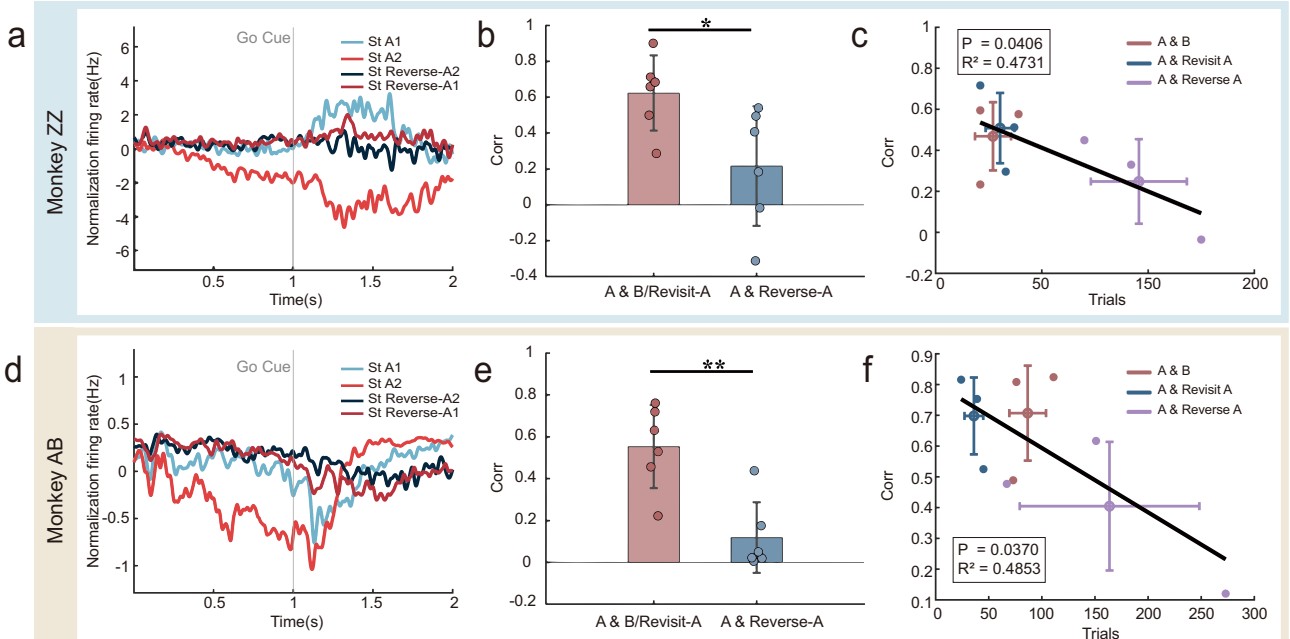

**Fig. 4 | Decision-related neural representations in the reverse task. a, d** Neural trajectories projected onto the principal component of the decision-related subspace for Task A and Reverse-A, shown for a representative recording session. **b, e** The similarity between the representation of the principal component of the decision subspace manifold in A/B/Revisit-A tasks and in the Reverse-A task. Similarity was quantified using Pearson correlation. Data are presented as mean ± SD. Error bars indicate standard deviation across three recording sessions. Each dot represents the similarity for one direction (up or down) manifold within a recording session, yielding six data points in total. ($n = 3$ sessions per monkey; two-tailed t-test on Pearson correlation coefficients, Monkey ZZ: $P = 0.03$, $g = 1.47$; Monkey AB:

$P = 0.002$, $g = 2.36$). **c, f** A linear relationship exists between the similarity of the representation of the decision subspace manifold and the number of trials required to learn the tasks. Each small, filled circle represents data from one recording session for a specific task condition (color-coded). Large hollow circles show condition averages. Cross-shaped error bars indicate standard deviations in both axes over three sessions. The coefficient of determination ($R^2$) and p-value of the regression model compared to the null model are indicated on the graph. ($n = 3$ sessions per monkey; linear regression, Monkey ZZ: $P = 0.04$, $R^2 = 0.47$; Monkey AB: $P = 0.04$, $R^2 = 0.49$). *, $P < 0.05$; **, $P < 0.01$; ***, $P < 0.001$.

$f = 1.09$; Monkey ZZ: $P = 0.65$, Cohen's $f = 0.39$; Monkey AB: $P = 0.10$, Cohen's $f = 1.06$. Supplementary Fig. 4a), confirming that in similar tasks, the decision-related representation was largely shared across individual tasks. Combined across all three monkeys, the classifiers maintained good generalization performance without significant degradation ($n = 9$ total sessions; 3 per monkey; one-way ANOVA, $P = 0.07$, Cohen's $f = 0.56$. Figure 3c). The decoder analyses suggested that population-level neural representations for motor decisions in PMd were shared across tasks. To rule out the possibility that this reflects intrinsic directional biases, we examined single-neuron preference indices across learning blocks (Supplementary Fig. 7; see Supplementary Notes 1.4.3 for more details) and found no consistent target preference. Thus, pre-existing biases are unlikely to explain the reuse of task-related neural dynamics (see Supplementary Notes 1.4 for additional single-neuron analyses).

### Decision-related manifolds are reused across similar tasks
The classifier analyses demonstrated the existence of stable NCS in the domain of motor decisions. Next, we examined more explicitly how such representations are embedded in population neural dynamics. The results from a representative recording session of monkey AB are shown in Fig. 3d, with similar results found in the other two monkeys (Supplementary Fig. 4c). We used demixed principal component analysis (dPCA)[26] to project population activity collected in tasks A, B/C, and Revisit-A into the stimulus- and decision-related subspaces (projecting to 20 dimensions, see Supplementary Notes 1.5 and 1.6 for details on dPCA method and dimensionality selection). Using a representative recording session from Monkey AB ($n = 45$ units, 240 trials), we found that, in the major component of decision subspace, neural dynamics associated with the same decision across different

tasks were clustered and separated from the cluster representing the opposite decision (Fig. 3d, lower-left), suggesting that the NCS were represented in the decision subspace. Consistent with the classifier results, the population dynamics representing different stimuli were largely mixed (Fig. 3d, lower-right, showing the principal component of the stimulus-related subspace).

We further examined the reuse of low-dimensional neural manifolds in the decision subspace across individual tasks. First, we conducted the dimension reduction for task A data in the decision subspace. Then, we projected the data from tasks B/C and Revisit-A into the same subspace. We found that, similar to the results described above, the decision manifolds obtained from task A were reused for other similar tasks (as illustrated by a representative recording session from each monkey; Fig. 3e, showing the principal component of the decision subspace; monkey AB, ZZ's results in Supplementary Fig. 10).

### Decision NCS reuse correlates with learning efficiency
Next, we analyzed the population activities in the Reverse-A task. It helps to discriminate if the reused neural manifolds indeed reflect the decision per se, or merely reflect the planned movement direction. If the latter was the case, we would expect that the reuse could be found in the Reverse-A task as well, since the movement directions in this task were the same as in other tasks. The results show that the opposite was true, i.e., stable neural manifolds shared across tasks A, B/C, and Revisit-A were not reused in the Reverse-A task (as illustrated by a representative session in Fig. 4a, d). Quantitative analysis confirmed that the similarity of the manifold in the decision subspace between the task A and Reverse-A was significantly lower than that between the task A and tasks B/C/Revisit-A ($n = 3$ sessions per monkey; Pearson correlation, Monkey ZZ: $P = 0.03$, $g = 1.47$; Monkey AB: $P = 0.002$,

$g = 2.36$; Fig. 4b, e), suggesting that the information encoded in these representations was decision-related, rather than movement-related. When combining data from both monkeys, the manifold similarity between Task A and Reverse-A remained significantly lower than that between Task A and B/C/Revisit-A ($n = 6$ sessions; Pearson correlation; $P < 0.001$, $g = 1.84$; Supplementary Fig. 12).

Then, we examined the functional role of the NCS reuse in the decision subspace identified above. In Fig. 4c, f, for monkeys AB and ZZ ($n = 3$ sessions per monkey), we plotted the number of trials needed each session to learn the task B, Revisit-A, and Reverse-A versus the degree to which the NCS identified in task A was reused (measured by the correlation between the manifolds in the major principal decision subspace). The results showed a significant negative relationship between the two (Monkey ZZ: $P = 0.04$, $R^2 = 0.47$; Monkey AB: $P = 0.04$, $R^2 = 0.49$). In the case of the Reverse-A task, learning was actually delayed, as it required more trials to learn compared to task A. These results suggest that whether the NCS can be reused may strongly affect subsequent learning, i.e., it may facilitate learning for similar tasks but delay learning for dissimilar tasks.

### Near-orthogonal separation between subspaces

We have demonstrated above that the NCS were only preserved and reused in the decision but not the stimulus-related subspace (see Supplementary Notes 1.9 for details of stimulus-related subspaces). Next, we investigated how the brain reduces overlap between decision- and stimulus-related representations. Computationally, an orthogonal relationship is the ideal form to minimize possible interference between different dimensions. To test whether such near-orthogonality exists between the decision- and stimulus-related subspaces, we selected principal components from each subspace based on their explained variance (see Supplementary Notes 1.12 for details of near-orthogonality). For the stimulus-related subspace, which had a higher dimensionality, we included components that explained more than 1% of the variance and together accounted for at least 70% of the total variance in that subspace. For the decision subspace, components that individually explained more than 1% variance were included, without applying a cumulative threshold (see Methods for details and Supplementary Fig. 13 for cumulative variance explained in both subspaces). We then computed the angles between the decision and stimulus-related subspaces by calculating the cosine of the principal angle between their respective orthonormal bases. An angle close to 90° compared to shuffle levels indicates near-orthogonality and reduced representational overlap. As a control, we used a shuffling strategy to randomize the stimulus and decision labels for each trial, recalculating the angular separation between the subspaces (see Supplementary Notes 1.7 for further discussion under additional shuffle control conditions). We found that the angle derived from actual data was significantly closer to orthogonal than the shuffling results (Watson-Williams test; $n = 3$ sessions per monkey; Monkey ZZ: $P = 0.04$, $g = 1.27$; Monkey AB: $P = 0.005$, $g = 1.77$; Monkey XW: $P = 0.002$, $g = 2.02$; Fig. 5a). When combining data from three monkeys ($n = 9$ sessions), the angle between the two subspaces remained significantly closer to orthogonal compared to the shuffled control (Watson-Williams test; $P < 0.001$, $g = 1.66$; Supplementary Fig. 15). The scatter plot in Fig. 5b illustrates the computational advantages of near-orthogonal encoding. Taking one session from monkey AB as an example, the analysis shows that decision separability is greatly reduced in the stimulus-related dimension ($P = 0.49$, $\eta^2 = 0.002$), while showing good discrimination on the decision dimension ($P < 0.001$, $\eta^2 = 0.5752$). Conversely, stimulus separability in the decision dimension mainly depends on the same movement direction (e.g., blue-red represents the same movement direction in two tasks, red-green represents another direction), failing to well distinguish different visual stimuli of the same direction (St A1 vs St B1: $P = 0.53$, $\eta^2 = 0.0032$; St A2 vs St B2: $P = 0.46$, $\eta^2 = 0.0044$; St A1 vs St A2: $P < 0.001$, $\eta^2 =$

0.3372; St A1 vs St B2: $P < 0.001$, $\eta^2 = 0.2634$; St A2 vs St B1: $P < 0.001$, $\eta^2 = 0.2884$; St B1 vs St B2: $P < 0.001$, $\eta^2 = 0.2192$), while being able to better distinguish visual stimuli of the same movement direction on the stimulus-related dimension (St A1 vs St B1: $P = 0.002$, $\eta^2 = 0.0791$; St A2 vs St B2: $P < 0.001$, $\eta^2 = 0.1621$; St A1 vs St A2: $P = 0.534$, $\eta^2 = 0.0032$; St A1 vs St B2: $P < 0.001$, $\eta^2 = 0.1426$; St A2 vs St B1: $P < 0.001$, $\eta^2 = 0.1005$; St B1 vs St B2: $P = 0.15$, $\eta^2 = 0.0173$; see more details in Supplementary Fig. 16). To better visualize the near-orthogonal representation, we plotted the neural dynamics in three-dimensional space. Figure 5c illustrates the near-orthogonal relationship between these two subspaces. We selected a session with an average angle between the two subspaces of 81° for monkey AB to depict the temporal evolution of neural signals along these axes. As a control, an example with a 25° angle is also shown, which was calculated for minor components in the two subspaces that explained much less variance. Clearly, when the decision subspace is approximately orthogonal to the stimulus-related subspace, visual stimuli and motor decisions can be effectively separated without interference (Fig. 5d, left side). However, when the relation is far from orthogonal, there is significant overlap between the two subspaces, distinguishing different visual stimuli and classifying motor decisions becomes intermingled (Fig. 5d, right side). Taken together, these results indicate a near-orthogonal relationship between the decision and visual stimulus dimensions, which may contribute to non-interfering representation of decision and sensory stimuli.

## Discussion

In the present study, we demonstrated that stable NCS were embedded in the low-dimensional manifolds in the PMd of macaques during the process of learning a series of visuomotor mapping tasks. The NCS were represented and reused only in the decision subspace, and its reuse strongly affects subsequent learning, providing strong evidence for the schema-based hypothesis of learning to learn[21]. To distinguish this from the broader cognitive construct of schema in psychology, see Supplementary Notes 1.8 for further discussion.

Theoretically, by restricting the formation, preservation and reuse of the NCS to the decision subspaces, subsequent learning of similar rules can be accomplished by within-manifold perturbation, which can be implemented more easily[27,28]. At the same time, this also frees other task-related representational subspaces, e.g., the subspace representing visual stimuli or the motor responses, so that they can be modified to handle various sensory inputs and motor outputs in solving new problems. Our findings, for the first time, demonstrate that this is indeed the case for the primate brain, when it is engaged in a series of tasks that require quick and flexible learning. In addition, we found that such domain-specific preservation and reuse of NCS may rely on an approximately orthogonal relationship between the decision and visual subspaces. For detailed discussion regarding the near-orthogonal relationship between these two subspaces, see Supplementary Notes 1.10. This not only provides insight into the computational principles that may underlie the effective use of schemas in the brain, but also suggests a possible solution for artificial intelligence systems to preserve and reuse key knowledge in multi-task learning[29,30].

The PMd simultaneously encodes sensory evidence and decision-making behavior[31], and it can retrieve and maintain behavioral goals based on the visual stimuli without encoding the visual features of objects[32]. The PMd is closely connected anatomically to the parietal cortex, likely receiving visual information through this connection[33,34]. The PMd also receives limited input from the ventrolateral prefrontal cortex (vlPFC) and dorsolateral prefrontal cortex (dlPFC), both of which are involved in processing visual object information[35–37]. It is conceivable that the PMd encodes visual stimulus-related information. Thus, the PMd may combine visual information with behavioral goals, thereby playing a pivotal role in goal-directed motor planning and

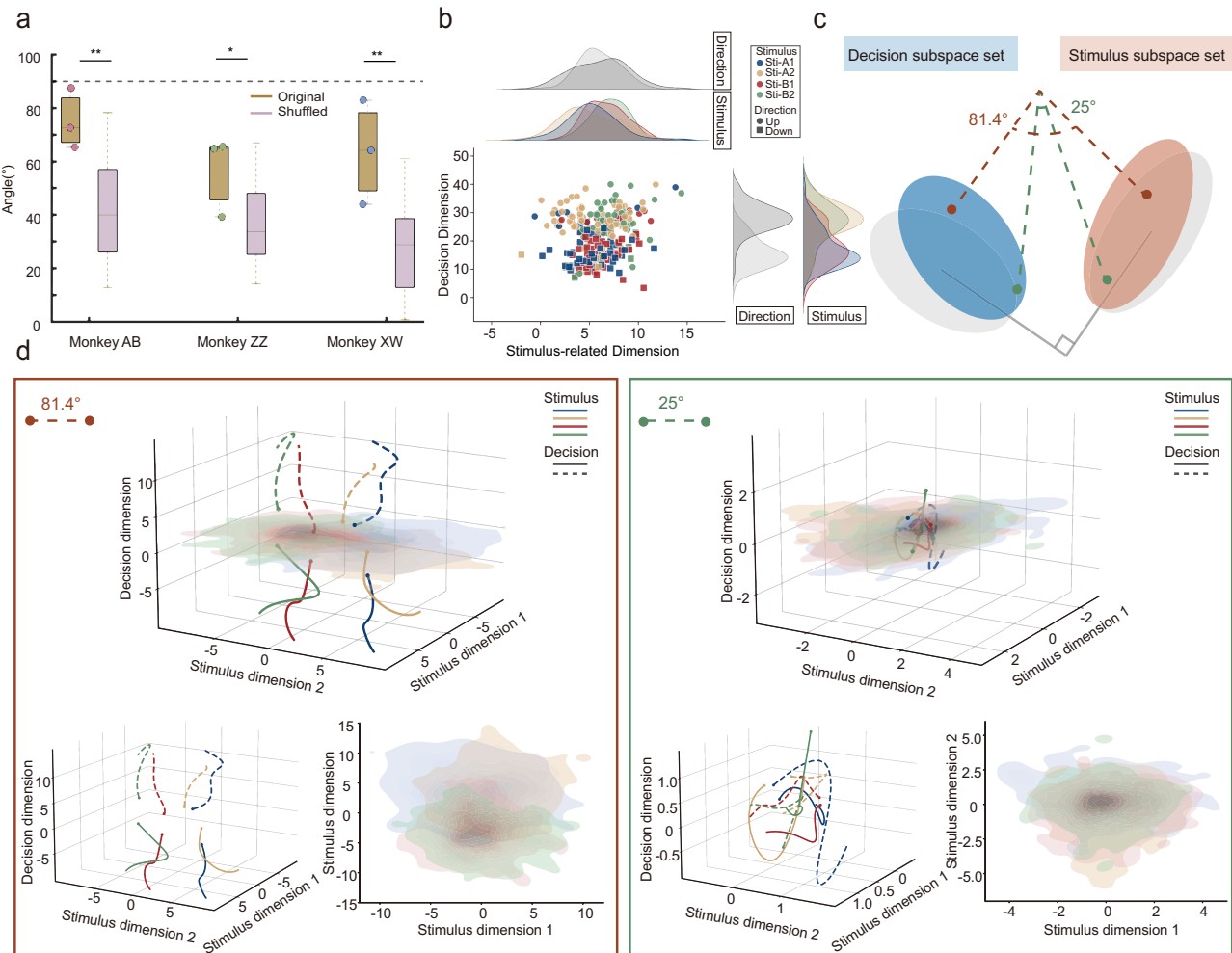

**Fig. 5 | Near-orthogonal relationship between decision- and stimulus-related subspace. a** The angle between the stimulus- and the decision-related subspace was compared with angles obtained after shuffling the decision and stimulus labels. Data are shown as box-and-whisker plots. The line within each box indicates the median; box edges represent the first (Q1) and third (Q3) quartiles; whiskers show the full data range. Colored scatter dots represent individual sessions, with each color corresponding to one monkey. (Original group: $n = 3$ per monkey; Shuffled group: $n = 30$ per monkey; Monkey ZZ: $P = 0.04$, $g = 1.27$; Monkey AB: $P = 0.005$, $g = 1.77$; Monkey XW: $P = 0.002$, $g = 2.02$). **b** Distribution of data for individual trials of monkey AB on the stimulus-related and decision plane. **c** Illustration of the angle

between the decision- and stimulus-related subspaces. Blue and orange areas represent decision- and stimulus-related subspace distributions, respectively. Red and green dashed lines mark example angles of 81.4° and 25°, with a gray right angle indicating near-orthogonality. **d** Neural trajectories in decision- and stimulus-related subspaces at different angles. Left: Neural trajectories and stimulus projections when the subspace angle is 81.4°. Right: Neural trajectories for a selected case at 25°. Lines represent trial-averaged trajectories, and shaded areas represent the variability across trials for each stimulus. *, $P < 0.05$, **, $P < 0.01$, ***, $P < 0.001$, Watson–Williams test.

execution[38]. There is evidence that high-level cortical areas such as the motor, premotor, and prefrontal cortices have the ability to encode and integrate information across multiple dimensions, providing the neural foundation for complex and flexible cognitive functions[39,40]. Orthogonal coding is key to the parallel processing of multidimensional information[41]. Evidence of orthogonal coding representing different attributes of objects and contextual information has been found in several brain regions[42,43]. For example, Yang et al. identified orthogonal coding of visual and motor information in the visual cortex[44]. Zhang Y et al. discovered orthogonal coding of sensory and motor information in the motor cortex of monkeys[45]. Flesch et al. show that task identity, stimulus–response, and decision variables are encoded in orthogonal low-dimensional manifolds rather than within a single shared subspace, in both human prefrontal and posterior parietal cortices[46]. These findings suggest that near-orthogonal coding may be a widespread form of representation in the brain, enabling parallel computation. Our finding adds to this understanding by revealing that near-orthogonal coding can minimize interference

among various domains. This is critical for effective compartmentalization of different processing even within the same cortical region, and as the example we showed here it supports flexible use of domain-specific schemas.

Interestingly, our study also highlighted the challenges faced by monkeys in reversal learning tasks, which differ from those in similar learning situations. Reverse tasks are an important experimental paradigm for assessing cognitive flexibility[47]. The specific neural mechanisms underlying reversal learning remain incompletely understood. In the field of cognitive psychology, some studies have proposed an inseparable relationship between cognitive flexibility and schemas[48]. Our results showed that the lack of NCS in reverse tasks suggests that learning reversal mapping rules may rely on different neural mechanisms (see Supplementary Notes 1.11 for further discussion). These differences prompt further consideration of the relationship between cognitive flexibility and schema utilization. This is consistent with previous psychological studies indicating that disrupting schemas may enhance cognitive flexibility[49]. The observed

impairment in reversal learning aligns with prior findings that reward reduction induces behavioral adjustment difficulties, with the anterior cingulate cortex (ACC) critically involved in triggering adaptation signals[50,51]. These results suggest a possible distributed functional model in which the ACC monitors reward history and initiates adaptation, while downstream regions such as the PMd implement reorganization of action representations.

Previous studies have found NCS mostly in higher cognitive brain regions such as the orbitofrontal cortex, hippocampus, and prefrontal cortex[52]. However, we also observed the reuse of NCS in the PMd, a region closely associated with cognitive flexibility and motor control. Given that NCS facilitate the reuse of prior knowledge in adaptive learning, it is plausible that this representational pattern is not confined to specific brain regions, but is a fundamental organizational principle that extends across various cortical areas. The ability to preserve and adapt previously learned rules could be critical for many cognitive processes, such as decision-making, sensory processing, and motor control, implying that NCS may be a general organizational principle for efficiently managing cognitive flexibility and stability in a wide range of tasks.

In summary, our results provide evidence for a computational framework for resolving the stability-plasticity dilemma by embedding reusable NCS in specialized subspaces, ensuring both the preservation of prior knowledge and the adaptability required for new learning. This computational framework not only deepens our understanding of schema-based learning in the brain but also suggests potential strategies for artificial intelligence systems designed to balance stability and flexibility across multiple tasks. Moreover, the evidence of NCS in the PMd supports the hypothesis that schemas may be a universal organizing principle across cortical regions, involved in preserving and adapting knowledge for diverse cognitive and motor functions.

## Methods

### Animals and experiment conditions

Three male macaques (monkey AB weights 9.8 kg, aged 9 years; monkey ZZ weights 10.6 kg, aged 10 years; monkey XW weights 8.9 kg, aged 6 years) were trained to perform visuomotor mapping tasks. All experimental protocols were approved by the Animal Care and Use Committee of the Institute of Automation, Chinese Academy of Sciences, in accordance with ethical guidelines. During the task, the monkey was seated in a chair, with heads fixed through a surgically implanted titanium post and left hands were constrained. A 17-inch capacity touch screen (60 Hz frequency; 1024 × 768 dpi) was placed in front of the monkey. The animals used their right hands to touch the screen to perform the tasks. A fixed grip bar with two built-in infrared sensors (EE-SPY302-1, AOYOU) was mounted on the front of the monkey chair. The infrared sensor detected whether the monkey's hand was gripping the bar.

Monkey XW's left PMd cortex was implanted with a 96-channel Utah array (400 um pitch, electrode length 1 mm; Blackrock, USA). Monkey AB and ZZ's left PMd cortices were implanted with two 48-channel Utah arrays (double-headed electrode with 48 channels per head, 400 um pitch, electrode length 1 mm; Blackrock, USA) (Supplementary Fig. 1). A Cereplex Direct (Blackrock, USA) multi-channel data acquisition system was used to collect data in the frequency range of 0.1 Hz - 5500 Hz with a sampling rate of 30,000 Hz. The parameters in the Mountainsort[53] program were customized to automatically classify single-unit and multi-unit. Specifically, a 250 Hz Butterworth high-pass filter was used to process the raw data. A threshold of -5.5 times the standard deviation was used to detect action potentials within each electrode. When the distance between an action potential waveform and the cluster it belongs to was greater than or less than 3 times the MSE (mean squared error) of the cluster distribution, the action potential waveform was defined as background noise. A unit without an inter-spike interval (ISI) of less than 2 ms and an isolation

score greater than 2.33 from other clusters was defined as a single unit. The rest were marked as multiple units. SU and MU were pooled together to form the population activities, which were analyzed in the present study. We applied the same spike sorting pipeline to all tasks within the same session using Offline Sorter (Plexon, USA). For each session, spike sorting was performed once across the entire recording, and the same waveform templates and classification parameters were used across all tasks. This ensured that the same units were tracked consistently across tasks within a session. A total of 728 units (350 SUs, 378 MUs) were recorded across three monkeys over three sessions. The mean (± SD) unit counts per session were: Monkey XW: 75 ± 14 SUs, 82 ± 4.5 MUs; Monkey ZZ: 17 ± 4.4 SUs, 26.3 ± 3.1 MUs; Monkey AB: 24.7 ± 5.5 SUs, 17.7 ± 9.1 MUs.

### Behavior tasks

The monkeys were required to learn a series of new visual stimuli-action pairs, revisit a previously learned pair, and learn the reverse mapping of a previously learned pair (Fig. 1a). At the beginning of each session, a red dot appeared in the middle of the monitor. When the monkey held the bar for 1 second, the red dot disappeared, and a visual stimulus was presented on the screen for 1 second. Immediately after the offset of stimulus, two buttons appeared on the upper-right and lower-right positions of the screen as the Go Cue, prompting the monkey to release the bar and start hand moving. The monkey needed to touch the correct button according to the visual stimulus presented within 5 seconds to receive a juice reward. Incorrect trials were indicated by a specific sound followed by a black screen for 5 seconds.

### Data analysis

Behavioral data were collected across multiple sessions (Monkey AB: 5 sessions, 383.8 ± 106.2; Monkey XW: 5 sessions, 225.4 ± 33.2; Monkey ZZ: 4 sessions, 390.3 ± 105.83 trials per session, mean ± SD). Specifically, the trial distribution across different tasks was as follows: monkey AB completed 113.0 ± 41.7 trials in pair A, 94.0 ± 21.7 trials in pair B, 50.0 ± 0.0 trials in Revisit A, and 163.7 ± 103.6 trials in Reverse task; monkey ZZ completed 109.3 ± 42.6 trials in pair A, 52.7 ± 2.1 trials in pair B, 50.3 ± 0.6 trials in Revisit A, and 144.7 ± 68.7 trials in Reverse task; monkey XW completed 54.7 ± 6.4 trials in pair A, 51.0 ± 0.0 trials in pair B, 51.7 ± 1.2 trials in pair C, and 51.3 ± 3.2 trials in Revisit A. For behavioral data analysis, task acquisition was defined as the point when accuracy reached 90% within a sliding window of 15 trials. For reverse tasks that were not fully acquired, we used the total number of trials completed as the measure of trials required for learning. To account for session-to-session variations in task engagement, trial counts were normalized by the total trials completed on each session. Reaction time (RT), defined as the time it takes for the monkey to leave the bar after seeing the go cue and touch the selected button, was calculated for different tasks. To evaluate significant differences in RT between different tasks, a two-sample t-test was employed ($P > 0.05$ to reject; *, $P < 0.05$; **, $P < 0.01$; ***, $P < 0.001$).The neural activity data from three monkeys over three sessions were analyzed (monkey AB/ ZZ/XW completed 420.7 ± 129.1/ 357.0 ± 109.3 / 209.0 ± 6.6 trials per session, mean ± SD). Our neural analyses were conducted at the session level (session-by-session, animal-by-animal analysis) rather than pooling all neural data together. The spike data were structured into a continuous binary raw format within one second before the onset of Go cue. The spike count was binned at 10 ms intervals, resulting in a matrix, $\mathbf{X} \in \mathbb{R}^{M \times N \times T}$, where $M$ is the number of trials, $N$ is the number of neurons, and $T$ is the number of time bins. To infer low-dimensional, denoised representations of neural population activity, we applied LFADS[25] to spiking data binned at 10 ms. LFADS models neural activity as the output of a latent dynamical system, using a sequential variational autoencoder with an encoder RNN to estimate trial-specific initial conditions and a decoder RNN to simulate latent trajectories. A linear readout maps latent factors to firing rates to

reconstruct observed spikes. For each recording session, LFADS was trained separately to extract 16-dimensional latent trajectories (see Supplementary Notes 1.3 for detailed justification of dimensionality selection). This approach preserved task-relevant temporal structure while reducing trial-to-trial variability, enabling downstream analyses in a common latent space. The low-dimensional neural data matrix was represented as $\mathbf{X}_l \in \mathbb{R}^{M \times 16 \times T}$. Subsequently, a two-layer CNN was used to classify the reduced neural data in response to image stimuli and motor decisions in similar learning tasks(Eq. 1).

$$
\begin{aligned}
H_1 &= P(f(X*W_1 + b_1)) \\
H_2 &= P(f(H_1*W_2 + b_2)) \\
H_2' &= D(H_2, p_1) \\
Y &= \varphi(H_2')
\end{aligned}
\tag{1}
$$

where * denotes one-dimensional convolution along the temporal axis. $W_1, W_2$ are learnable convolutional filters in the first and second layers. $b_1, b_2$ are bias terms added after convolution. $f(\bullet)$ is an element-wise nonlinearity, implemented as ReLU (Rectified Linear Unit). $P(\bullet)$ indicates max-pooling applied after activation. $D(H; p)$ denotes dropout with drop probability $p$, applied to reduce overfitting. $\varphi(\bullet)$ represents softmax output mapping. The network architecture for classification consists of two convolutional layers, each with n_filter = 8, kernel = 3, stride = 1, padding = 1, and one fully connected layer, with dropout rate = 0.5. Data from the series of new stimulus-action pair learning tasks were used to train both the visual stimulus classifier (Note: Monkey XW classified 6 categories of images, while monkeys ZZ and AB classified 4 categories) and the motor decision classifier (all three monkeys classified 2 categories, up and down). The dataset was divided into 80% for training and 20% for testing. With the stimulus model parameters $W$ fixed, the unseen neural data from the revisit-A task were fed into the same CNN model to classify visual stimuli. Similarly, with the decision model parameters fixed, the neural data from the revisit-A task were fed into the same model for decision classification. To further validate, a decision classifier was trained using only the data from task A. The fixed-parameter model was then tested using neural data from task B/ C and revisit-A.

Manifolds related to motor decision-making and visual stimuli were derived from the neural activity in the PMd[26]. Spike trains for each trial were smoothed using a Gaussian kernel ($\sigma = 50ms$). Neural data were represented as $\mathbf{X} \in \mathbb{R}^{N \times S \times D \times M}$, where $M$ is the maximum number of trials across all conditions, $N$ is the number of neurons, $S$ is the type of visual stimulus, and $D$ is the direction of motor decision-making. Using dPCA to project neural data into different dimensions. Trials under different conditions were averaged to obtain $\mathbf{X}_{ave} \in \mathbb{R}^{N \times S \times D}$, which was then marginalized[54] into different factors: $X_{ave} = x_t + x_{ts} + x_{td} + x_{tsd} + x_{noise}$, where $x_t$ represents condition independent term, $x_{ts}$ the stimulus term, $x_{td}$ the decision term, $x_{tsd}$ the stimulus-decision interaction term, and $x_{noise}$ the noise term. Dimensionality reduction was then applied to the demixed components to extract the subspace $\mathbf{Y} \in \mathbb{R}^{p \times S \times D \times T}$, where $p$ is the number of principal components (PCs) used for further analysis ($p = 20$), with each dimension corresponding to a specific term noted before (see Supplementary Notes 1.6 for detailed justification of dimensionality selection). The eigenvector matrix was denoted as $\mathbf{W} \in \mathbb{R}^{N \times p}$. The decision dimension was referred to as the decision subspace $\mathbf{Y}_{decsion} \in \mathbb{R}^{p_{decision} \times T}$, where $p_{decision}$ is the number of decision components, and the stimulus dimension as the stimulus-related subspace $\mathbf{Y}_{stimuli} \in \mathbb{R}^{p_{stimuli} \times T}$, where $p_{stimuli}$ is the number of stimulus components (see Supplementary Notes 1.5 for more mathematical details about dPCA). For further analysis of manifold reuse, dPCA was performed on the neural data only from task A to obtain the projection matrix $W_A$. Neural data from tasks B/C and revisit-A were projected into task A's subspaces using the same matrix $W_A$.

In the decision subspace $Y_{decsion}$, the dimension with the highest proportion of variance $\mathbf{S} \in \mathbb{R}^{S \times D \times T}$ was selected. The Pearson correlation coefficients $rho_{i,j}$ between the upward and downward decision manifolds in the similar learning tasks were calculated, where cov represents covariance, $rho$ is the Pearson correlation coefficient, $i$ refers to task A, $j$ refers to task B/Revisit-A/Reverse-A, $c$ indicates whether the motor decision direction was up or down, and $\sigma_i^c$ is the variance of $S_i^c$. The correlation was calculated as:

$$
rho\left(S_i^c, S_j^c\right) = \frac{cov\left(S_i^c, S_j^c\right)}{\sqrt{\sigma_i^{c2} \sigma_j^{c2}}}
\tag{2}
$$

The average correlation coefficient between the two decisions was taken as the decision manifold correlation coefficient to measure the degree of manifold reuse. A two-sample t-test was conducted to assess significant differences in the decision manifold correlation coefficient between the task Reverse-A and the tasks A, B/C, and Revisit-A. Linear regression analysis was performed on the number of trials to criterion and the decision manifold correlation coefficient, and the p-value and the linear regression coefficient $R^2$ were calculated.

The angle between the stimulus-related subspace and the motor decision subspace was computed using the MATLAB subspace function, which calculates the principal angle between two subspaces. For the stimulus-related subspace, which had a higher dimensionality (Supplementary Fig. 13), we selected the components that explained more than 1% of the variance and together accounted for at least 70% of the total variance in that subspace. For the decision subspace, components with more than 1% variance explained were included, without an additional cumulative threshold, due to its lower overall dimensionality (Supplementary Fig. 13). The angle between the stimulus-related subspace and the motor decision subspace was calculated using the MATLAB subspace function:

$$
\theta = \arccos\left(\sigma_{max}\left(U_d^\top U_s\right)\right)
\tag{3}
$$

where $\mathbf{U}_d \in \mathbb{R}^{n \times d}$ is an orthonormal basis for the decision subspace, obtained via singular value decomposition (SVD) of neural activity aligned to decision variables. $\mathbf{U}_s \in \mathbb{R}^{n \times s}$ is an orthonormal basis for the stimulus-related subspace. $\theta$ represents the smallest principal angle between the two subspaces, indicating their relative orientation in neural representational space. To test whether the observed near-orthogonality between stimulus and decision subspaces exceeded chance levels, we randomly permuted the stimulus and decision labels independently across trials and recalculated the angle between the subspaces. Repeating this procedure 10 times generated a null distribution of subspace angles under the assumption that there was no true relationship between stimulus and decision encoding. The observed angle was then compared against this distribution to assess whether the near orthogonality exceeded chance levels. Watson–Williams test was conducted for significance testing. The three-dimensional neural dynamics schematic selected as an example corresponds to an angle of 81.4° between the stimulus-related subspace and decision-related subspace for monkey AB, while the counterexample involves an angle of 25° between dimensions of decision and visual stimuli that explain less than 1% variance for monkey AB.

## Inclusion & Ethics statement

The experimental plan was approved by the Biomedical Research Ethics Review Committee of the Institute of Automation, Chinese Academy of Sciences (approval number: IA-201904). All experimental and surgical procedures involving animals were performed in accordance with the NIH Guide for the Care and Use of Laboratory Animals.

## Reporting summary

Further information on research design is available in the Nature Portfolio Reporting Summary linked to this article.

## Data availability

The processed data used to generate results in this study are provided in the Source Data file. The raw electrophysiological data are available under restricted access due to institutional policies. Access can be requested by contacting the corresponding author at shan.yu@nlpr.ia.ac.cn. Requests will be responded to within two weeks, and data will remain available for one year after access is granted. Source data are provided with this paper.

## Code availability

The code used to analyze the data and generate the figures in this study is available on GitHub at https://github.com/kathytian-kx/DomainSpecific_SchemaReuse.git, and has been deposited in Zenodo (https://doi.org/10.5281/zenodo.17909108). The repository includes data preprocessing scripts, decoding algorithms, dPCA implementation, manifold analysis routines, and figure generation code. Additional analysis code is available from the corresponding author upon reasonable request.

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

## Acknowledgements

We thank Zichen Liu for technical assistance and Prof. Jingfeng Zhou for valuable suggestions. This work was supported by the STI 2030—Major Project 2021ZD0200402 to S.Y., the Strategic Priority Research Program of the Chinese Academy of Sciences (CAS) (XDB1010302) to S.Y. and the CAS Project for Young Scientists in Basic Research (Grant No. YSBR-041) to Y.C.

## Author contributions

K.T. conceived the project with Y.C. and S.Y., and wrote the manuscript with S.Y.; Z.Z., X.H., S.C., J.G., and K.T. performed the implantation surgeries; Z.Z. and S.C. conducted behavioral experiments and data recording; Z.Z., S.C., and K.T. preprocessed neural recordings; K.T. wrote analysis code, analyzed data, and visualized results; N.G. contributed to analysis code and visualization for orthogonal results; S.Y. and J.G. supervised the project. All authors commented on the manuscript.

## Competing interests

The authors declare no competing interests.
