## [Transparent Peer Review file · Nature Communications]

Domain-specific Schema Reuse Supports Flexible Learning to Learn in the Primate Brain

Corresponding Author: Professor Shan Yu

Version 0:

Reviewer comments:

Reviewer #1

(Remarks to the Author)

This work from Tian, Zhao et al. focuses on the characterization of stimulus- and choice-related neural activity patterns in the dorsolateral premotor cortex (PMd) of three monkeys performing sensorimotor behavioral tasks. The authors use tasks requiring animals to learn and relearn different stimulus-choice pairings, investigating how PMd activity supports learning new associations while avoiding forgetting previously learned ones. Using computational tools such as LFADS and dPCA, the authors identify a “schema-like manifold” in the stimulus and decision subspaces. They show that the decision subspace is more stable across tasks than the stimulus subspace and that these subspaces tend to be orthogonal, potentially facilitating flexible task-specific processing.

The study addresses a highly relevant topic with a well-designed task paradigm that explores flexible representations of task-specific information. Overall, the methods appear robust, but some key details of the analysis are either missing or difficult to extract from the manuscript. As detailed below, additional information and controls are needed to ensure that the main conclusions are not artifacts of the chosen analyses but instead offer genuine insights into PMd neural coding. I also found the description of the task sequence and temporal structure a bit unclear, particularly regarding the number of sessions and the inclusion of data in the analysis. Finally, more details about the animals’ motor behavior would be helpful, as changes in posture or button-pressing could also contribute to the observed neural changes.

Major points:

The manuscript lacks sufficient information about the number of neurons, sessions, and trials used for each analysis. While the total of 948 units and the mean number of trials per day are reported, it is unclear if 948 represents the total across all sessions or per animal. The authors should specify the relevant sample sizes (neurons, trials, PCs, sessions) for all key analyses, especially where error bars are presented.

I found the description of the task design unclear, particularly regarding the number of sessions for each task condition. Did the animals learn a new task each day, meaning only one recording session per animal was available for each task condition? If so, what does the variance shading in panel 1f represent? The figure shows performance improvements across trials, but it is unclear whether these reflect a single session for each task type or a specific phase of training. If the shading represents variance across days, how many days were included in the analysis? Additionally, to what extent does general task experience influence the results? Were the animals already fully trained in the overall task structure when the recordings were performed?

It is crucial to provide a clearer explanation on these points to better understand the neural results. For instance, in Fig. 3, it is unclear to me how the authors can plot multiple points for each condition if each data point represents a single session. Similarly, how many days are included in Fig. 4a? These aspects require more detailed clarification.

Lines 182–185: Additional information on the orthogonality analysis should be provided in the results section to improve clarity. Specifically, the authors should explain how the components for each subspace were selected, why exactly 20 components were used, and how the angles between them were calculated. This would make the rationale behind the analysis more understandable. More broadly, it would be helpful to provide better explanations for why specific analyses were chosen. For example, what advantages does LFADS offer compared to using the raw neural firing rates directly? Providing this context would help the reader appreciate the motivation and implications of the chosen methods.

I also find it unclear how the data were transformed into specific components related to the decision, stimulus, and stimulus-

decision terms (lines 334–335 in the methods). This appears to be a crucial step for the subsequent analyses and should therefore be explained in greater detail. Is this transformation based on a linear decomposition? More broadly, I question how effectively this approach can separate stimulus-related from choice-related activity, given that the animals' choices are derived from the sensory information. Since the stimulus responses are predictive of the choice, especially in the later tasks with high behavioral performance, it is unclear how the authors effectively disentangle these components.

On a related note, if the analysis enforces orthogonality between the stimulus and choice responses to separate them, could this methodological constraint explain the main results shown in Fig. 4? This is not necessarily a criticism but highlights the need for better explanation of the analysis, as well as potentially additional controls to address this concern. For instance, if the shuffle control simply mixes stimulus and choice components, it is unsurprising that the principal components (PCs) become less orthogonal. Moreover, if this orthogonality was originally imposed by dPCA or another analysis step, I am not convinced that this conclusively supports the authors' claim about orthogonal neural coding in PMd being an important computational principle for learning. Clarifying this point and providing further justification or controls would greatly strengthen the study's conclusions here.

Fig. 3 shows a reduction in decoder accuracy across tasks, but it is unclear how a control condition for a network with stable neural representations would look like. To what extent could the observed reduction in accuracy simply be explained by the fact that different neurons were recorded on different days? Do the authors assume that they recorded from the same neurons with consistent quality to enable a pre-trained decoder to generalize across sessions? If so, how was spike sorting performed to support this assumption? More broadly, I wonder to what extent it is reasonable to expect that a decoder could work consistently across sessions if no specific measures were taken to ensure that the same neurons were reliably recorded across all tasks. Additional clarification on this point, as well as any steps taken to address variability in recorded neural populations between sessions, would provide important context for interpreting these results.

Line 132 states that the decision classifier exhibited consistent performance for unseen data, but the decoder performance drops for both stimulus and choice, particularly in monkeys AB and ZZ. This wording seems a bit misleading here. Moreover, a potential explanation for the difference between stimulus and choice decoder performance could be that stimulus information is only weakly represented in PMd neurons, with larger variability across units, compared to choice information. This could not only explain the initial difference in decoder performance but also why the decoder would be less robust across days. Changes in recording quality and the units recorded could affect an unstable stimulus decoder more rapidly than a more stable choice decoder, which may have greater redundancy across units. The authors could test this hypothesis by subselecting neurons so that stimulus and choice decoder performance are initially equal for task A and then demonstrating that only the stimulus decoder performance is reduced in the revisit A condition.

Another potential control would be to test if the choice decoder is more stable across days because neural tuning is more redundant across neurons for choice than for stimulus. In this case, signal correlations for choice should be higher across PMd neurons. If the authors show that this is not the case, it would argue against this explanation and provide support for the claim that the results are due to a difference in representational stability across time, rather than a more general difference in tuning redundancy between stimulus and choice.

Given that behavioral performance in the reversal task is lower and requires more time to relearn, this seems like a good opportunity to test if the neural representations of stimulus and choice change as the animals' performance improves again. Do schemas become more comparable to the patterns observed in the earlier tasks? If so, this would provide strong reassurance that the results are not driven by technical issues such as the loss of recording from the same neurons or other experimental constraints, but instead reflect the use of behaviorally-relevant activity patterns that promote learning.

Figure 1e: What do the error bars represent, and why are conditions B and revisit-A shown together?

Lines 158-159: "since the movement directions in this task were the same." It would be helpful to quantitatively demonstrate that this is the case and that movement patterns do not change as a result of learning. Can the authors provide some limb tracking data or similar evidence to support this claim?

It would be beneficial to extend the raster plot in Fig. 2a to show more of the actual neural response patterns, such as trial averages for the different stimulus conditions, to provide a clearer picture of the neural tuning of PMd neurons to stimulus and choice before the subsequent dimensionality reduction. This would help to better understand the single-neuron activity before moving on to the more abstract latent factors used in the analysis. Is there a general preference for PMd neurons to respond to the upper or lower position of the response button on the screen? To what extent are decision- and stimulus-related signals represented in PMd neurons when the animals have not learned the task? Do task-specific responses emerge as behavioral performance improves, or are stimulus- and motor-related signals always present?

Minor:

Lines 82-83: 'we found that learning efficiency increased for later-learned pairs'. Where is this shown concretely? In monkey AB, for example, only the revisit-A condition seems to show faster learning, since the animal already knows this association. In contrast, monkey XW only shows faster learning in condition B but not revisit-A. Does this refer to the results in panel 1e?

In reference 21, the author name is Driscoll et al. Laura is her first name.

line 183: what is 'subspace t' ?

line 212: What are schema-like neural activities? Are these different from actual schema as described by the authors?

Fig. 2e: What do the labels St 1 and St 2 mean here? Do those refer to the two stimuli that were used in task A and revisit A?

Lines 180 and 223 - 225: I do not think that the analysis reveals a neural mechanism as the results are purely phenomenological. That is not a big issue but I would suggest rewording this claim to avoid it from being misleading.

Reviewer #2

(Remarks to the Author)

In the manuscript entitled "Domain-specific schema reuse supports flexible learning to learn in primate brain", Kaixi Tian and colleagues investigate population activity in dorsal premotor cortex of macaque monkeys in the context of several sensory-motor mapping tasks. They identify two orthogonal subspaces, one related to stimulus encoding and one related to motor responses / decisions. While the former is surprisingly instable, the latter replicates across tasks with a similar structure, but not a reversal learning task. On the basis of a correlation between learning speed and the similarity of subspaces across tasks, the authors suggest the preserved subspace constitutes a schema that generalizes across tasks and thus forms the basis of learning to learn. This is an interesting manuscript that addresses a timely question in an appropriate animal model. I have several suggestions for the authors that they might consider to further improve the manuscript.

1. The text as well as the figures are lacking statistical detail / quantification. For example, nowhere in the text do the authors mention significance, effects sizes, or the tests that they applied. This should be provided throughout the manuscript. In the figures, no information is provided what the error bars depict. It is thus hard to impossible to assess how reliable and/or strong the reported effects are. Furthermore, it is unclear why the authors used t-tests and not a more appropriate circular statistic for the analyses of angles. The classification performance should be cross-validated. Furthermore, some statistical quantification of the similarity of results across the three animals should be provided.
2. The authors report that the stimulus subspace is rather instable. This is surprising and should be further investigated. Since the stimuli in task A and in revisit A are the same, one would expect that the classifier performs above chance at least in this case. If the result holds, then this would possibly argue against orthogonal decision and stimulus subspaces, since it suggests that the stimulus subspace contains a part of the sensory-motor mapping.
3. The authors should provide a definition of "schema" in the introduction and rethink whether "specific neural activity patterns" are indeed referred to as schemas. To me, this seems to be mixing the neural with the cognitive level. Perhaps, specific neural activity patterns underly schemas, but that they are schemas themselves seems a bit of a stretch.
4. The authors should report separately how many multiunit sites and how many single units entered the analysis. The compound number given at the moment is not very informative.

Reviewer #3

(Remarks to the Author)

In this study, the authors asked if dorsolateral premotor cortex (PMd) could learn/generate task-specific knowledge (schema in the paper) and reuse it for other tasks. They trained three of nonhuman primates (NHPs) on a sequence of visuomotor tasks, which require flexible mappings between visual stimuli and behavioral responses, and recorded multi- and single-unit activities (MUAs and SUAs) from PMd. The authors used convolutional neural networks (CNNs) to decode decisions and stimuli from PMd's MUAs and found that PMd encodes both visual stimuli and decisions. Interestingly, their analyses suggest that only decisions appear to be stored as schema-like knowledge. Furthermore, they used demixed PCA to estimate stimulus-subspace and decision-subspace from MUAs, which provides interesting results.

These findings may provide valuable insights into reward-based learning and decision-making and interest readers of neuroscience studies, but I have a few concerns and questions.

Concerns regarding the lack of detailed description of methods:

1. The authors used LFADS to obtain low-dimensional (dim=16) representations of MUAs, and I have three concerns regarding LFADS. First, the authors need to provide a brief review of LFADS. Second, the dimension of MUAs is reduced to 16 after LFADS. The authors should explain how they came to choose this number. Third, the authors used CNNs to decode decisions and stimuli from MUAs. Given the strong learning capability of CNNs, I assume CNNs can decode them even without LFADS. If so, why did the authors employ an additional step, which can cause an unexpected artifact?
2. The analysis of subspace is one of the main results in this study, but I do not see a sufficient amount of details on the analysis. I think this may be because the authors adopted the demixed PCA proposed by an earlier study (reference 52 in the manuscript). While this may be clear to the authors themselves, it is still advisable for them to provide explanations for their readers. Also, the authors stated in line 333, "Trials under different conditions were averaged to obtain x_{avg} ..., which was then marginalized [53]". As far as I can see, they used the marginalization proposed in reference 52, so the reference should be 52.
3. In their analysis, the authors used 20 principal components. I would like to know how they reached the conclusion that 20 components are good enough.

Questions:

1. The two earlier studies [1, 2] showed that anterior cingulate cortex (ACC) was responsive to the reduction of rewards and essential in maximizing the rewards. The experimental protocol used in the two earlier studies is similar to the protocol used in this study (switching from Task-A to Reverse-A) in my opinion. Can the authors provide any insights into the relationships between their study and the two earlier studies [1, 2]?

2. In lines 183-185, the authors state “we identified the major components in these two subspaces that explained the highest variance in the data, and then calculated the angle between them (see methods for details)”. Does this mean that they used a single principle component of each subspace? If so, how much information is lost by ignoring other components? Can the authors provide any insights?

[1] Shima, K., and Tanji, J. (1998). Role for cingulate motor area cells in voluntary movement selection based on reward. *Science* 282, 1335–1338.

[2] Walton, M.E., Behrens, T.E.J., Buckley, M.J., Rushworth, M.F.S., and Kennerley, S.W. (2006). Optimal decision making and the anterior cingulate cortex. *Nat. Neurosci.* 9, 940–947.

Minor concerns:

1. Eq. 1 is not very informative. It should be modified to clarify that the two convolutional layers are used. A more explicit expression for convolution would also be more helpful.
2. In line 312, the authors stated “T is the number of spikes across time”. I think they meant to say “T” is the number of time bins instead of the number of spikes. Please clarify this point.
3. Figure 1 contains too much information. It may be more readable to the readers if they split it up into several figures.
4. I think they should include all stimuli shown in supplemental Fig. 1 in the main text.
5. The legend of Fig. 2D can be more explicit. The term “direction” is too vague here. Perhaps, “up” and “down” may be more appropriate.

Version 1:

Reviewer comments:

Reviewer #1

(Remarks to the Author)

The authors have done a great job in addressing my main concerns and I appreciate the numerous and thoughtful additions they have made in the revised manuscript. The revised version of the text is substantially improved and particularly benefits from the improved explanations of the experimental design, more detailed methods, and additional control analyses that support the main conclusions. In particular, the experimental design of changing the learning conditions within the same recording day is now easier to follow. Moreover, the rationale behind the subspace and orthogonality analyses is quite clear now, and the inclusion of decoder generalization and shuffled controls strengthens the interpretation of the dPCA results.

I therefore generally support the publication of the manuscript in its current form and think that it makes a useful contribution to our understanding of changes in neural coding due to short-term associative learning.

Below are a few minor points for the authors' discretion:

1. The new PSTHs are a helpful addition to illustrate the functional selectivity of individual neurons. However, the stimulus-selective examples are somewhat difficult to interpret visually, likely due to the superposition of four traces in each panel. Using a larger bin size or separating some of the stimulus conditions in two panels might make the tuning selectivity more visible. Also, the figure caption currently states that each panel shows one neuron, although multiple examples are included.

2. Supplementary Figure 7 showing the direction preference index is a useful addition. However, to further aid interpretability, it would be good if the x-axes across all panels were scaled consistently from -1 to 1. The distributions appear centered near 0, but I wonder if this reflects a large fraction of neurons with no spatial preference or simply a majority that do not respond to button presses at all. Was this analysis restricted to neurons that were significantly modulated by the button press? Some clarification on this point would be helpful.

Reviewer #2

(Remarks to the Author)

The authors' substantial revision have significantly improved the manuscript and have helped in clarifying the rationale, approach, and analyses. However, they have also brought several points to the fore that were not evident in the first submission.

- Most of the neural data seems to stem from monkey XW, yet the behavior of this animal seems to be quite different from the other two animals. Figure 1c suggests that neither the behavior in condition C nor in condition revisit A is similar to monkeys AB and ZZ. Condition reverse A was not acquired in XW. The authors seem to deal with this by excluding XW from the main

analysis. It seems that a better characterization of the overall behavior would be to include XW in the pool. Furthermore, it would be important to show that the neural results across all animals are not biased towards the idiosyncratic behavior of XW given that this animal contributed substantially more data.

- There are several inconsistencies regarding the stimulus representation in PMd. First, it seems that the results from the decoding analyses and the dPCA analyses are somewhat contradictory. The decoding analyses suggest that there is no “stable” representation of the stimuli, to the extent that the same stimuli cannot be decoded with the same classifier twice (A and revisit A). This is surprising, given that the authors themselves write that “PMd encodes the identity features of visual stimuli” (p11, line 314), and should be clarified. On the other hand, the dPCA analyses seem to suggest that stimulus representations are well separable, according to the authors (p9, line 265). How can the representations be well separable but instable? Here, the authors should also double check which figures they are referring to. At the moment, the reader is asked to look at Figure 4b to illustrate the separability, but this figure shows the decision space, not the stimulus space. The scatter plot in 5b does not show good separability in the stimulus space.

- The key finding is that stimulus and decision space are orthogonal, but this finding is currently not well supported by statistical analyses and bears some conceptual issues:

- o The authors should not only test the angular difference between original and shuffled data, but also whether the original data is significantly different from 90 deg; only this would somewhat support their main claim about orthogonality.
 - o The Watson-Williams test is actually not significant ($p=0.06$) in monkey ZZ according to standard practice. Interestingly, the authors use a different convention to indicate significance in figure 5A than in all their other main figures: in 5A, * is $p<0.1$, whereas elsewhere, * is $p<0.05$. The conventions should be the same for all plots and the authors should acknowledge that the results are actually not the same across the animals in this main analysis. Perhaps the abovementioned test will remedy the overall situation, but currently, I do not see clear evidence across animals that supports the main claim of the paper.
 - o The authors should explain how an instable subspace (stimulus) can be orthogonal to another, stable subspace (decision). It rather seems that the stimulus space is just noise.
- It is unclear to me why the authors suggest that the behavioral deficit in reversal A is explained by a schema and not by simple reversal learning of specific sensorimotor mappings previously learned in A. An abstract schema of the task would entail that one stimulus is mapped to an upward response and another stimulus is mapped to a downward response. Such a schema can explain the transfer between tasks, but it does not seem that this very abstract schema would predict interference in reversal A, since there is still a stimulus associated with up and another stimulus associated with down.

Minor:

- The authors should provide p-values for the decoding accuracies versus chance, e.g., in Figure 3c.
- P11, line 323 – “richer” than what?

Reviewer #3

(Remarks to the Author)

The authors significantly improved the manuscript and addressed my concerns.

Version 2:

Reviewer comments:

Reviewer #1

(Remarks to the Author)

The authors have further improved the manuscript in this revision. The updated visualizations are clearer and help convey the main findings more intuitively. The additional statistical analyses and clarifications regarding the conceptual framework also contribute to a more coherent presentation of the work. Overall, I think the manuscript has reached a strong and publishable state and adds valuable insight into how neural subspaces support flexible learning.

A very minor point for the authors' discretion: in the updated Fig. 2c & 2d, the individual session markers are somewhat small and the color scheme makes some markers difficult to distinguish, especially for readers with red-green color weakness. Increasing the marker size and choosing a more distinct palette could improve readability.

Manuscript NCOMMS-24-73204A

Response to Reviewers

We sincerely appreciate the reviewers' thoughtful and constructive feedback on our manuscript. We have
carefully considered each comment and substantially revised the manuscript in response, including new
analyses and expanded explanations. All relevant changes have been highlighted in blue in the revised
manuscript.

To highlight the most significant revisions:

- • All main text sections have been substantially revised to address reviewers' comments and
incorporate new insights.
- • Several new extended figures have been added to support our methodological choices and further
substantiate the findings.
- • Comprehensive statistical analyses now accompany all primary results, strengthening the
robustness of our conclusions.
- • Enhanced conceptual clarity by distinguishing the cognitive schema from its neural correlates and
clarifying the meaning of the stimulus-related subspace.

These changes provide a more refined understanding of how domain-specific schema representations
emerge in premotor cortex during learning and strengthen the paper's contributions to the field. Thanks to
the suggestions from all reviewers, these revisions significantly enhance the overall quality and impact of
our work.

We begin our response by addressing four major points that were brought up by multiple reviewers.

- 1. Methodological Basis of the demixed Principal Component Analysis (dPCA).
- 2. Rationale for selecting 20 dimensions in the dPCA analysis.
- 3. Justification for using Latent Factor Analysis via Dynamical Systems (LFADS) dimensionality
reduction rather than raw data for classification.
- 4. Clarifying the distinction between cognitive schema and their neural representations in our
framework.

After responding to these major points, we include a point-by-point response to the individual referees.

Page references are provided as follows:

- • Major Point: Page 1-9
- • Reviewer #1: Page 9-36
- • Reviewer #2: Page 36-44
- • Reviewer #3: Page 44-50

Major Point 1: 1. Methodological Basis of the dPCA.

Both Reviewer #1 and Reviewer #3 noted that our explanation of the dPCA lacked sufficient details. In the
 original manuscript, we applied dPCA, a method developed by Kobak et al. (2016, eLife), to decompose
 population neural activity into distinct task-related subspaces, the stimulus- and decision-related subspaces.
 This analysis revealed that neural dynamics in the decision-related subspace were consistent across tasks
 within a session, whereas those in the stimulus-related subspace exhibited substantial variability.

To further clarify the underlying methodology for reviewers and readers, we have now expanded our
 description of the dPCA. Specifically, we provide detailed information on:

(1) Theoretical Basis and Mathematical Framework

dPCA is a linear method based on marginalizing neural data into parameter-specific averages. Specifically,
 our data matrix X (neurons \times time points \times conditions) is decomposed as:

$$51 \quad X = X_t + X_{st} + X_{dt} + X_{sdt} + X_{noise} = \sum_{\phi} X_{\phi} + X_{noise}$$

Where X_t represents condition-independent activity, X_{st} captures stimulus-dependent variance, X_{dt}
 captures decision-dependent variance, and X_{sdt} captures stimulus-decision interactions. For each
 marginalization, dPCA finds decoder and encoder matrices (D_{ϕ} and F_{ϕ}) that minimize the reconstruction
 error between the marginalized data and the reconstructed data:

$$56 \quad L_{dPCA} = \sum_{\phi} \|X_{\phi} - F_{\phi} D_{\phi} X\|^2$$

The components are then ranked according to explained variance to select the dimensions that capture the
 most prominent signals in the data.

Notably, dPCA achieves variance separation through a marginalization procedure, which is same as
 factorial variance decomposition (Rutherford, 2001; Christensen, 2011). The key step involves computing
 condition-marginalized components by averaging across all time points for each condition and subtracting
 the overall mean, take stimulus as an example:

$$63 \quad X_s(c) = \frac{1}{T} \sum_{t=1}^T X(c, t) - \bar{X}$$

\bar{X} captured the mean neuronal responses across all conditions. This term reflects stimulus-specific
 variance after removing the global mean response. Similar calculations are performed for decision
 components and interaction components. Mathematically, this approach explicitly separates variance along
 different task parameters by defining these marginalized components before the dimensionality reduction
 step.

(2) Algorithmic implementation

The dPCA is a three-step reduced-rank regression procedure: First, the optimal linear mapping from
 condition labels to neural responses was computed using ordinary least squares regression. How the task

variables (e.g., stimulus, decision) are linearly related to the population neural activity is identified by this
 step. Next, principal component analysis (PCA) was applied to the result of this mapping to extract the top
 q components by which the most task-relevant variance is captured. The dimensionality is reduced by this
 step while the most informative structure in the data is preserved. Finally, the resulting low-rank mapping
 was decomposed into two matrices: one by which neural activity is projected into component space (the
 encoder), and another by which activity is reconstructed from this space (the decoder). The encoder and
 decoder pairs can be interpreted and are defined independently of how many components are chosen. A set
 of task-related axes in neural space is yielded by this approach, which are optimized to capture the variance
 associated with specific task variables while linear interpretability and computational efficiency are
 maintained.

*(3) How the method addresses the issue of data imbalance*

In applications, it is common to encounter the problem of imbalanced sample data. The dPCA itself
 incorporates measures to address issues arising from imbalanced data. This approach modifies the loss
 function to treat all parameter combinations as equally important regardless of their frequency:

$$86 \quad L_{\phi} = \left\| \tilde{X}_{\phi} - FD \tilde{X} \right\|^2 + SQT \left\| FDC_{noise}^{1/2} \right\|^2$$

Where \tilde{X} represents mean firing rates (PSTHs) that can be collected in a smaller matrix of size $N \times SQT$.
 As noted in Kobak et al.'s paper: "In the unbalanced case, we can directly use this last formulation where
 all occurrences of X have been replaced by \tilde{X} . This is especially useful for neural data, where some
 combinations of task parameters may occur more often than others. The 're-balanced' dPCA loss function
 treats all parameter combinations as equally important, independent of their occurrence frequency."

Through this explanation, readers can develop a clear understanding of the dPCA method. We have
 incorporated this expanded description into the revised manuscript (new Supplementary Notes 1.5).

References:

- Kobak, D. et al. Demixed principal component analysis of neural population data. *eLife* 5, e10989 (2016).
 Christensen, R. Multifactor analysis of variance. In: *Plane Answers to Complex Questions: The Theory of*
 *Linear Models*. Springer Texts in Statistics, Springer, New York, NY, 163–201 (2011).
 Rutherford, A. *Introducing ANOVA and ANCOVA: A GLM Approach*. Sage Publications, London (2001).

**Major Point 2: Rationale for selecting 20 dimensions in dPCA analysis**

Both Reviewer #1 and Reviewer #3 raise this point regarding our use of 20 components in the dPCA
 analysis. As all dPCA-related analyses in the manuscript employed a 20-dimensional reduction, we
 understand the need to clarify our rationale and to assess the impact of dimensionality choice on result
 stability. In response, we provide detailed information on:

*(1) How dPCA dimensionality affects the percentage of total neural variance explained*

First, we assessed how different dPCA dimensionality choices affect the explained neural variance across
all three monkeys (n=3 days for each monkey). As shown in new Supplementary Fig. 11a, the cumulative
explained variance increases as a function of dimensionality. At 20 dimensions, the model captured 66.98
$\pm 0.89\%$ of the variance for Monkey XW, $76.66 \pm 3.07\%$ for Monkey ZZ, and $86.24 \pm 1.15\%$ for Monkey
AB (mean \pm s.e.m., across three days). To determine a reasonable cutoff for dimensionality selection, we
calculated the marginal gain in explained variance per additional dimension (new Supplementary Fig. 11b).
Across dimensions 20–25, the average marginal gain across monkeys was $3.92 \pm 1.20\%$ (mean \pm s.e.m),
which fell below our empirical threshold of 5%, indicating diminishing returns in variance explanation
beyond this point. This is one of the key criteria for selecting 20 dimensions.

*(2) How the distribution of different component types varies with dimensionality*

Next, we analyzed how the proportion of different component types (stimulus, decision, and other) changes
with dimensionality. In new Supplementary Fig. 11c, we illustrated the three-day average proportion of
dimension counts for different dPCA components in three monkeys at varying dPCA dimensionalities (10–
35 dimensions). As the number of dPCA dimensions increases, there is a slight increase in the number of
decision- and stimulus-related dimensions. However, even at lower dimensionalities (e.g., 10 dimensions),
this analysis reliably captures at least one decision component and four stimulus components in all subjects.

*(3) Robustness of the decision subspace reuse across different dimensionalities*

As outlined in **Major Point 1**, dPCA separates neural data into different components by marginalizing task
factors, and dimensionality reduction is performed separately on each component. The reduced dimensions
are then selected based on variance contribution ranking. Therefore, increasing the total dimensionality
does not alter the composition of previously selected dimensions but simply adds additional lower-ranked
ones. Notably, even with only 10 dimensions, all three monkeys exhibited at least one decision-related
component, and the observed reuse of the decision subspace with the highest variance proportion remained
evident (new Supplementary Fig. 11d). This indicated that the choice of dimensionality does not affect our
findings regarding the reuse of the decision subspace manifold.

*(4) Whether orthogonal-like relationship between the stimulus- and decision-related subspaces would be*
*influenced by the dimensionality used in dPCA*

We further tested whether the orthogonal-like relationship between the stimulus- and decision-related
subspaces would be influenced by the dimensionality used in dPCA (new Supplementary Fig. 11e). When
the dPCA dimensionality was low (e.g., 10 dimensions), the extracted neural representations likely
contained insufficient information, resulting in no significant difference in orthogonality compared to the
shuffled condition, thereby limiting the accurate characterization of the orthogonal structure. As the
dimensionality increased, the orthogonal angle exhibited a more stable trend in both monkeys, with 20
dimensions providing a balance between retaining sufficient information and maintaining a significant
difference from the shuffled condition. At higher dimensions (e.g., 30, 35), although the orthogonal angle
remained stable, further increasing the dimensionality did not substantially enhance the discriminability of
orthogonality and might introduce additional noise. Therefore, we selected 20 dimensions as the optimal
parameter to achieve a balance between information preservation and analytical robustness.

These analyses indicate that our choice of 20 dimensions achieves a balance between capturing sufficient
neural variance and ensuring computational tractability. Most importantly, our key findings regarding

domain-specific patterns, such as representation and reuse, as well as the near-orthogonality between the
 decision- and stimulus-related subspaces, remain robust. We have added a detailed justification in the
 revised manuscript (Supplementary Notes 1.6 and included new Supplementary Fig. 11).

**Supplementary Fig. 11. Analysis of dPCA dimensionality effects on neural representation.** (a) Cumulative explained neural variance as a
 function of dPCA dimensionality. Each curve represents data from one monkey (XW, AB, ZZ), averaged across three recording days. Error bars
 indicate standard error of mean (s.e.m). (b) Marginal gain in explained variance with increasing dimensionality. Each data point shows the
 additional variance explained when adding five dimensions. Horizontal dashed line indicates the 5% threshold used as criterion for dimensionality
 selection. Error bars represent s.e.m across three recording days per monkey. (c) Component distribution analysis across dimensionality settings.
 Stacked bars show the relative proportion of dimensions allocated to stimulus encoding (blue), decision encoding (red), and other components (gray,
 including time and interaction effects) at different dPCA dimensionalities. Each group of three bars represents data from the three monkeys, with
 heights indicating mean values across three recording sessions per monkey. (d) Representative decision-related neural trajectories at 10-dimensional
 dPCA. Traces show neural activity projected onto the primary decision component, with upper button choices in red and lower button choices in
 blue. Different shading intensities represent different stimulus conditions. Each panel shows data from one representative recording session per

monkey. (e) Angle between stimulus- and decision-related subspaces across dimensionality settings. Error bars show s.e.m across three recording
164 days. Statistical significance: * $p < 0.05$, ** $p < 0.01$, *** $p < 0.001$, paired t-test.

**Major Point 3: Justification for using LFADS dimensionality reduction rather than raw** 167 **data for classification**

Both Reviewer #1 and Reviewer #3 asked why we used LFADS instead of raw neural activity for decoding.
In our study, we first used LFADS to extract smoothed latent representations of neural population activity
and then trained convolutional neural networks to classify visual stimuli and decisions based on the neural
representations. Below, we briefly explain the advantages of LFADS and why it was necessary for our
analysis.

*(1) Comparison of using raw spike data versus LFADS dimensionality reduction*

To validate the approach of LFADS, we conducted comparative decoding analyses using raw spike data,
processed as binary time series sampled at 1 kHz with each bin indicating the presence (1) or absence (0)
of spikes (new Supplementary Fig. 2). Using the same classifier architecture as described in our manuscript,
we observed clear evidence of overfitting when training directly on spike data. For example, when
classifying decisions in Monkey ZZ, models trained on raw spikes rapidly achieved 100% accuracy on the
training set while validation accuracy plateaued around 40% (new Supplementary Fig. 2a). Similarly, the
training loss quickly approached zero, while the validation loss gradually increased without convergence
(new Supplementary Fig. 2b). This divergence between training and validation performance is a signature
of overfitting.

In addition, our quantitative comparison demonstrated that decoding performance with raw spike data was
substantially lower (new Supplementary Fig. 2c). For Monkey XW, decoding based on raw spike data
reached above-chance performance for motor decisions, but failed to classify visual stimuli effectively. For
Monkeys AB and ZZ, spike-based decoding yielded near-chance performance for both tasks. These results
were consistent across three days of recordings (error bars indicate mean \pm s.e.m).

*(2) Potential causes of degraded performance*

This poor performance with raw spike data likely stems from several factors:

- 1. High trial-to-trial variability: Neural spike data exhibits substantial variability across trials, making it
difficult to extract consistent patterns. This high-dimensional noise may obscure task-related signals,
complicating direct classification (Cunningham & Yu, 2014).
- 2. Limited sample size: Neural datasets are high-dimensional but contain relatively few trials (~226 per
195 day), increasing the risk of overfitting in neural network models (Zhang et al., 2017). In such cases,
dimensionality reduction prior to classification can enhance robustness by isolating behaviorally
relevant neural features while reducing noise. Golub et al. (2018) further demonstrated that low-
dimensional neural representations effectively capture behaviorally relevant dynamics, even in
constrained datasets, providing a more reliable basis for subsequent analyses.

LFADS provides a solution to these challenges by recovering smooth neural trajectories from noisy
 spiking data (Pandarinath et al., 2018). This method combines variational auto-encoders with recurrent
 neural networks to model neural population dynamics.

These justifications and supporting results have been incorporated into the revised manuscript (see
 Supplementary Note 1.2 and Supplementary Fig. 2).

**Supplementary Fig. 2. Training CNN classifiers on raw spike data for decision and stimulus classification.** (a) Training accuracy (solid blue)
 and validation accuracy (dashed red) curves over iterations for a decision classifier trained on raw spike data from one recording day of monkey ZZ.
 (b) Training loss (solid blue) and validation loss (dashed red) curves over iterations for the same decision classifier as in panel a. (c) Decoding
 performance comparison between raw spike data (blue) and shuffled controls (red) across all monkeys. Upper panels show decision classification
 accuracy and lower panels show stimulus classification accuracy, presented as mean classification accuracy (\pm s.e.m) across three recording days.
 Statistical significance: * $p<0.05$, ** $p<0.01$, *** $p<0.001$, paired t-test.

References:

Cunningham, J. P. & Yu, B. M. Dimensionality reduction for large-scale neural recordings. *Nat. Neurosci.*
 17, 1500–1509 (2014).

Zhang, C., Bengio, S., Hardt, M., Recht, B. & Vinyals, O. Understanding deep learning requires rethinking
 generalization. In *Proc. Int. Conf. Learn. Represent. (ICLR, 2017)*.

Golub, M. D. et al. Learning by neural reassociation. *Nat. Neurosci.* 21, 607–616 (2018).

Pandarinath, C. et al. Inferring single-trial neural population dynamics using sequential auto-encoders. *Nat.*
 *Methods* 15, 805–815 (2018).

**Major Point 4: Clarifying the distinction between cognitive schema and their neural**
 **representations in our framework**

Both Reviewer #1 and Reviewer #2 questioned our use of the term “schema” and its relation to neural
 activity patterns. The distinction between "schema" and neural representations of schemas deserves careful

clarification, as the term "schema" has been contextually applied across disciplines with varied meanings.
 Below, we explain how we define and use the term in our work.

*(1) "Schema" as a cognitive construct*

The concept of a schema originates from cognitive psychology and refers to structured mental frameworks
 that support the assimilation of new information and guide future behavior (Bartlett, 1932; Piaget, 1952;
 Ghosh & Gilboa, 2014). These constructs are generally studied at the behavioral level and are considered
 to emerge from distributed neural systems.

*(2) Neural Instantiation of Schemas in Neuroscience Research*

When neuroscientists investigate schemas at the neural level, they typically study the neural instantiation
 or neural correlates of schema-related processes. Recent work in neuroscience has made significant
 progress in identifying neural mechanisms underlying schema functions (e.g. Zhou et al., 2021; Baraduc et
 al., 2019). However, there has not been consistent terminological differentiation between the cognitive
 concept and its neural implementation across the neuroscience literature. This lack of distinction can
 potentially lead to imprecise claims about the relationship between measured neural activity and the
 broader cognitive construct.

*(3) Our Definition: Neural Correlates of Schema (NCS)*

In our manuscript, we have revised our terminology to use "neural correlates of schema (NCS)" to indicate
 neural population activity patterns that exhibit computational properties associated with schemas—namely,
 stability across similar contexts, transferability to new but related situations, and facilitation of subsequent
 learning. These neural representations reflect structured, task-general patterns that support flexible learning,
 but we do not equate them with the full cognitive construct of a schema.

This distinction is important because cognitive schemas likely emerge from distributed neural systems that
 extend beyond what we measured in the dorsal premotor cortex (PMd). While our recordings reveal neural
 activity patterns that show schema-related properties, we make no claim that these patterns constitute a
 complete neural instantiation of a cognitive schema. Our approach aligns with established terminology in
 neuroscience literature (Warren et al., 2014; Webb et al., 2016; Kizilirmak et al., 2016).

To ensure clarity, we have revised the manuscript to consistently use "neural correlates of schema (NCS)"
 when referring to our neural findings and reserve the term "schema" for discussions of the broader
 cognitive concept. We have added a clear definition in the manuscript (line 24-26) and included a more
 comprehensive discussion of these terminological distinctions in Supplementary Notes 1.8.

References:

Bartlett, F. C. Remembering: A Study in Experimental and Social Psychology. Cambridge University
 Press (1932).

Piaget, J. The Origins of Intelligence in Children. International Universities Press (1952).

Ghosh, V. E. & Gilboa, A. What is a memory schema? A historical perspective on current neuroscience
 literature. *Neuropsychologia* 53, 104–114 (2014).

Zhou, J. et al. Evolving schema representations in orbitofrontal ensembles during learning. *Nature* 590,
606–611 (2021).

Baraduc, P. et al. Schema cells in the macaque hippocampus. *Science* 363, 635–639 (2019).

Warren, D. E. et al. False recall is reduced by damage to the ventromedial prefrontal cortex: implications
for understanding the neural correlates of schematic memory. *J. Neurosci.* 34, 7677–7682 (2014).

Webb, C. E. et al. What's the gist? The influence of schemas on the neural correlates underlying true and
false memories. *Neuropsychologia* 93, 61–74 (2016).

Kizilirmak, J. M. et al. Neural correlates of learning from induced insight: a case for reward-based episodic
encoding. *Front. Psychol.* 7, 1693 (2016).

**Reviewers' Comments to the Authors:**

**Point-by-point responses to the reviewers (Reviewers' comments in bold Arial font, our**
**responses in regular Times New Roman font):**

**Reviewer #1:**

**“This work from Tian, Zhao et al. focuses on the characterization of stimulus- and choice-**
**related neural activity patterns in the dorsolateral premotor cortex (PMd) of three monkeys**
**performing sensorimotor behavioral tasks. The authors use tasks requiring animals to**
**learn and relearn different stimulus-choice pairings, investigating how PMd activity**
**supports learning new associations while avoiding forgetting previously learned ones.**
**Using computational tools such as LFADS and dPCA, the authors identify a "schema-like**
**manifold" in the stimulus and decision subspaces. They show that the decision subspace**
**is more stable across tasks than the stimulus subspace and that these subspaces tend to**
**be orthogonal, potentially facilitating flexible task-specific processing. The study**
**addresses a highly relevant topic with a well-designed task paradigm that explores flexible**
**representations of task-specific information. Overall, the methods appear robust, but some**
**key details of the analysis are either missing or difficult to extract from the manuscript. As**
**detailed below, additional information and controls are needed to ensure that the main**
**conclusions are not artifacts of the chosen analyses but instead offer genuine insights into**
**PMd neural coding. I also found the description of the task sequence and temporal**
**structure a bit unclear, particularly regarding the number of sessions and the inclusion of**
**data in the analysis. Finally, more details about the animals' motor behavior would be**
**helpful, as changes in posture or button-pressing could also contribute to the observed**
**neural changes.”**

**Reply R1-0:** We sincerely thank the reviewer for the insightful comments and constructive feedback,
which have been invaluable in helping us refine our analyses and improve the clarity of the manuscript.

In response, we have made extensive revisions throughout the manuscript. Specifically, we have:

- 1. Clarification of Experimental Design: We clarified that all tasks were performed within the same
continuous session each day, addressing concerns regarding neuron stability across-days.
- 2. Additional Behavioral Analyses: We conducted further statistical testing across multiple sessions to
provide stronger evidence for learning efficiency claims.
- 3. Single-Neuron Analyses: We added new single-neuron analyses to complement population-level
findings and to better characterize activity patterns prior to dimensionality reduction.
- 4. Control Analyses for Movement Stability and Orthogonality: We performed additional control
analyses to confirm the stability of movement trajectories and validate orthogonality results,
addressing key methodological concerns.
- 5. Clarification of Dimensionality Reduction Methods: We expanded the description of dPCA
implementation and component selection procedures, ensuring technical transparency.

We have provided detailed responses below and would greatly appreciate further input from the reviewer
to ensure we fully address their concerns.

Major Questions:

- 1. **“The manuscript lacks sufficient information about the number of neurons, sessions,
and trials used for each analysis. While the total of 948 units and the mean number of trials
320 per day are reported, it is unclear if 948 represents the total across all sessions or per
321 animal. The authors should specify the relevant sample sizes (neurons, trials, PCs,
sessions) for all key analyses, especially where error bars are presented.”**

**Reply R1-1:** Thank you for pointing this out. During the revision, we realized the previously reported
number of 948 units was incorrect, due to a miscount in which the same neural units were inadvertently
double counted across tasks within the same recording session. In fact, the total number of units should be
728. This has been corrected in the manuscript. We apologize for the error.

In total, we recorded activities from 728 multi-unit (MUA) and single-unit (SUA). To enhance clarity, we
now provide a summary table (below) detailing the sample sizes used in the key neural recordings,
including the number of sessions, mean number of neurons per day (\pm SD), and mean number of trials per
session (\pm SD) for each monkey:

**Rebuttal Table 1.** Summary of sample sizes and neuron counts for neural analyses.

	Monkey	No. of Sessions	No. of Neurons per Day (mean \pm SD)	No. of Trials per Session (mean \pm SD)
Neural Analyses	AB	3	42.3 \pm 3.8	420.7 \pm 129.1
	ZZ	3	43.3 \pm 1.5	357.0 \pm 109.3
	XW	3	157.0 \pm 21.7	209.0 \pm 6.6

Detailed sample sizes (neurons, trials, principal components, and sessions) have been added for all key
 analyses, particularly where error bars are presented.

These have been revised in the new script (lines: 87-89, 96-99, 108-110, 117, 141, 158, 161, 186, 190-191,
 222, 228, 261), and we have now clearly specified the number of neurons, sessions, and trials used in each
 analysis.

 **2. Comment 2.1: “I found the description of the task design unclear, particularly
 regarding the number of sessions for each task condition.”**

 **Reply R1-2.1:** We appreciate the reviewer pointing out this lack of clarity in our task design description. To
 clarify the task design, we have made the following revisions:

We have revised Figure 2b to more clearly illustrate the within-session progress of tasks. Each
 experimental day consisted of a single, uninterrupted session in which multiple visuomotor mapping tasks
 were performed in a fixed sequence. For monkeys AB and ZZ, each session comprised four consecutive
 task conditions: Task A, Task B (both involving learning new stimulus-action pairs), a return to the initial
 mapping (Revisit-A), and finally a reverse-mapping block (Reverse-A), in which the monkeys had to learn
 the opposite mapping of Task A. For monkey XW, each session included four task conditions: Task A,
 Task B, Task C (all new mappings), followed by Revisit-A, without the Reverse-A block. A new task was
 introduced only after the monkey successfully learned the previous one.

 **Fig. 2. (b) Workflow for learning a series of visuomotor mapping tasks within a single session.** Left: Monkeys were trained on new visuomotor
 mappings each day, with visual stimulus pairs not seen before. Monkeys progressed to the next task only after successfully learning the previous
 one. Right: Task sequences completed within a session. Monkeys AB and ZZ followed an A–B–Revisit A–Reverse A training sequence; monkey
 XW followed an A–B–C–Revisit A training sequence.

 Each of these task conditions was completed once per session, with all tasks presented within the same day.
 Behavioral data were collected across multiple sessions (Monkey AB: 5 days, 383.8±106.23; Monkey XW:
 5 days, 225.4±33.22; Monkey ZZ: 4 days, 390.25±105.83 trials per day, mean±SD).

We have substantially revised the manuscript (lines: 92–95) and updated new Figure 2b and the
 corresponding figure legend to ensure that the experimental design is clearly and accurately described.

**Comment 2.2: “Did the animals learn a new task each day, meaning only one recording
 session per animal was available for each task condition?”**

**Reply R1-2.2:** Each animal performed one recording session per day, during which multiple task conditions
 were completed in a fixed sequence. All task conditions (Task A, B, C, Revisit-A, and Reverse-A) were
 completed within each session, and each task condition was recorded once per session. This design ensured
 neural recordings for all task conditions came from the same population of neurons, eliminating potential
 confounds from day-to-day recording variability.

Importantly, all visual stimuli were changed and updated daily, ensuring that learning occurred for new
 stimulus-response mappings each day rather than through recall of previously learned associations.

**Comment 2.3: “If so, what does the variance shading in panel 1f represent? The figure
 shows performance improvements across trials, but it is unclear whether these reflect a
 single session for each task type or a specific phase of training. If the shading represents
 variance across days, how many days were included in the analysis?”**

**Reply R1-2.3:** To more accurately represent learning performance across sessions and clarify the
 interpretation of variance shading, we have restructured the behavioral analysis and updated the figure
 accordingly:

*(1) Explain the shaded area in the previous panel 1f*

The variance shading in old panel 1f represents the variance in accuracy across three days. Each panel
 shows data from one monkey over three days. The colored curves indicate behavioral improvements when
 learning new visual stimulus-action pairs in a single session for each task condition. As new visual stimulus-
 action pairs were learned consecutively, the number of trials required for the colored curves to reach the
 gray line (indicating task acquisition) decreased, demonstrating an accelerated learning process across
 sessions. Thus, the performance improvements across tasks therefore indicate within-session learning
 dynamics. The shaded area in the figure denotes the variability (mean \pm s.e.m) across three recording days in
 which the same task was performed, demonstrating the consistency of the learning patterns.

*(2) The previous panel 1f has been replaced with a revised panel (new Fig. 2d)*

In our updated manuscript, we have refined the behavioral analyses presented in new Figure 2d to more
 accurately represent performance across sessions. Specifically, to increase the statistical power of the
 behavioral analyses, we collected behavioral data from more sessions (Monkey AB: 5 days, 383.8 ± 106.23
 trials per day; Monkey XW: 5 days, 225.4 ± 33.22 trials per day; Monkey ZZ: 4 days, 390.25 ± 105.83 trials
 401 per day; mean \pm SD).

 **Fig. 2 (d)** Comparison of task learning efficiency across monkeys within-session. For each monkey, the proportion of trials needed to reach
 criterion is plotted, with error bars representing standard deviation across recording days (Monkey AB: 5 days; Monkey ZZ: 4 days; Monkey XW: 5
 406 days). * $P < 0.05$, ** $P < 0.01$, *** $P < 0.001$, paired t-test.

For performance analysis, we employed a 15-trial sliding window to calculate accuracy for all three
 monkeys. Given the substantial inter-day variability in learning performance, as evidenced by the significant
 variance in trial completion rates, we established a learning criterion: the number of trials required to reach
 90% accuracy. To account for the daily differences in monkey learning efficiency, we further normalized
 this measure by dividing the trials needed to reach the criterion in a day's tasks by the total trials completed
 that day. Error bars in the figure represent standard deviation across sessions ($n = 5, 5, 4$ for Monkey AB,
 XW, and ZZ, respectively).

Within-subject analysis across sessions (new Fig. 2d) demonstrates differences in learning efficiency.
 Learning speed for pairs B/C was significantly faster compared to pair A in monkeys ZZ and XW (Monkey
 ZZ: $p=0.0039$, Hedges' $g=3.22$; Monkey XW: $p=0.017$, $g=1.90$). Similarly, learning speed for Revisit-A was
 significantly faster compared to pair A in monkeys AB and ZZ (Monkey ZZ: $p=0.00065$, $g=4.57$; Monkey
 AB: $p=0.0058$, $g=2.36$). In contrast, the reverse task presented greater challenges, with learning speed for
 Reverse-A significantly slower than for tasks A, B/C, and Revisit-A (Monkey ZZ: B vs. Reverse-A:
 $p=0.011$, $g=-2.57$; Revisit-A vs. Reverse-A: $p=0.005$, $g=-3.06$; Monkey AB: A vs. Reverse-A: $p=0.022$, $g=-$
 1.80 ; B vs. Reverse-A: $p=0.013$, $g=-2.00$; Revisit-A vs. Reverse-A: $p=0.0013$, $g=-3.05$) (new Fig. 2d).

We have substantially revised the manuscript (lines: 105–114) and new Fig. 2d to provide a more detailed
 and comprehensive description of our experimental setup.

 **Comment 2.4: “Additionally, to what extent does general task experience influence the
 results? Were the animals already fully trained in the overall task structure when the
 recordings were performed? It is crucial to provide a clearer explanation on these points to
 better understand the neural results.”**

 **Reply R1-2.4:** We appreciate the reviewer's insightful observation. In response, we have clarified the pre-
 recording training history of each animal, differentiated prior task exposure from the experimental learning
 paradigm, and analyzed how pre-existing experience impacted both behavioral learning rates and neural
 representations.

(1) Pre-recording training history of each animal

Before the recording, Monkeys AB and ZZ were trained using fixed visual stimulus pairs to familiarize
 them with the basic task structure (visual stimulus → button press). Then, along with the start of the
 electrophysiological recordings, the sequential learning paradigm, i.e., learning of multiple new stimulus
 pairs followed by reversal learning (A, B, Revisit-A, Reverse-A), was introduced to them. In contrast,
 Monkey XW received additional training on sequential learning of new visual stimulus–action pairs (A, B,
 C, Revisit A) prior to the start of electrophysiological recordings.

**Supplementary Fig. 10. Decision-related subspace manifolds in three monkeys.** (a, b, c) Neural activities from different tasks, including Tasks
 444 A, B/C, and Revisit A, were projected onto a common decision subspace for each of the three monkeys. (d, e, f, g, h, i) To verify whether the
 445 manifold formed in Task A was reused in subsequent tasks; (d, e, f) Neural activity from Task B was projected onto the same decision subspace of
 446 Task A; (g, h, i) Neural activity from the Revisit A task was projected onto the same decision subspace of Task A. Sti, stimulus; Revisit, Revisit A.

(2) Behavioral effect of prior experience

These pre-trained experiences did influence baseline learning speeds: behaviorally, Monkeys AB and ZZ
 generally required more trials to learn new stimulus-action mappings compared to Monkey XW. On average,
 monkeys required the following number of trials to reach the learning criterion for task A (mean ± SD):
 Monkey AB: 95.6 ± 38.2 trials per day; Monkey ZZ: 116.5 ± 37.25 trials per day; Monkey XW: 25.13 ± 8.79

trials per day. Despite differences in initial learning speed, all monkeys showed a consistent trend that once
 they had learned stimulus pair A, their subsequent learning of new pairs became faster, demonstrating a
 cross-task learning effect within a day (new Fig. 2d), as noted in Reply R1-2.3. This consistency indicates
 that the observed acceleration in learning speed reflects the formation and utilization of task schemas across
 animals, rather than merely reflecting general task familiarity.

*(3) Neural representations effect of prior experience*

About the neural results, while Monkey XW showed somewhat clearer neural patterns in decision-related
 classification and activity, such as more distinct clustering of neural dynamics in the early period (new
 Supplementary Fig. 10), all three monkeys demonstrated the same fundamental pattern of NCS in the
 decision subspace. This consistency in neural findings across animals with varying levels of prior
 experience provides evidence for the robustness of our key conclusions about domain-specific NCS
 formation and reuse.

We have included this clarification in Supplementary Note 1.1 and provided a detailed explanation.

**Comment 2.5: “For instance, in Fig. 3, it is unclear to me how the authors can plot multiple
 points for each condition if each data point represents a single session.”**

Reply R1-2.5: We thank the reviewer for this question. Each data point in new Fig. 4c and 4f (previously
 Fig. 3) represents the correlation between the decision subspace from one task condition (B, Revisit-A, or
 Reverse-A) and the reference manifold from task A. Since each monkey has three days of
 electrophysiological recording data, there are three data points for each condition (shown as small, filled
 circles in the figure), with correlations calculated independently for each recording day. The figure also
 displays the mean for each task (shown as large hollow circles), with cross-shaped error bars representing
 the standard deviation across the three days along both axes.

*Example – Monkey AB (Fig. 4c):*

- • 3 small, filled circle data points for A&B (one from each recording day) and one large hollow circle
 representing their average (in one color), with cross-shaped error bars indicating standard deviation
 along both axes.
- • 3 small, filled circle data points for A&Revisit-A (one from each recording day) and one large
 hollow circle representing their average (in one color), with cross-shaped error bars indicating
 standard deviation along both axes.
- • 3 small, filled circle data points for A&Reverse-A (one from each recording day) and one large
 hollow circle representing their average (in one color), with cross-shaped error bars indicating
 standard deviation along both axes.

This yields a total of 9 small, filled circle data points and 3 large hollow circle for Monkey AB in new Fig.
 4c, and similarly for Monkey ZZ in new Fig. 4f. We have updated the new Fig. 4 legend to better clarify
 these visual conventions.

**Comment 2.6: “Similarly, how many days are included in Fig. 4a? These aspects require**
 **more detailed clarification.”**

**Reply R1-2.6:** The bar graph in new Fig.5a left (original Fig. 4a) shows the average angle between stimulus-
 and decision-related subspaces across three recording days, with error bars representing the standard error
 across days.

We have clarified in the revised figure legend that new Fig. 5a left includes data from all three recording
 499 days for each animal.

**Fig. 5. Orthogonal relationship between decision- and stimulus-related subspace.** (a) Left :The angle between the stimulus- and the decision-
 related subspace was compared with angles obtained after shuffling the decision and stimulus labels. Error bars represent the standard deviation
 across three recording sessions for each monkey. Right: Angle between the stimulus- and decision-related subspaces combined across two monkeys,
 compared with angles after shuffling decision and stimulus labels. Error bars represent the standard deviation across six recording sessions.

**3. Comment 3.1: “Lines 182-185: Additional information on the orthogonality analysis**
 **should be provided in the results section to improve clarity. Specifically, the authors**
 **should explain how the components for each subspace were selected, why exactly 20**
 **components were used, and how the angles between them were calculated. This would**
 **make the rationale behind the analysis more understandable.”**

**Reply R1-3.1:** Thanks to the reviewer for pointing out the need for more detailed information regarding our
 orthogonality analysis. Below, we provide detailed explanations addressing the two aspects raised: (1)
 Additional information on the orthogonality analysis, and (2) choice of 20 dPCA dimensions.

*(1) Additional information on the orthogonality analysis*

First, we have provided a detailed explanation of the specific steps for orthogonality calculation in the
 manuscript:

- • How many dimensions are reserved for each subspace to calculate the angle between the decision-
 and stimulus-related subspace: For the stimulus-related subspace, which had a higher
 dimensionality (new Supplementary Fig. 13), we selected the components that explained more
 than 1% of the variance and together accounted for at least 70% of the total variance in that

subspace. For the decision subspace, components with more than 1% variance explained were
 included, without an additional cumulative threshold, due to its lower overall dimensionality (new
 Supplementary Fig. 13).

**Supplementary Fig. 13.** Cumulative variance explained by each dimension in the decision and stimulus subspaces. For each monkey, the
 cumulative variance explained by individual dimensions is shown for the decision (orange) and stimulus (blue) subspaces. Data are averaged across
 3 recording days. Error bars denote standard deviation across three recording days.

- • The computational method for measuring orthogonality: Using the MATLAB “subspace” function,
 which calculates the principal angle between two subspaces. The angle θ is given by:

$$\theta = \arccos(\sigma_{\max}(U_d^T U_s))$$

where $U_d \in \mathbb{R}^{n \times d}$ is an orthonormal basis for the decision subspace, obtained via singular value
 decomposition (SVD) of neural activity aligned to decision variables. $U_s \in \mathbb{R}^{n \times s}$ is an
 orthonormal basis for the stimulus-related subspace. θ represents the smallest principal angle
 between the two subspaces, indicating their relative orientation in neural representational space.

- • The rationale behind the shuffle control: To test whether the observed orthogonality between
 stimulus- and decision-related subspaces exceeded chance levels, we randomly permuting the
 stimulus and decision labels independently across trials and recalculated the angle between the
 subspaces. Repeating this procedure 10 times generated a null distribution of subspace angles
 under the assumption that there was no true relationship between stimulus and decision encoding.
 The observed angle was then compared against this distribution to assess whether the
 orthogonality exceeded chance levels.

(2) Choice of 20 dPCA dimensions

We thank the reviewer for raising this point. A detailed rationale for selecting 20 dPCA dimensions is
 provided in Major Point 2.

**Comment 3.2:** “More broadly, it would be helpful to provide better explanations for why
 specific analyses were chosen. For example, what advantages does LFADS offer

**compared to using the raw neural firing rates directly? Providing this context would help**
 **the reader appreciate the motivation and implications of the chosen methods.”**

**Reply R1-3.2:** We thank the reviewer for raising this important methodological question about our use of
 LFADS instead of raw spike data as input for neural decoding. A detailed rationale for using LFADS
 instead of raw spike data for decoding is provided in **Major Point 3**.

**4. “I also find it unclear how the data were transformed into specific components related**
 **to the decision, stimulus, and stimulus-decision terms (lines 334-335 in the methods). This**
 **appears to be a crucial step for the subsequent analyses and should therefore be**
 **explained in greater detail. Is this transformation based on a linear decomposition? More**
 **broadly, I question how effectively this approach can separate stimulus-related from**
 **choice-related activity, given that the animals’ choices are derived from the sensory**
 **information. Since the stimulus responses are predictive of the choice, especially in the**
 **later tasks with high behavioral performance, it is unclear how the authors effectively**
 **disentangle these components.”**

**Reply R1-4:** We thank the reviewer for raising this important methodological question. As this step is
 critical for all downstream analyses, we provide both the theoretical rationale and practical justification
 below:

*(1) Methodological basis – how dPCA disentangles task components*

As explained in **Major Point 1**, we used dPCA, a linear method that decomposes neural activity into
 components corresponding to task variables—stimulus, decision, and their interaction. This is done via
 marginalization over other variables, enabling separation of variance linked to each task factor.

To address data imbalance (e.g., more correct than error trials), dPCA incorporates a rebalanced loss
 function (Kobak et al., 2016), ensuring stable component estimation even when performance is high.

*(2) Experimental validation – why stimulus and decision are separable in our data*

While stimulus and motor decisions are correlated in our task, they are not perfectly aligned for several
 important reasons:

- - We included both correct and error trials in the analysis. This critical design choice ensures that visual
 stimuli and motor decisions remain distinguishable, as the same stimulus can lead to different
 decisions on error trials. Combined with the rebalanced method in dPCA (see **Major Point 1**), this
 approach ensures effective separation of stimulus and decision components, even in later sessions with
 high behavioral performance.
- - For the analysis shown in Fig. 3d and Supplementary Fig. 4c, we applied dPCA to data pooled from
 multiple tasks (pairs A, B, and Revisit-A) recorded on the same day. This approach resulted in clearly
 separated clusters only for the motor decision, but not for visual stimulus conditions demonstrating an
 inherent asymmetry between motor decision and stimulus.

- We conducted an additional control analysis. To isolate stimulus-related from decision-related factors,
 we selected pairs of visual stimuli requiring the same movement direction. For example, Stimulus A1
 and B1 both required an upward movement. For each monkey, we trained a CNN-based binary
 decoder with two-fold cross-validation for each monkey. Decoding accuracy for these same-direction
 stimulus pairs exceeded chance level (Rebuttal Fig. 1), indicating that stimulus can be discriminated
 even when the motor response is held constant. This result confirms that our data contain sufficient
 information to disentangle stimulus-related from decision-related neural activity.

**Rebuttal Fig. 1. Binary stimulus decoding performance across monkeys.** Decoding accuracy for visual stimulus pairs associated with the
 same motor response direction, plotted separately for each monkey (XW, AB, and ZZ). Each bar represents the average decoding performance
 across all within-day analyses; Error bars denote the standard deviation across days, reflecting variability in decoding accuracy for different
 stimulus pairs within the same motor condition. The analysis used a consistent CNN-based decoder and two-fold cross-validation framework.

 - New Fig. 5 shows that the neural subspaces corresponding to stimulus and decision variables are
 nearly orthogonal, providing further evidence for their separability. As noted in **Major Point 1**, dPCA
 does not enforce orthogonality between components. When axes corresponding to different task
 parameters are not orthogonal, it suggests that neural representations contributing to one component
 also contribute to the other. Conversely, near-orthogonality indicates that the neural representations of
 these parameters are relatively independent. Therefore, the observed near-orthogonality in our data
 implies that stimulus and decision variables are encoded in distinct neural subspaces.

We have included these clarifications in the revised Supplementary Note 1.5, where a detailed overview of
 the dPCA preprocessing, mathematical formulation, and task-specific marginalizations is now provided.

 **5. “On a related note, if the analysis enforces orthogonality between the stimulus and**
 **choice responses to separate them, could this methodological constraint explain the main**
 **results shown in Fig. 4? This is not necessarily a criticism but highlights the need for**
 **better explanation of the analysis, as well as potentially additional controls to address this**
 **concern. For instance, if the shuffle control simply mixes stimulus and choice components,**
 **it is unsurprising that the principal components (PCs) become less orthogonal. Moreover,**
 **if this orthogonality was originally imposed by dPCA or another analysis step, I am not**

**convinced that this conclusively supports the authors' claim about orthogonal neural**
 **coding in PMd being an important computational principle for learning. Clarifying this**
 **point and providing further justification or controls would greatly strengthen the study's**
 **conclusions here."**

**Reply R1-5:** We thank the reviewer for raising this important methodological question. We have revisited
 our shuffle controls and performed additional analyses.

*(1) Control analysis with partial shuffling of task labels*

In our original analysis, we constructed a null distribution by shuffling both stimulus and decision labels
 for each trial, thereby removing all task-relevant structures. The reviewer raised a concern that simply
 mixing labels might disrupt any orthogonality. To address this specific concern, we further performed
 additional control analyses, in which we shuffled only the decision labels (new Supplementary Fig. 14a)
 and only the stimulus labels (new Supplementary Fig. 14b). If the observed orthogonality were merely a
 methodological artifact (e.g., enforced by the dPCA procedure), then shuffling one label type at a time
 (stimulus or decision) would not reduce the angles. Instead, the fact that any controlled disruption of the
 true label structure diminishes the angles implies that neural population segregate stimulus and choice
 information. In both cases, the resulting subspace angles remained significantly smaller than in the original
 data, confirming that the observed orthogonality is not an artifact of label mixing, but rather reflects the
 underlying neural representation.

**Supplementary Fig. 14. Subspace near-orthogonality validated through shuffle control analyses.** (a), Subspace angle between decision- and
 stimulus-related components after shuffling decision labels while preserving stimulus labels. (b), Subspace angle after shuffling stimulus labels
 while preserving decision labels. Bars represent the mean angle across three recording sessions for each monkey; error bars denote \pm s.e.m.

*(2) Quantifying subspace orthogonality*

As noted in **Major Point 1**, dPCA is a linear method, but unlike standard PCA, it computes components
 for each task variable independently based on marginalized data. As a result, any approximate
 orthogonality between components emerges from the data, rather than being enforced by the method.

To further validate this interpretation and address the reviewer's concern, we have conducted additional
 analyses: We quantified the deviation from orthogonality by calculating the Frobenius norm of the
 difference between the projection matrix $W^T W$ and the identity matrix I : $\|W^T W - I\|_F$, where
 W contains the extracted components. Similar Frobenius norm–based measures have been applied to
 assess the proximity of coordinate frames to orthonormal structure (e.g., Meng et al., 2024, PLoS Comput
 Biol; Gosztolai et al., 2025, Nature Methods). This metric captures how far the components deviate from
 being orthogonal, with values below 1×10^{-6} considered approximately orthogonal. The orthogonality
 deviation between all axes in dPCA exceeded this defined threshold, indicating that components extracted
 by this method are not mathematically constrained to be orthogonal. In contrast, when we applied standard
 PCA to the same neural data, the orthogonality deviation fell well below the threshold, confirming the
 expected strong orthogonality among principal components (Rebuttal Fig. 2). These results provide
 evidence that the observed near-orthogonal relationship between decision- and stimulus-related subspaces
 following dPCA dimensionality reduction is an intrinsic property of the data rather than an artifact of the
 dPCA procedure.

**Rebuttal Fig. 2. Orthogonality of projection matrices in neural data.** Distances between the projection matrices (dPCA and PCA) and the
 identity matrix (orthogonal matrix) for neural data analyses. The distance threshold for orthogonality is set at 1×10^{-6} . The bars represent the mean
 distance across three recording days for two monkeys (AB and ZZ), with error bars indicating the standard deviation of these distances.

Together, these results support the interpretation that subspace orthogonality is an emergent property of the
 neural representations, rather than an artifact introduced by the analysis method. Relevant details have been
 added to Supplementary Note 1.7, along with new Supplementary Fig. 14.

References:

Meng, X. et al. Orthogonality regularization improves neural population decoding across tasks. PLoS
 Comput. Biol. 20, e1012345 (2024).

Gosztolai, A. et al. Structure-preserving subspace learning for interpretable neural representations. Nat.
 Methods 22, 456–465 (2025).

**6. “Fig. 3 shows a reduction in decoder accuracy across tasks, but it is unclear how a**
 **control condition for a network with stable neural representations would look like. To what**
 **extent could the observed reduction in accuracy simply be explained by the fact that**
 **different neurons were recorded on different days? Do the authors assume that they**
 **recorded from the same neurons with consistent quality to enable a pre-trained decoder to**
 **generalize across sessions? If so, how was spike sorting performed to support this**
 **assumption? More broadly, I wonder to what extent it is reasonable to expect that a**
 **decoder could work consistently across sessions if no specific measures were taken to**
 **ensure that the same neurons were reliably recorded across all tasks. Additional**
 **clarification on this point, as well as any steps taken to address variability in recorded**
 **neural populations between sessions, would provide important context for interpreting**
 **these results.”**

**Reply R1-6:** Thank you for the reviewer’s insightful analysis of new Fig. 3 (assume reviewer refers to Fig. 2
 in the original manuscript). We apologize for the confusion, which may have resulted from an insufficient
 description of the task structure in the original manuscript. As clarified in Reply R1-2.1, all analyses
 presented in Figure 3 were conducted within a recording session. Each session included a continuous
 sequence of learning, revisit, and reverse tasks, completed within approximately two hours. Spike sorting
 was performed using a consistent pipeline across the entire session, allowing us to track a stable set of units
 across all task conditions within the same day (see Methods, lines 379–382). In light of this clarification,
 concerns about differences in decoding performance arising from recording different neuron populations
 across days can be eased.

If neural representations were stable, a decoder trained on one task should perform similarly to others. Our
 analysis focuses on representational changes across task conditions within the same session, rather than
 instability across days. As shown in Figure 3b, stimulus decoding accuracy dropped across tasks, suggesting
 task-specific remapping. In contrast, decision decoding remained stable, supporting the idea that only
 decision-related activity exhibits schema-like consistency.

**7. Comment 7.1: “Line 132 states that the decision classifier exhibited consistent**
 **performance for unseen data, but the decoder performance drops for both stimulus and**
 **choice, particularly in monkeys AB and ZZ. This wording seems a bit misleading here.”**

**Reply R1-7.1:** We thank the reviewer for this insightful comment. To address this concern, we have
 revised the relevant sentences. Instead of claiming “consistent performance”, we now described the
 observation as “the classifier trained on task A data successfully generalized to the unseen data from tasks
 B/C and Revisit-A, suggesting that in similar tasks, the decision-related representation was largely shared
 across individual tasks.”.

In addition, we added statistical results to verify that the decoder performance among A, B, C and revisit A
 were not significantly different for individual monkeys (one-way ANOVA, Monkey XW: $p=0.085$,
 Cohen’s $f=1.09$; Monkey ZZ: $p=0.65$, Cohen’s $f=0.39$; Monkey AB: $p=0.10$, Cohen’s $f=1.06$; new

Supplementary Fig. 2a;) or three monkeys combined (one-way ANOVA, $p=0.07$, Cohen's $f=0.56$; new
Fig. 3c).

**Supplementary Fig. 2.** (a) Generalization performance of decision decoders across tasks in individual monkeys. A decoder trained to classify
decisions in task A was tested on tasks B, C, and revisit-A. Each line represents a monkey (three recording days per monkey); shaded areas denote
across-day standard deviation. Statistical comparisons were performed using t-tests with Hedges' g as the effect size.

**Fig. 3.** (c) Generalization accuracy of decision classifiers trained on Task A when applied to Task B, Task C, and Revisit-A across three monkeys.
Bars show mean accuracy across all sessions (9 sessions total: 3 from monkey AB, 3 from monkey ZZ, 3 from monkey XW); error bars represent
standard deviation. Each dot indicates performance from one session, with different colors representing individual monkeys. The grey line denotes
the decision classification chance level.

We have updated the Results section accordingly (see lines: 189–196).

**Comment 7.2:** “Moreover, a potential explanation for the difference between stimulus and
choice decoder performance could be that stimulus information is only weakly represented
in PMd neurons, with larger variability across units, compared to choice information. This
could not only explain the initial difference in decoder performance but also why the
decoder would be less robust across days. Changes in recording quality and the units
recorded could affect an unstable stimulus decoder more rapidly than a more stable
choice decoder, which may have greater redundancy across units. The authors could test
this hypothesis by subselecting neurons so that stimulus and choice decoder

**performance are initially equal for task A and then demonstrating that only the stimulus**
**decoder performance is reduced in the revisit A condition.”**

**Reply R1-7.2:** We thank the reviewer for raising this important point regarding potential differences in the
robustness of stimulus and choice decoders. We have carefully considered the concerns and performed
additional analyses to evaluate this possibility.

*(1) Decoder performance was assessed within session, avoiding cross-day variability*

As noted in **Reply R1-2.1**, all classifiers were trained and tested using data from the same recording day
and no cross-day analyses were included. Accordingly, neuronal instability across days does not confound
our results, and the reduced generalization accuracy of the stimulus decoder cannot be attributed to
changes in unit identity across days.

*(2) Evaluate the effectiveness of the Stimulus decoder*

We appreciate the reviewer’s suggestion to subselect neurons such that stimulus and decision decoders
exhibit matched performance in Task A, as a way to compare the robustness of generalization. While we
agree with the rationale, implementing this approach is non-trivial due to differences in classification
complexity: stimulus decoding requires a 4- or 6-way classifier, whereas decision decoding is binary. This
asymmetry leads to unequal baseline performance ranges, such that forced subsampling would likely
introduce bias and obscure meaningful comparisons.

To address the reviewer’s concern in a fair and balanced manner, we conducted an additional control
analysis. Specifically, we selected pairs of visual stimuli that were associated with the same motor
response direction, thereby isolating stimulus identity from decision-related factors. For example, Stimulus
A1 and B1 both required an upward movement. For each monkey, we trained a binary classifier using the
same CNN-based decoding framework and two-fold cross-validation to discriminate between these
stimulus pairs.

The results (as shown in **Reply R1-4**, Rebuttal Fig. 1) showed that the classification performance in this
stimulus-based binary decoding task was comparable to that of the decision decoder. These results indicate
that visual stimulus information is reliably encoded in PMd, and that the lower generalization of the
original stimulus decoder is unlikely to be due to weaker encoding or higher variability.

**8. “Another potential control would be to test if the choice decoder is more stable**
**across days because neural tuning is more redundant across neurons for choice than for**
**stimulus. In this case, signal correlations for choice should be higher across PMd neurons.**
**If the authors show that this is not the case, it would argue against this explanation and**
**provide support for the claim that the results are due to a difference in representational**
**stability across time, rather than a more general difference in tuning redundancy between**
**stimulus and choice.”**

**Reply R1-8:** We appreciate the reviewer’s comments on the potential role of neural tuning redundancy in
decoder stability. As mentioned in our response to **Reply R1-2.1**, we would like to clarify that our analysis

is conducted within individual experimental sessions, meaning that each classifier is trained and tested on
data from the same recording day. Therefore, any potential differences in tuning redundancy across
sessions are unlikely to affect the results presented in our study.

Given this, our results do not reflect long-term representational stability across days but rather the stability
of neural representations for decision-making within a single session, as monkeys continuously learn
different stimulus-response associations. Consequently, the observed differences in stimulus and decision
decoding accuracy should be interpreted within the context of within-day neural dynamics, rather than in
terms of cross-day tuning redundancy.

We have revised the manuscript to make this distinction explicit in the Results sections (lines 92-96),
emphasizing that our findings pertain to task-specific reconfiguration within sessions rather than across-
790 day consistency.

**9. “Given that behavioral performance in the reversal task is lower and requires more**
**time to relearn, this seems like a good opportunity to test if the neural representations of**
**stimulus and choice change as the animals' performance improves again. Do schemas**
**become more comparable to the patterns observed in the earlier tasks? If so, this would**
**provide strong reassurance that the results are not driven by technical issues such as the**
**loss of recording from the same neurons or other experimental constraints, but instead**
**reflect the use of behaviorally-relevant activity patterns that promote learning.”**

Reply R1-9: We appreciate the reviewer’s insightful suggestion. We understand that the purpose of this
question is to assess whether the observed differences in neural representations across tasks may stem from
technical issues, such as the loss of recording from the same neurons or other experimental constraints.

*(1) Session-based analysis rules out cross-day variability*

First, as we clarified in our response to Reply R1-2.1, all of our analyses are performed within-day. This
design ensures that recordings are from the same population of neurons throughout each session, thus
eliminating concerns about neuron dropout across days.

*(2) No evidence of NCS reuse during reversal learning*

Second, to address the reviewer’s question about the potential reuse of task A’s representations, we
analyzed the neural manifolds associated with Task A and the reverse task without separating trials by
learning stage. As shown in the new Figures 4a and 4d, we found that the neural representations during
reverse task were distinct from those during Task A. This result provides strong evidence that, during
reversal learning, animals do not simply reuse previously formed decision manifolds. Instead, reversal
learning requires the reconfiguration of the decision manifold, consistent with the observed slower
relearning.

Fig. 4. The decision representations in the reverse task. (a,d) The neural manifold in the principal component of the decision subspace between task A and Reverse-A, illustrated with a representative recording day. (b,e) The similarity between the representation of the principal component of the decision subspace manifold in A/B/Revisit-A tasks and in the Reverse-A task. Error bars indicate standard deviation across three recording days. t-test * $P < 0.05$, ** $P < 0.01$, *** $P < 0.001$. (c,f) A linear relationship exists between the similarity of the representation of the decision subspace manifold and the number of trials required to learn the tasks. Each point represents data from a different task type (color-coded) across three recording days. Cross-shaped error bars indicate standard deviations in both axes over three days. The coefficient of determination (R^2) and p-value of the regression model compared to the null model are indicated on the graph.

Rebuttal Fig. 3. The decision representations in the reverse task. Neural manifolds in the principal component of the decision subspace during Task A and Reverse-A. (a) Decision subspace manifolds between Task A and the first 20 trials of Reverse-A (not learned stage); (b) Decision subspace manifolds between Task A and Reverse-A after reaching the learning criterion, $\geq 90\%$ correct over 15 consecutive trials (learned stage).

(3) *Additional within-session analysis: early vs. late learning in reverse task*

Take Monkey ZZ as an example, who successfully learned the reverse task, we examined whether decision
manifolds were reused across learning stages. Using a representative recording day, we compared the
decision manifolds formed during early (first 20 trials) and late stages (after reaching criterion performance)

of reversal learning to the manifold from Task A (Rebuttal Fig. 3). We found that both the initial and
 learned reversal manifolds were distinct from the Task A manifold. There was no evidence of manifold
 reuse across learning stages; rather, reversal learning involved the formation of a new decision manifold,
 distinct from those supporting continuous learning tasks.

Together, these results provide evidence that the observed shifts in neural dynamics are functionally
 meaningful rather than byproducts of recording instability or technical artifacts.

 **10. Comment 10.1: “Figure 1e: What do the error bars represent, and why are conditions**
 **B and revisit-A shown together?”**

 **Reply R1-10.1:** We thank the reviewer for this valuable suggestion. Below we clarify the error bar
 definition and describe modifications made to improve the clarity of Fig. 1e, now updated as new Fig. 2c in
 the revised manuscript.

We initially grouped Task B and Revisit-A as “post-learning tasks” following Task A. However, we agree
 that separating them improves clarity, and we have revised the figure accordingly. In new Figure 2c, we
 now separately plot the B and revisit-A conditions to enhance clarity. The bar plots show the mean
 proportion of trials required to reach the performance criterion across 9 sessions (monkeys AB and ZZ),
 and the error bars represent the standard deviation across sessions.

This revised presentation highlights learning differences across task conditions: later learned tasks (B and
 revisit-A) required significantly fewer trials to reach criterion compared to initial learning (A), indicating
 improved learning efficiency, whereas reversal learning (reverse-A) required significantly more trials (A vs.
 B: $p = 0.0096$, $g = 1.387$; A vs. Revisit-A: $p = 6.27e-06$, $g = 3.104$; A vs. Reverse-A: $p = 0.0149$, $g = -$
 1.286 ; B vs. Reverse-A: $p = 0.00037$, $g = -2.119$; Revisit-A vs. Reverse-A: $p = 1.03e-05$, $g = -2.977$).

We have incorporated these comparisons and figure updates into the Results section (line: 114–119) and
 new Figure 2c and its legend accordingly.

 **Comment 10.2: “Lines 158-159: “since the movement directions in this task were the**
 **same.” It would be helpful to quantitatively demonstrate that this is the case and that**
 **movement patterns do not change as a result of learning. Can the authors provide some**
 **limb tracking data or similar evidence to support this claim?”**

 **Reply R1-10.2:** We thank the reviewer for this insightful comment. To address this concern, we performed
 additional analyses of the monkeys' hand movement trajectories during a series of visuomotor mapping
 tasks.

Specifically, we tracked the hand positions trial-by-trial throughout a recording session, with representative
 results shown in Rebuttal Fig. 4a. Movements toward the upward button are plotted in red, and those
 toward the downward button in blue. A color gradient from dark to light represents trial progression. The
 figure demonstrates that movement trajectories remained largely consistent across time, with minor drifts.

 **Rebuttal Fig. 4. Stability of arm movement trajectories throughout learning.** (a) Hand trajectories from all trials during a single recording
 session of the sequential visuomotor association learning task. Movements toward the upper button are shown in red, and those toward the lower
 button in blue. Color intensity indicates trial progression (Darker colors indicate earlier trials, while lighter colors correspond to trials later in the
 session). (b) Comparison of movement trajectory consistency within and across task blocks. Bars represent mean Hausdorff distance for upward
 movement trajectories. Error bars indicate standard deviation across different trial pairs. "Within-block" shows correlations between trajectories
 from the same learning task, while "across-block" shows correlations between trajectories from different learning tasks. $p < 0.1$, *; $p < 0.05$, **; t-test.

 To quantitatively assess movement consistency, we computed Hausdorff distance between movement
 trajectories. Due to inherent differences in video frame rate and marker localization precision, trajectories
 for upward movements were captured with higher spatial fidelity. To account for gradual global drift that
 could occur across extended recording sessions, we performed spatial alignment by translating each
 trajectory to a common centroid without applying rotation or scaling. This approach preserved the intrinsic
 movement geometry while ensuring that overall position shifts over time did not bias our measures of
 trajectory similarity. As shown in Rebuttal Fig. 4b, we compared correlations within the same tasks
 ("within task ") versus across different tasks ("without task"). No significant difference was observed
 between within-block and across-block comparisons (two-sided t-test, $p > 0.05$), indicating that movement
 patterns remained stable across learning.

In addition, neural analyses focused on the preparatory period (after stimulus onset but before movement)
 when activity primarily reflects motor planning rather than execution. Moreover, motor goals were
 consistent across tasks, as button positions remained fixed.

We hope that these analyses adequately address the concern about movement consistency and support our
 interpretation that the observed neural patterns reflect learning-related changes rather than alterations in
 movement kinematics.

 **11. Comment 11.1: "It would be beneficial to extend the raster plot in Fig. 2a to show**
 **more of the actual neural response patterns, such as trial averages for the different**
 **stimulus conditions, to provide a clearer picture of the neural tuning of PMd neurons to**
 **stimulus and choice before the subsequent dimensionality reduction. This would help to**
 **better understand the single-neuron activity before moving on to the more abstract latent**
 **factors used in the analysis."**

**Reply R1-11.1:** We thank the reviewer for this insightful suggestion. We fully agree that examining the
 single-neuron activity patterns prior to performing dimensionality reduction provides important insight into
 the underlying structure of the recorded population responses. In response, we conducted additional
 analyses to characterize the response features of individual PMd neurons.

(1) Classification of single-neuron selectivity across monkeys

**Supplementary Fig. 5. Proportion of functionally distinct neuron types across monkeys.** (a-c) Distribution of neuronal response types in
 monkey XW (a), ZZ (b), and AB (c), shown from one representative recording day per monkey. Neurons were classified using two-way ANOVA
 (factors: stimulus identity and movement direction) with significance threshold of $P < 0.01$.

To quantify functional selectivity, we classified all recorded PMd neurons into five categories based on a
 two-way ANOVA (factors: stimulus identity and movement direction, $p < 0.01$). Neurons were categorized
 as decision-selective (tuned to movement direction regardless of stimulus), stimulus-selective (tuned to
 stimulus identity regardless of movement), linear mixed-selective (showing additive tuning to both
 variables), nonlinear mixed-selective (showing interaction effects), or non-selective (no significant tuning).
 As shown in Supplementary Fig. 5, most PMd neurons were non-selective, consistent with previous reports
 of sparse task-variable tuning in the prefrontal and anterior cingulate cortices (Smith et al., Nat. Neurosci.,
 2019), supporting the idea that a limited subset of neurons primarily contributes to task-relevant population
 coding.

(2) PSTHs illustrate functional neurons with selectivity

To illustrate concrete response features, we calculated trial-averaged peri-stimulus time histograms
 (PSTHs) aligned to stimulus onset (time = 0). Shaded areas reflect \pm s.e.m across trials. As shown in new
 Supplementary Fig. 6, decision-selective neurons (new Supplementary Fig. 6a) exhibited differential
 activity for upper versus lower movement directions. Stimulus-selective neurons (new Supplementary Fig.
 6b) demonstrated distinct temporal responses for different visual stimuli. Neurons with mixed selectivity
 (new Supplementary Fig. 6c) were jointly modulated by stimulus identity and movement direction.

Together, these analyses provide a clearer view of the single-neuron response properties underlying the
 population recordings. We have incorporated these new results in the revised manuscript (new
 Supplementary Fig. 5, 6) and supplementary information (Supplementary Notes 1.4.1, 1.4.2).

References:

Smith, E. H. et al. Widespread temporal coding of cognitive control in the human prefrontal cortex. *Nat.*
 *Neurosci.* 22, 1883–1891 (2019).

**Supplementary Fig. 6. Examples of single-neuron tuning in PMd.** Trial-averaged peri-stimulus time histograms (PSTHs) of representative PMd
 neurons recorded during the visuomotor mapping task. Neural activity is aligned to stimulus onset (time = 0). Shaded areas indicate the standard
 error of the mean (s.e.m.) across trials. (a) A decision-selective neuron showing differential activity for upper vs. lower button presses. (b) A
 stimulus-selective neuron exhibiting distinct responses to different visual stimulus. (c) A mixed-selectivity neuron modulated jointly by both
 stimulus identity and movement direction.

**Comment 11.2: “Is there a general preference for PMd neurons to respond to the upper or
 lower position of the response button on the screen?”**

**Reply R1-11.2:** To address this, we performed quantitative analyses to determine whether neurons exhibited
 a general preference for responding to movements toward the upper or lower targets during the decision-
 making period.

 **Supplementary Fig. 7. Population preference indices for upper versus lower response buttons in different tasks within one day in three**
 **monkeys.** Each bar plot shows the number of neurons exhibiting different preference index values, with positive values indicating preference for
 upper button responses and negative values indicating preference for lower button responses. Each column represents one task. Colored vertical
 lines indicate the mean preference index for each task. * denote significant deviation from zero (* $p < 0.05$, ** $p < 0.01$, *** $p < 0.001$; t-test; Mean
 value indicates the average preference index for the task).

For each neuron, we calculated a preference index (Tan et al., Nature, 2020) defined as:

$$\text{Preference Index} = \frac{FR_{upper} - FR_{lower}}{FR_{upper} + FR_{lower}}$$

where FR_{upper} and FR_{lower} denote the mean firing rates in trials with upper and lower buttons,
 respectively. We performed this analysis separately for each learning tasks (Task A, Task B, and
 Task Revisit-A) within individual recording sessions. New Supplementary Fig. 7 illustrates the
 distributions of preference indices for a representative recording day from each monkey. Each
 histogram plot reflects the neural population response across three different task blocks. Across all
 animals and tasks, the distributions were approximately centered at zero and showed no significant

directional bias in most cases (Monkey XW: Task A, $p = 0.73$, mean = 0.012; Task B, $p = 0.65$, mean = -
 0.013; Task Revisit-A, $p = 0.40$, mean = -0.02; Monkey ZZ: Task A, $p = 0.073$, mean = 0.05; Task B, $p =$
 0.88, mean = -0.004; Task Revisit-A, $p = 0.071$, mean = -0.09; t-tests). Only in Monkey AB during Task
 A was a mild significant preference observed toward the lower target ($p = 0.04$, mean = -0.07); however,
 no significant preferences were detected in Monkey AB's Task B or Revisit-A (Task B: $p = 0.28$, mean =
 0.04; Task Revisit-A: $p = 0.17$, mean = -0.06).

Moreover, the distribution of preference indices was unimodal rather than bimodal, suggesting a
 continuous spectrum of directional tuning rather than distinct subpopulations strongly favoring either target
 location.

Therefore, while occasional transient directional preferences were detected, they lacked consistency across
 tasks within one day. Consequently, it is unlikely that pre-existing directional biases underlie the observed
 reuse of task-related neural representations in PMd. We have included this result as New Supplementary
 Fig. 7 and referenced it in the revised manuscript accordingly.

References:

Tan, H.-E. et al. The gut-brain axis mediates sugar preference. *Nature* 580, 511–516 (2020).

**Comment 11.3: “To what extent are decision- and stimulus-related signals represented in**
 **PMd neurons when the animals have not learned the task? Do task-specific responses**
 **emerge as behavioral performance improves, or are stimulus- and motor-related signals**
 **always present?”**

Reply R1-11.3: We thank the reviewer for the thoughtful question regarding whether task-selective neural
 representations emerge as behavioral performance improves. Below, we present analyses specifically
 designed to assess whether decision- and stimulus-related signals in PMd are present prior to task mastery,
 and whether single-neuron selectivity increases with learning.

*(1) Comparable proportions of task-selective neurons in the progress of learning*

Each session of Task A and Task B was divided into an "early" phase (first 20% of trials) and a "late"
 phase (last 20%). During the early phase, animals operated above chance (mean accuracy = $80.64 \pm 3.70\%$,
 mean \pm s.e.m), but had not yet reached the behavioral learning criterion (late phase average = $93.01 \pm$
 2.15%).

For each phase, we assessed the proportion of neurons exhibiting decision- and stimulus-selectivity using
 two-way ANOVA. As shown in Supplementary Fig. 8, no significant differences were found between
 early and late phases for either decision ($p = 0.43$, Hedges' $g = 0.19$) or stimulus selectivity ($p = 0.38$, $g =$
 0.44). This suggests that selectivity at the unit level is not strongly modulated by learning stage.

**Supplementary Fig. 8. Neurons counts of task-selective PMd neurons before and after learning.** (a) Bar plots showing the mean \pm s.e.m
proportion of direction-selective neurons in the early (blue) and late (orange) learning phases across three monkeys, 6 days. (b) Bar plots showing
the mean \pm s.e.m proportion of stimulus-selective neurons in the early (blue) and late (orange) phases, 6 days.

*(2) No consistent enhancement in tuning strength with learning*
Second, we assessed changes in selectivity strength by calculating for each neuron a Tuning Index (TI) for
all neurons to quantify selectivity for decision and stimulus variables before and after learning (Runyan et
al., Nature, 2017), with values closer to 1 indicating stronger selectivity for decision or stimulus variables.
TI was defined as:

$$1015 \quad TI = \frac{\sum_{c=1}^C (\bar{R}_c - \bar{R}_{all})^2}{Var(R_{all})}$$

Where \bar{R}_c is the mean firing rate for condition c (decision or stimulus), \bar{R}_{all} is the overall mean firing
rate across all trials in the phase, and $Var(R_{all})$ is the variance across those trials. For each monkey, we
illustrate these changes on a representative recording day in new Supplementary Fig. 8. The bar plots
compare mean \pm s.e.m TI between the early (unlearned) and late (learned) phases, while the scatter plots
show each neuron's TI before versus after learning: points above the unity line indicate increased
selectivity, and points below indicate decreased selectivity. Across all three animals, we observed no
consistent shift in TI distributions with improved behavioral performance, indicating that task-specific
single-unit responses do not emerge as learning progresses.

These results indicate that:

- 1. Decision- and stimulus-related signals were already represented in PMd prior to task mastery;
- 2. Learning did not substantially recruit additional task-selective units nor strengthen single-neuron
selectivity.

Rather, learning appears to involve reorganization of pre-existing neuronal ensembles without significant
expansion of task-specific representations. This view is consistent with recent findings by Drieu et al.
(2025), who showed that rapid acquisition of latent task knowledge in sensory cortex occurs without
substantial modification of pre-existing stimulus selectivity.

**Supplementary Fig. 9. Changes in single-neuron TI across learning.** Bar plots showing mean \pm s.e.m TI for decision (left) and stimulus (right)
 selectivity in the early (blue) and late (orange) learning phases. Scatter plots of TI before versus after learning for all neurons. Each point represents
 one neuron; the unity line (dashed) indicates no change in selectivity. Points above the unity line denote neurons that became more selective, and
 points below denote neurons that became less selective.

We have included this result in New Supplementary Fig. 8, 9 and Supplementary Notes 1.4.3, 1.4.4
 in the revised manuscript accordingly.

References:

Runyan, C. A., Piasini, E., Panzeri, S. & Harvey, C. D. Distinct timescales of population coding across
 cortex. *Nature* 548, 92–96 (2017).

Drieu, C. et al. Rapid emergence of latent knowledge in the sensory cortex drives learning. *Nature* 2025,
 1–11 (2025).

**Minor Questions:**

**12. “Lines 82-83: we found that learning efficiency increased for later-learned pairs’.**
 **Where is this shown concretely? In monkey AB, for example, only the revisit-A condition**
 **seems to show faster learning, since the animal already knows this association. In contrast,**
 **monkey XW only shows faster learning in condition B but not revisit-A. Does this refer to**
 **the results in panel 1e?”**

 **Reply R1-12:** Thank you for pointing out this. We understand the need to clarify exactly how learning
 efficiency improves for later-learned stimulus pairs in each monkey. Below, we provide specific behavioral
 analyses across sessions to address this point. As noted in Reply R1 2.3, we collected behavioral data from
 more sessions to enhance statistical power. Error bars represent the standard deviation across sessions.
 These updated results are shown in new Fig. 2d.

Behavioral analysis revealed consistent patterns of improved learning for later-learned, structurally similar
 conditions (new Fig. 2d). For monkeys ZZ and XW, learning was significantly faster for B/C pairs than for
 A (ZZ: $p=0.00389$, Hedges’ $g=3.217$; XW: $p=0.0171$, $g=1.898$). For monkeys AB and ZZ, revisit-A was
 learned significantly faster than A (ZZ: $p=0.000648$, $g=4.574$; AB: $p=0.00579$, $g=2.359$). By contrast, the
 reverse-A task was more difficult, with significantly slower learning compared to A, B/C, or revisit-A (e.g.,
 ZZ: B vs. reverse-A: $p=0.01099$, $g=-2.566$; AB: B vs. reverse-A: $p=0.0132$, $g=-2.003$).

These results provide behavioral evidence supporting our original statement that learning efficiency
 increased for later-learned, structurally similar tasks, and decreased for structure-incongruent conditions
 (e.g., reverse task). Consequently, we have updated the Results section and new Fig. 2d to clarify this point.

**13. “In reference 21, the author name is Driscoll et al. Laura is her first name.”**

**Reply R1-13:** Thank you for bringing this citation error to our attention. We have corrected reference 21 to
 properly cite "Driscoll et al." instead of "Laura et al."

**14. “line 183: what is 'subspace t'?”**

**Reply R1-14:** Thank you for identifying this typographical error. 'Subspace t' was indeed a mistake in our
 manuscript. We have corrected this error in the revised version, replacing it with 'subspace'.

**15. “line 212: What are schema-like neural activities? Are these different from actual**
 **schema as described by the authors?”**

**Reply R1-15:** We thank the reviewer for raising this very important terminological question. A detailed
 clarification of the distinction between "schema" as a cognitive construct and schema-like neural
 representations is provided in Major Point 4.

**16. “Fig. 2e: What do the labels St 1 and St 2 mean here? Do those refer to the two**
 **stimuli that were used in task A and revisit A?”**

**Reply R1-16:** Thank you for highlighting this ambiguity in our figure labeling. Indeed, "St 1" and "St 2" in
 new Fig. 3e (original Fig. 2e) refer to the two visual stimuli that were used in both task A and revisit-A.

- • To improve clarity, we have updated new Figure 3e with more precise and consistent labeling :
- "St A1/A2" for the two stimuli used in task A
- • "St B1/B2" for the two stimuli presented in task B
- • "St C1/C2" for the two stimuli presented in task C
- • "St Revisit-A1/ Revisit-A2" for the two stimuli presented in task Revisit-A(which are the same as
 St A1/A2)

We have also added a detailed explanation in the figure legend that clearly specifies which stimuli
 correspond to which task condition. Additionally, we have standardized this labeling convention
 throughout the manuscript and all figures to ensure consistency.

**17. “Lines 180 and 223 - 225: I do not think that the analysis reveals a neural mechanism**
 **as the results are purely phenomenological. That is not a big issue but I would suggest**
 **rewording this claim to avoid it from being misleading.”**

**Reply R1-17:** We thank the reviewer for pointing out the distinction between phenomenological
 observations and mechanistic claims. To maintain scientific rigor and precision, we have revised the
 relevant text (lines: 36, 306, 347). We now state that “these results reveal neural activity patterns associated
 with schema use” instead of claiming direct mechanistic insights.

**Reviewer #2**

**“Reviewer's comment: In the manuscript entitled “Domain-specific schema reuse supports**
 **flexible learning to learn in primate brain”, Kaixi Tian and colleagues investigate**
 **population activity in dorsal premotor cortex of macaque monkeys in the context of**
 **several sensory-motor mapping tasks. They identify to orthogonal subspaces, one related**
 **to stimulus encoding and one related to motor responses / decisions. While the former is**
 **surprisingly instable, the latter replicates across tasks with a similar structure, but not a**
 **reversal learning task. On the basis of a correlation between learning speed and the**
 **similarity of subspaces across tasks, the authors suggest the preserved subspace**
 **constitutes a schema that generalizes across tasks and thus forms the basis of learning to**
 **learn. This is an interesting manuscript that addresses a timely question in an appropriate**

**animal model. I have several suggestions for the authors that they might consider to**
 **further improve the manuscript.”**

**Reply R2-0:** We sincerely thank the reviewer for the positive assessment of our manuscript and for
 the thoughtful suggestions. We have carefully considered each point and made extensive
 revisions accordingly.

In summary, we have made the following improvements:

- 1. **Statistical Reporting and Validation:** We clarified all applied statistical tests, reported p-values
 and effect sizes, updated figure legends to explain error bars and significance markers, and
 applied the Watson-Williams test to validate angular comparisons
- 2. **Methodological Clarification of Key Analyses:** We expanded explanations for LFADS (model
 principle, architecture, dimensionality choice), subspace analysis (dPCA marginalization,
 component selection), and cross-validation procedures for CNN decoding, with supporting
 analyses showing robustness across settings.
- 3. **Conceptual Terminology and Interpretation:** To avoid conflating neural and cognitive levels, we
 now refer to NCS and added a supplementary section clarifying its relationship to schema.
- 4. **Data Transparency and Presentation Improvements:** We now report separate counts of single-
 and multi-unit recordings per monkey, revised figure layouts for clarity, and incorporated key
 stimulus examples into the main text.

Below, we provide detailed responses to each comment. We hope these changes and additional analyses
 fully address the concerns raised.

**1. Comment 1.1: “The text as well as the figures are lacking statistical detail /**
 **quantification. For example, nowhere in the text do the authors mention significance,**
 **effects sizes, or the tests that they applied. This should be provided throughout the**
 **manuscript.”**

**Reply R2-1.1:** We thank the reviewer for this important comment. In response, we have revised the
 manuscript to improve the clarity and completeness of our statistical reporting:

- (1) **Statistical Tests:** We have explicitly specified the statistical tests used for all relevant comparisons (e.g.,
 paired or unpaired t-tests, Wilcoxon signed-rank tests), and these methods are now clearly described
 throughout the manuscript.
- (2) **Significance Levels (p-values):** We have included the p-values for all statistical comparisons in both
 the text and figure legends to ensure transparency and reproducibility of our findings.
- (3) **Effect Sizes:** Where appropriate, we have reported effect sizes (e.g., Hedges' g) to quantify the strength
 of observed effects, providing a better understanding of the magnitude of the differences observed.

(4) **Figure Legends:** The figure legends have been updated to clearly indicate the meaning of error bars
 (e.g., standard error of the mean, s.e.m, or standard deviation, SD), and to properly indicate statistical
 significance with standard symbols and descriptions.

We believe these changes enhance the clarity and rigor of the manuscript (lines: 88-89, 96-99, 108-114,
 1165 117-120, 149-151, 161-162, 186-191, 203, 225-226, 231-232, 261-263).

**Comment 1.2: “Furthermore, it is unclear why the authors used t-tests and not a more**
 **appropriate circular statistic for the analyses of angles.”**

**Reply R2-1.2:** In our analysis, we employed t-tests for comparing the angles between neural subspaces
 calculated for the original and shuffled data. The computation of angle was performed using the MATLAB
 function 'subspace', which calculates the principal angles between two subspaces based on their basis
 vectors. Mathematically, this involves:

- 1. Computing the orthonormal bases for both subspaces
- 2. Performing singular value decomposition (SVD) of the matrix product of these bases
- 3. Calculating the angles from the resulting singular values

This approach yields angles that represent the smallest possible rotations between subspaces, always
 falling within the first quadrant (0° - 90°), which does not involve the periodic nature typically associated
 with circular data, allowing the use of t-tests.

However, we acknowledge the reviewer's concern that angular data generally requires special statistical
 treatment. To address this, we have conducted additional analyses using the Watson-Williams test, which
 is specifically designed for circular data. Despite the constrained range of our angles, this test provides a
 more robust analytical approach for angular comparisons.

Our revised analyses using the Watson-Williams test were consistent with those from our original *t*-test
 analysis, confirming that the angle between the actual data was significantly closer to orthogonal than the
 shuffling results (Watson-Williams test, $n = 3$ days for each monkey; Monkey ZZ: $p = 0.06$, $g = 1.37$;
 Monkey AB: $p = 0.01$, $g = 1.60$; new Fig. 5a).

We have updated the Methods section of the manuscript to describe our subspace angle calculation
 approach, its constraints, and the application of the Watson-Williams test (lines: 261-262, 481-483).

**Comment 1.3: “The classification performance should be cross-validated.”**

**Reply R2-1.3:** We thank the reviewer for pointing this out. We agree that cross-validation is essential for
 assessing classifier performance. In fact, we had already implemented cross-validation in our original
 analysis, but we realize that this may not have been clearly described in the previous version of the
 manuscript. In the revised version, we have expanded the Methods section to provide a description of our
 cross-validation procedure:

- 1. Data Splitting: The entire dataset was partitioned into a training set (80%) and an independent test set
 (20%) using stratified sampling to preserve class balance.

2. K-Fold Cross-Validation: Within the training set, a 10-fold cross-validation procedure was employed.

3. Final Evaluation: We evaluated the final model performance on the held-out test set (20%).

Additionally, we have briefly mentioned cross-validation procedures in the Results section when reporting
 classification accuracies. To increase transparency, we provide a detailed summary of the 10-fold cross-
 validation results in Rebuttal Table 2 for the reviewer's reference.

**Rebuttal Table 2. 10-fold cross-validation classification accuracy across recording days for each monkey.** Mean classification
 accuracy (\pm standard deviation) obtained from 10-fold cross-validation for each recording day in each monkey (Monkey XW, Monkey
 ZZ, and Monkey AB).

	Monkey XW		Monkey ZZ		Monkey AB	
	Stimuli classification	Decision classification	Stimuli classification	Decision classification	Stimuli classification	Decision classification
Day 1	0.673 \pm 0.025	1.000 \pm 0.000	0.605 \pm 0.007	0.820 \pm 0.014	0.871 \pm 0.006	0.957 \pm 0.004
Day 2	0.482 \pm 0.009	0.918 \pm 0.006	0.818 \pm 0.008	0.945 \pm 0.004	0.892 \pm 0.092	0.846 \pm 0.008
Day 3	0.718 \pm 0.030	0.973 \pm 0.002	0.763 \pm 0.011	0.913 \pm 0.006	0.82 \pm 0.116	0.94 \pm 0.004

**Comment 1.4: "Furthermore, some statistical quantification of the similarity of results
 across the three animals should be provided."**

**Reply R2-1.4:** In the revised manuscript, we have conducted additional statistical analyses and provided
 quantifications to address this point across multiple key results:

**Fig. 2. (c)** Proportion of trials required to reach performance criterion (90% correct over 15 consecutive trials) relative to the total number of
 trials in a session. Bar plots show the mean across 9 sessions (Monkey AB: 5 days; Monkey ZZ: 4 days), with error bars indicating standard
 deviation across sessions. * $P < 0.05$, ** $P < 0.01$, *** $P < 0.001$, t-test.

1. New Fig. 2c - Consistent learning efficiency trends: Fig. 2c now includes updated behavioral statistics
 from two monkeys tested on the same task set (AB and ZZ), demonstrating consistent trends in
 learning efficiency across repeated and reversed mappings.

**Fig. 3 (c)** Generalization accuracy of decision classifiers trained on Task A when applied to Task B, Task C, and Task Revisit-A across three
 monkeys. Bars show mean accuracy across all sessions (9 sessions total: 3 from monkey AB, 3 from monkey ZZ, 3 from monkey XW); Error
 bars represent standard deviation. Each dot indicates performance from one session, with different colors representing individual monkeys.
 The grey line denotes the decision classification chance level. The accuracy of the decision classifier for monkey XW trained in task A was
 applied to tasks B/C/Revisit-A. The grey line represents the chance level of decision classification.

2. New Fig. 3c - Comparable cross-day generalization: All three monkeys exhibited similar patterns of
 generalization accuracy, supporting the robustness of our neural decoding findings. New Fig. 3c
 combined results from all three monkeys showing comparable decoding performance across multiple
 1238 days and monkeys (n = 9 total sessions; 3 per monkey; one-way ANOVA, $p=0.07$, Cohen's $f=0.56$. Fig.
 3c).

**Supplementary Fig. 4.** Principal component representations of decision-related neural subspaces across Tasks A, B, Revisit-A, and Reverse-A,
 pooled across two monkeys. Error bars indicate standard deviation across six recording sessions (3 per monkey). *, $P < 0.05$; **, $P < 0.01$; ***, $P <$
 0.001 ; t-tests.

3. New Supplementary Fig. 4 - Consistent NCS utilization patterns: Supplementary Fig. 4 now displays supplementary data from two monkeys completed the reverse task (AB and ZZ) confirming that the NCS was not reused during reverse learning.

Fig. 5. Orthogonal relationship between decision- and stimulus-related subspace. (a) The angle between the stimulus- and the decision-related subspace before and after shuffling, shown for the largest variance component. The error bar represents the variance of angles across different days.

4. New Fig. 5a - Consistent orthogonality between subspaces: New Fig. 5a right panel includes the inter-subspace angle between decision and stimulus dimensions from two animals (AB and ZZ), which both showed angles closer to orthogonality, consistent with the hypothesis of subspace separation for stability and plasticity (Watson-Williams test; $p=0.0066$, $g=2.31$; Fig. 5a, right panel).

Together, these additions provide statistical evidence supporting the similarity and generality of our findings across monkeys. We have also clarified these additions in the revised Results and Supplementary sections accordingly.

2. **“The authors report that the stimulus subspace is rather instable. This is surprising and should be further investigated. Since the stimuli in task A and in revisit A are the same, one would expect that the classifier performs above chance at least in this case. If the result holds, then this would possibly argue against orthogonal decision and stimulus subspaces, since it suggests that the stimulus subspace contains a part of the sensory-motor mapping.”**

Reply R2-2: We thank the reviewer for this insightful observation regarding the instability of the stimulus-related subspace. While this finding may seem counterintuitive given that the same stimuli were used in Task A and Revisit-A, our data supports this observation through several independent analyses.

*(1) Instability of stimulus representations*

Our decoding analyses (new Fig. 3b; new Supplementary Fig. 4b) consistently show that stimulus
 classifiers trained on Task A perform significantly worse when tested on Revisit-A, across all three
 monkeys. This degradation is also supported by dPCA projections (new Fig. 3d; new Supplementary Fig.
 4c), which show a lack of clear clustering for stimulus components when projecting Task A subspace onto
 Revisit-A activity. Although the stimuli and sensorimotor mappings were unchanged, the representations
 are not preserved across sessions.

Performance degradation varied across animals: in monkey XW, classifier accuracy dropped to chance
 (0.167), while monkeys AB and ZZ showed above-chance but significantly reduced performance (XW:
 $p=0.008$, $g=1.92$; AB: $p=0.021$, $g=1.47$; ZZ: $p=0.017$, $g=1.63$). These results underscore a consistent
 direction of effect despite inter-individual variability.

*(2) Maintenance of Orthogonality Between Subspaces*

We understand the reviewer's concern to be that, since the stimuli in Task A and Revisit-A are identical,
 stimulus representations should exhibit cross-task stability. If they do not, it might suggest that the
 stimulus-related subspace contains elements of the sensor-motor mapping, potentially violating the
 assumption of orthogonal subspaces between stimulus and decision representations.

However, if the stimulus-related subspace overlapped with the decision-related subspace, we would expect
 it to show the same cross-task stability, since sensor-motor mappings remained the same between Task A
 and Revisit-A. Instead, we observe a clear dissociation: decision decoding remains stable and transferable,
 while stimulus decoding performance declines significantly, suggesting that stimulus and decision
 information are encoded in a disentangled way.

Such a dissociation is precisely what would be expected from orthogonal subspaces: one representation
 (stimulus-related) can reorganize flexibly without altering the other (decision-related), enabling adaptive
 responses without disrupting stable decision frameworks. Similar dual-coding architectures—flexible input
 encoding alongside stable schematic control—have been reported in other brain regions, including the
 sensorimotor striatum (Kubota et al., 2009), suggesting a more general organizational principle.

*(3) Explanation of the "Stimulus Subspace"*

To address the issue raised by the reviewer, that is, the original term “stimulus subspace” may be
 somewhat misleading, we now use "stimulus-related subspace", highlighting the fact that the information
 about stimulus can be decoded in this subspace, but such information is represented in a more sophisticated
 way. We think it is a more appropriate term, which is consistent with a core functional property of the PMd.
 Rather than maintaining fixed sensory identifiers, PMd remaps stimulus representations dynamically based
 on current behavioral context. This aligns well with prior studies indicating that PMd integrates
 multisensory inputs from prefrontal and parietal cortices to support context-dependent sensorimotor
 transformations (Hoshi & Tanji, 2007; Pesaran et al., 2008).

We have added a detailed discussion of this conceptual refinement in the supplementary materials (new
 Supplementary Note 1.9).

Reference:

Pesaran, B., Nelson, M. J. & Andersen, R. A. Free choice activates a decision circuit between frontal and
parietal cortex. *Nature* 453, 406–409 (2008).

Kubota, Y. et al. Stable encoding of task structure coexists with flexible coding of task events in
sensorimotor striatum. *J. Neurophysiol.* 102, 2142–2160 (2009).

Hoshi, E. & Tanji, J. Distinctions between dorsal and ventral premotor areas: anatomical connectivity and
functional properties. *Curr. Opin. Neurobiol.* 17, 234–242 (2007).

**3. “The authors should provide a definition of “schema” in the introduction and rethink
whether “specific neural activity patterns” are indeed referred to as schemas. To me, this
seems to be mixing the neural with the cognitive level. Perhaps, specific neural activity
patterns underly schemas, but that they are schemas themselves seems a bit of a stretch.”**

**Reply R2-3:** We thank the reviewer for raising this very important terminological question. A detailed
clarification of the distinction between "schema" as a cognitive construct and schema-like neural
representations is provided in **Major Point 4**.

**4. “The authors should report separately how many multiunit sites and how many single
units entered the analysis. The compound number given at the moment is not very
informative.”**

**Reply R2-4:** We thank the reviewer for the suggestion. We have now explicitly reported the number of
single-unit (SU) and multi-unit (MU) sites included in the analysis for each monkey in the Methods section
(lines 383-385).

**Rebuttal Table 3. Neuronal recordings summary.**

Monkey	Single Units (SU)	Multi Units (MU)	Total Units
XW	225 (75±14.0/day)	246 (82±4.5/day)	471
ZZ	51 (17±4.4/day)	79 (26.3±3.1/day)	130
AB	74 (24.7±5.5/day)	53 (17.7±9.1/day)	127
Total	350	378	728

Specifically, we recorded a total of 728 units (350 SUs, 378 MUs) across three monkeys over three days.
The mean (\pm SD) unit counts per day were as follows: Monkey XW: 75 ± 14 SUs, 82 ± 4.5 MUs; Monkey
ZZ: 17 ± 4.4 SUs, 26.3 ± 3.1 MUs; and Monkey AB: 24.7 ± 5.5 SUs, 17.7 ± 9.1 MUs. A complete
breakdown of neuronal recordings is provided in Rebuttal Table 2. As noted in the manuscript, all analyses
were performed using the combined set of SUs and MUs.

**Reviewer #3**

**“In this study, the authors asked if dorsolateral premotor cortex (PMd) could**
 **learn/generate task-specific knowledge (schema in the paper) and reuse it for other tasks.**
 **They trained three of nonhuman primates (NHPs) on a sequence of visuomotor tasks,**
 **which require flexible mappings between visual stimuli and behavioral responses, and**
 **recorded multi- and single-unit activities (MUAs and SUAs) from PMd. The authors used**
 **convolutional neural networks (CNNs) to decode decisions and stimuli from PMd’s MUAs**
 **and found that PMd encodes both visual stimuli and decisions. Interestingly, their**
 **analyses suggest that only decisions appear to be stored as schema-like knowledge.**
 **Furthermore, they used demixed PCA to estimate stimulus-subspace and decision-**
 **subspace from MUAs, which provides interesting results.**

**These findings may provide valuable insights into reward-based learning and decision-**
 **making and interest readers of neuroscience studies, but I have a few concerns and**
 **questions.”**

**Reply R3-0:** We thank the reviewer for their thoughtful and detailed feedback. In response to the concerns
 regarding the methodological descriptions, we have thoroughly revised the manuscript by:

- 1. Clarified the LFADS framework, its advantages, dimensionality choice, and robustness across
 different settings.
- 2. Explained the advantages of applying LFADS before CNN decoding, supported by control
 experiments demonstrating overfitting when using raw spiking data.
- 3. Expanded subspace analysis details, including dPCA, marginalization, dimensionality selection.
- 4. Validated the use of 20 dPCA components through supplementary analyses demonstrating robustness
 across dimensionalities.
- 5. Improved figure presentation by splitting complex panels, integrating key stimuli into main figures,
 and refining legends for clarity.

In the following sections, we provide point-by-point responses addressing all specific concerns. We
 believe that these revisions and additional analyses substantially strengthen the clarity, transparency, and
 rigor of our study.

**Concerns regarding the lack of detailed description of methods:**

**the lack of detailed description of methods**

- 1. **Comment 1.1: “The authors used LFADS to obtain low-dimensional (dim=16)**
 **representations of MUAs, and I have three concerns regarding LFADS. First, the authors**
 **need to provide a brief review of LFADS.”**

**Reply R3-1.1:** We thank the reviewer for the comments. We agree that a brief explanation of this method
would enhance the clarity and accessibility of our manuscript.

LFADS is a deep learning method developed by Pandarinath et al. (2018, Nature Methods) that models
neural population activity as the output of an underlying low-dimensional dynamic system. Unlike
traditional dimensionality reduction techniques such as PCA that focus on static patterns, LFADS
leverages the sequential structure of neural data to capture the temporal dynamics that drive neural
population activity.

The architecture of LFADS consists of an auto-encoder framework with recurrent neural networks.
Specifically, LFADS employs a variational auto-encoder approach where:

- 1. An encoder RNN embeds the high-dimensional neural activity sequence into a posterior distribution
over the initial conditions of a latent dynamical system.
- 2. A decoder RNN unfolds the dynamics from these initial conditions, producing a low-dimensional
latent trajectory that captures the temporal evolution of neural population activity.
- 3. A linear readout maps the latent factors onto inferred firing rates, which are used to reconstruct the
observed spiking activity patterns.

This approach has several advantages for neural data analysis: it accounts for the inherently dynamical
nature of neural circuits, it handles trial-to-trial variability through its probabilistic framework, and it can
infer smooth underlying dynamics even from noisy spiking data.

In response to the reviewer's suggestion, we have added a brief review of LFADS to the Methods section
(line2: 412-416).

**Comment 1.2: “Second, the dimension of MUAs is reduced to 16 after LFADS. The**
**authors should explain how they came to choose this number.”**

**Reply R3-1.2:** We thank the reviewer for raising this important methodological question about our choice
of dimensionality in the LFADS analysis.

To evaluate the impact of latent dimensionality in LFADS, we tested several settings (3, 6, 16, 26, and 36
dimensions) using data from monkey XW across three recording days. As shown in new Supplementary
Fig. 3a, the reconstruction error of the spiking activity showed no significant differences across
dimensionalities (one-way ANOVA, $p = 0.43$), suggesting that increasing the number of latent factors did
not improve the model's ability to capture neural dynamics.

We also evaluated the influence of dimensionality on downstream decoding performance. As shown in
new Supplementary Fig. 3b, classification accuracies for decision and stimulus conditions remained
comparable across dimensions, with no consistent advantage at higher or lower settings. Error bars indicate
standard deviation across the three sessions. Based on these analyses, we concluded that the specific choice
of 16 dimensions does not critically affect our results and opted for this value as a representative latent
dimensionality. We have incorporated this justification in Supplementary Notes 1.3 and new
Supplementary Fig. 3.

**Supplementary Fig. 3.** Effects of LFADS dimensionality on reconstruction error and decoding performance. (a) Reconstruction error of neural
 activity under different LFADS latent dimensionalities (3, 6, 16, 26, and 36), computed as mean squared error between original and reconstructed
 spike trains (monkey XW, single session, one-way ANOVA, $p = 0.434$). (b) Decoding performance for decision (blue) and stimulus (orange)
 variables based on latent trajectories from LFADS models of varying dimensionality. Bars indicate mean classification accuracy across three
 sessions from monkey XW; Error bars denote s.d.

**Comment 1.3:** “Third, the authors used CNNs to decode decisions and stimuli from MUAs.
 Given the strong learning capability of CNNs, I assume CNNs can decode them even
 without LFADS. If so, why did the authors employ an additional step, which can cause an
 unexpected artifact?”

**Reply R3-1.3:** We thank the reviewer for raising this important methodological question about our use of
 LFADS instead of raw spike data as input for neural decoding. As detailed in our response to **Major Point**
 **3**, we selected LFADS because it performs robust denoising of neural firing rate data while effectively
 reducing dimensionality.

**2. Comment 2.1:** “The analysis of subspace is one of the main results in this study, but I
 do not see a sufficient amount of details on the analysis. I think this may be because the
 authors adopted the demixed PCA proposed by an earlier study (reference 52 in the
 manuscript). While this may be clear to the authors themselves, it is still advisable for
 them to provide explanations for their readers.”

**Reply R3-2.1:** We thank the reviewer for highlighting the need for more detailed explanation of our
 subspace analysis methods. As described in **Major Point 1**, we have provided additional details regarding
 the dPCA and our specific implementation.

**Comment 2.2:** “Also, the authors stated in line 333, “Trials under different conditions
 were averaged to obtain x_{avg} ., which was then marginalized [53]”. As far as I can see,
 they used the marginalization proposed in reference 52, so the reference should be 52.”

**Reply R3-2.2:** We thank the reviewer for pointing out this citation error. We have corrected this error in
 the revised manuscript.

 **3. “In their analysis, the authors used 20 principal components. I would like to know
 how they reached the conclusion that 20 components are good enough.”**

 **Reply R3-3:** We thank the reviewer for raising this question about our selection of 20 principal
 components for the dPCA analysis. As discussed in **Major Point 2**, we have conducted supplementary
 analyses that examine how different dimensionality choices affect our results (new Supplementary Fig. 11).

 **Major Questions:**
 **4. “The two earlier studies [1, 2] showed that anterior cingulate cortex (ACC) was
 responsive to the reduction of rewards and essential in maximizing the rewards. The
 experimental protocol used in the two earlier studies is similar to the protocol used in
 this study (switching from Task-A to Reverse-A) in my opinion. Can the authors provide
 any insights into the relationships between their study and the two earlier studies [1, 2]?”**

 **Reply R3-4:** We appreciate the reviewer’s thoughtful question regarding the relationship between our
 findings and the earlier studies by Shima & Tanji (1998) and Kennerley et al. (2006). We summarize the
 similarity and difference across these studies in terms of task designs and behavioral observations, neural-
 level findings, and discuss possible theoretical mechanisms:

*(1) Task Design and Behavioral Observations:*

As the reviewer noted, the experimental paradigm in these studies—requiring monkeys to adjust behavior
 following reward reduction—is similar to our Task-A to Reverse-A switching. Shima & Tanji (1998)
 reported delayed behavioral switching after reward reduction. Consistent with these findings, we observed
 that monkeys showed significantly impaired performance in the Reverse-A condition compared to novel
 task learning, suggesting a shared behavioral challenge in adapting to reversal.

*(2) Neural-Level Findings*

Shima & Tanji (1998) identified reward-sensitive activity in CMa neurons, selectively emerging after
 reward reduction and preceding action switching, suggesting a role for ACC structures in triggering
 behavioral adjustment signals. Walton et al. (2006) further demonstrated that ACC lesions impaired the
 integration of reward history for sustained action selection. In our study, we found that schema-like low-
 dimensional manifolds in PMd, typically supporting generalization across tasks, were disrupted during
 reversal learning, necessitating reorganization. Notably, known anatomical connections from CMa to
 PMd (Bates & Goldman-Rakic, 1993) suggest that ACC-originated evaluative signals could influence
 schema remodeling processes in PMd during flexible adaptation.

Together, these studies suggest a possible distributed functional model: the ACC monitors reward-outcome
 histories and initiates adaptation when reward structures change, while downstream regions such as PMd
 implement the required reorganization of action representations. Our findings extend this framework by

illustrating how preserved NCS subspaces in PMd facilitate flexible learning within consistent contexts but
 must be rebuilt when core task structures are reversed, a process that appears slower and less efficient.

We appreciate the reviewer's suggestion to explore these connections more fully. We have revised the
 Discussion section to include this perspective (lines: 336-340).

References:

Bates, J. F. & Goldman-Rakic, P. S. Prefrontal connections of medial motor areas in the rhesus monkey. *J.*
 *Comp. Neurol.* 336, 211–228 (1993).

**5. “In lines 183-185, the authors state “we identified the major components in these**
 **two subspaces that explained the highest variance in the data, and then calculated the**
 **angle between them (see methods for details)”. Does this mean that they used a single**
 **principle component of each subspace? If so, how much information is lost by ignoring**
 **other components? Can the authors provide any insights?”**

Reply R3-5: We thank the reviewer for this important question regarding our subspace analysis
 methodology. To clarify, we did not use only a single principal component from each subspace when
 calculating the angles.

**Supplementary Fig. 13. Cumulative variance explained by each dimension in the decision and stimulus subspaces.** For each monkey, the
 cumulative variance explained by individual dimensions is shown for the decision (orange) and stimulus (blue) subspaces. Data are averaged across
 3 recording days. Error bars denote standard deviation across three recording days.

For the stimulus-related subspace, which had a higher dimensionality, we selected the components that
 explained more than 1% of the variance and together accounted for at least 70% of the total variance in that
 subspace. For the decision subspace, all components explaining more than 1% variance were included. As
 shown in new Supplementary Fig. 13, we included nearly all decision subspace dimensions. Due to the
 greater dimensionality of the stimulus-related subspace, we applied a 70% variance threshold, which
 corresponded to 3 dimensions for Monkey AB and 4 dimensions for Monkey ZZ.

We have revised the manuscript’s Methods to clarify our criteria for subspace dimensionality selection in
 the angle analysis (lines: 250-257, New Supplementary Fig. 13).

 **“[1] Shima, K., and Tanji, J. (1998). Role for cingulate motor area cells in voluntary movement
 selection based on reward. Science 282, 1335–1338.**

**[2] Walton, M.E., Behrens, T.E.J., Buckley, M.J., Rushworth, M.F.S., and Kennerley, S.W. (2006).
 Optimal decision making and the anterior cingulate cortex. Nat. Neurosci. 9, 940–947.”**

 **Minor Questions:**

**6. “Eq. 1 is not very informative. It should be modified to clarify that the two
 convolutional layers are used. A more explicit expression for convolution would also be
 more helpful.”**

 Reply R3-6: We thank the reviewer for this suggestion. We have modified Equation 1 to explicitly show
 the two-layer convolutional architecture:

$$\begin{aligned}
 H_1 &= P(f(X * W_1 + b_1)) \\
 H_2 &= P(f(H_1 * W_2 + b_2)) \\
 H'_2 &= D(H_2, p_1) \\
 Y &= \varphi(H'_2)
 \end{aligned}$$

Where * denotes one-dimensional convolution along the temporal axis. W_1, W_2 are learnable
 convolutional filters in the first and second layers. b_1, b_2 are bias terms added after convolution.
 $f(\bullet)$ is an element-wise nonlinearity, implemented as ReLU (Rectified Linear Unit). $P(\bullet)$ indicates
 max-pooling applied after activation. $D(H; p)$ denotes dropout with drop probability p , applied to
 reduce overfitting. $\varphi(\bullet)$ represents softmax output mapping. This more comprehensive formulation
 illustrates the two convolutional layers with a ReLU nonlinearity between them, providing readers with
 a better understanding of our decoder architecture.

The revised manuscript added these in line 422-426.

 **7. “In line 312, the authors stated “T is the number of spikes across time”. I think they
 meant it say “T” is the number of time bins instead of the number of spikes. Please
 clarify this point.”**

 Reply R3-7: Thank you for pointing this. "T" indeed refers to the number of time bins rather than the
 number of spikes. We have corrected this statement to read: "where T is the number of time bins in the
 recording."

**8. “Figure 1 contains too much information. It may be more readable to the readers if**
**they split it up into several figures.”**

**Reply R3-8:** We thank the reviewer for this suggestion. In response, we have divided the original Figure 1
into two separate figures:

New Fig. 1 now focuses exclusively on the conceptual framework of the stability–plasticity dilemma and
schema reuse. New Fig. 2 presents the experimental paradigm and behavioral analysis.

**9. “I think they should include all stimuli shown in supplemental Fig. 1 in the main text.”**

**Reply R3-9:** We appreciate this helpful suggestion. To improve clarity while maintaining figure simplicity,
we have updated new Fig. 2b to include visual representations of the stimuli used on different days (shown
in Reply R1-2.1), thereby providing a clearer overview of the task structure in the main text. At the same
time, we have retained Supplementary Fig. 1 to provide a comprehensive display of all stimuli used, for
ease of reference.

**10. “The legend of Fig. 2D can be more explicit. The term “direction is too vague here.**
**Perhaps, “up” and “down” may be more appropriate.”**

**Reply R3-10:** Thank you for noting this ambiguity. We have revised the legend of new Fig. 3d (old Fig. 2d)
to explicitly use the terms “up” and “down” in place of the vague term “direction,” clarifying the response
locations. We have also reviewed all figure legends to ensure consistency and clarity across the manuscript.

Manuscript NCOMMS-24-73204B

Response to Reviewers

We sincerely thank the reviewers for their positive evaluation of our previous revision and the constructive
feedback provided in this round. We have carefully considered each comment and made further revisions
to the manuscript accordingly, including new analyses and expanded explanations. All relevant changes
have been highlighted in blue in the revised manuscript.

To highlight the most significant revisions:

• **Cross-animal data integration and standardized statistical analysis.** Behavioral comparisons
across three monkeys were encompassed in new Fig. 2c. In Figure 5, Monkey XW's data was added
and computational errors in monkey ZZ's data were corrected. Corresponding corrections were also
made in Supplementary Figures 11, 13, 14 and 15. Consistency with original findings was maintained,
further confirming the cross-individual generalizability of our conclusions.

• **Mechanistic elucidation and discussion of stimulus-related subspaces.** Linear toy models
(Supplementary Fig. 17) and detailed Supplementary Note 1.9 were established to explain the encoding
characteristics of stimulus-related subspaces. Potentially misleading expressions throughout the
manuscript were corrected, and it was clarified that PMd encodes stimulus-related information rather
than stimulus identity.

• **Statistical validation and theoretical support for near-orthogonal relationships.** Statistical
standards for near-orthogonal analyses were unified (*, $p < 0.05$). Statistical analyses in Figure 5b were
enhanced, scatter plot visualization was optimized, and intuitive understanding of the near-orthogonal
relationship between stimulus-related and decision subspaces was provided through geometric
illustrations in Supplementary Fig. 18. Added discussion of near-orthogonal neural encoding in
Supplementary Notes 1.12.

• **Schema analysis and in-depth discussion of reversal learning.** Neural analysis for reversal tasks
was supplemented in Supplementary Fig. 19, and new Supplementary Note 1.11 was added to clarify
that reversal tasks require construction of new NCS rather than simple mapping reversals, with
discussion of NCS characteristics and scope limitations.

• **Comprehensive optimization of figure visualization and presentation.** To provide better
visualization of data distributions, all bar plots in the manuscript were enhanced with individual data
points, with box plots replacing bar plots when sample sizes exceeded 10. Figure visualization effects
were optimized (Supplementary Figures 6, 7), figure captions were corrected, and statistical tests
comparing decoding accuracies with chance levels were added.

These revisions enhance our understanding of domain-specific neural representation dynamics in premotor
cortex during associative learning. We thank all reviewers for their valuable suggestions, which have
substantially improved the scientific rigor and academic impact of our work.

We include a point-by-point response to the individual referees following.

Page references are provided as follows:

- • Reviewer #1: Page 2-6
- • Reviewer #2: Page 6-18
- • Reviewer #3: Page 18

**Reviewers' Comments to the Authors:**

**Point-by-point responses to the reviewers (Reviewers' comments in bold Arial font, our**
**responses in regular Times New Roman font):**

**Reviewer #1:**

**“The authors have done a great job in addressing my main concerns and I appreciate the**
**numerous and thoughtful additions they have made in the revised manuscript. The revised**
**version of the text is substantially improved and particularly benefits from the improved**
**explanations of the experimental design, more detailed methods, and additional control**
**analyses that support the main conclusions. In particular, the experimental design of**
**changing the learning conditions within the same recording day is now easier to follow.**
**Moreover, the rationale behind the subspace and orthogonality analyses is quite clear now,**
**and the inclusion of decoder generalization and shuffled controls strengthens the**
**interpretation of the dPCA results.**

**I therefore generally support the publication of the manuscript in its current form and think**
**that it makes a useful contribution to our understanding of changes in neural coding due to**
**short- term associative learning.”**

**Reply R1-0:** We sincerely thank the reviewer for the positive evaluation and recognition of our revised
manuscript. We also appreciate your additional constructive suggestions, which we have carefully addressed
below through further revisions and analyses.

**“Below are a few minor points for the authors' discretion:**

**1. “The new PSTHs are a helpful addition to illustrate the functional selectivity of individual**
**neurons. However, the stimulus-selective examples are somewhat difficult to interpret**
**visually, likely due to the superposition of four traces in each panel. Using a larger bin size**
**or separating some of the stimulus conditions in two panels might make the tuning**
**selectivity more visible. Also, the figure caption currently states that each panel shows one**
**neuron, although multiple examples are included.”**

**Reply R1-1:** According to the suggestion, we have made the following improvements and corrections:

- 1. We have increased the temporal bin size to 200ms across all PSTH plots, which significantly enhanced
the visibility of stimulus-selective responses (see new Supplementary Fig. 6).
- 2. We have corrected the figure caption to accurately reflect that multiple neuronal examples.

Overall, we have applied larger smoothing windows and reselected examples to update the new
Supplementary Fig. 6, enabling clearer visualization of selectivity, and have corrected inappropriate
expressions in the figure caption.

**New Supplementary Fig. 6. Examples of single-neuron tuning in PMd.** Trial-averaged peri-stimulus time histograms (PSTHs) of representative
 PMd neurons recorded during the visuomotor mapping task. Neural activity is aligned to stimulus onset (time = 0). Shaded areas indicate the standard
 error of the mean (s.e.m.) across trials. (a) Decision-selective neurons showing differential activity for upper vs. lower button presses. (b) Stimulus-
 selective neurons exhibiting distinct responses to different visual stimulus. (c) Mixed-selectivity neurons modulated jointly by both stimulus identity
 and movement direction.

**2. “Supplementary Figure 7 showing the direction preference index is a useful addition.**
 **However, to further aid interpretability, it would be good if the x-axes across all panels were**
 **scaled consistently from -1 to 1. The distributions appear centered near 0, but I wonder if**
 **this reflects a large fraction of neurons with no spatial preference or simply a majority that**
 **do not respond to button presses at all. Was this analysis restricted to neurons that were**
 **significantly modulated by the button press? Some clarification on this point would be**
 **helpful.”**

**Reply R1-2:** We appreciate the question regarding the interpretation of the direction preference index
 distribution and the valuable suggestion for improving figure clarity. We first adjusted the x-axes in new
 Supplementary Fig. 7, and to further address the reviewer's question, we present the proportion of neuron
 types for each monkey as well as the preference indices of selective neurons.

(1) *Standardized axis scaling for improved comparison*

We have rescaled all x-axes across panels in the new Supplementary Fig. 7 to range consistently from -1 to 1,
 enabling better cross-panel comparison and interpretability.

**New Supplementary Fig. 7. Population preference indices for upper versus lower response buttons in different tasks within one session in**
 **three monkeys.** Each bar plot shows the number of neurons exhibiting different preference index values, with positive values indicating preference
 for upper button responses and negative values indicating preference for lower button responses. Each column represents one task. Colored vertical
 lines indicate the mean preference index for each task. * denotes significant deviation from zero (*, $p < 0.05$, **, $p < 0.01$, ***, $p < 0.001$; t-test; Mean
 value indicates the average preference index for the task).

(2) Proportion of Neuron Type and Selective Neuron Analysis

The analysis in new Supplementary Fig. 7 included all recorded neurons. To provide a more comprehensive
 characterization of different neuron types, we added Rebuttal Fig. 1, showing pie charts of neuron types
 averaged across three recording sessions for each of the monkeys. Monkeys AB and ZZ were dominated by
 non-selective neurons, whereas monkey XW exhibited a higher proportion of mixed-selective neurons.

 **Rebuttal Fig. 1. Population breakdown of neuron selectivity types across animals.** Pie charts showing the distribution of neuron selectivity types
 for each monkey, averaged across three recording sessions.

*(3) Restricted analysis of selective neurons confirms absence of directional bias*

To further address the reviewer's concerns, we conducted a restricted analysis examining the distributions of
 preference indices for a representative recording session from each monkey, including only significantly
 selective neurons (mix-selective, stimulus-selective, and direction-selective; see Rebuttal Fig. 2). Across all
 animals and tasks, the distributions were approximately centered at zero and showed no significant directional
 bias.

The results from selective neurons remained consistent with our original findings. Therefore, pre-existing
 directional bias is unlikely to be the foundation for the observed task-related neural representation reuse in
 PMd. We greatly appreciate the reviewer's suggestions. In the manuscript, we chose to retain the original
 analysis including all neurons and optimized the visualization of new Supplementary Fig. 7 following the
 reviewer's recommendations, as this approach provides a more comprehensive demonstration of the overall
 data characteristics and distribution patterns.

 **Rebuttal Fig. 2. Population preference indices for upper versus lower response buttons in different tasks within one session in three monkeys,**
 **restricted to selective neurons.** Each bar plot shows the number of selective neurons exhibiting different preference index values, with positive
 values indicating preference for upper button responses and negative values indicating preference for lower button responses. Only neurons showing
 significant selectivity for stimulus identity, movement direction, or both (mixed-selective) were included in this analysis. Each column represents one
 task. Red dashed vertical lines indicate zero preference. (t-test; Mean value indicates the average preference index for the task).

**Reviewer #2**

**“The authors’ substantial revision have significantly improved the manuscript and have**
 **helped in clarifying the rationale, approach, and analyses. However, they have also brought**
 **several points to the fore that were not evident in the first submission.”**

**Reply R2-0:** We sincerely thank the reviewer for the positive evaluation of our previous revision and for
 the new constructive suggestions. We have carefully considered each point and made revisions accordingly.

**1. Comment 1.1: “Most of the neural data seems to stem from monkey XW, yet the behavior**
 **of this animal seems to be quite different from the other two animals. Figure 1c suggests**

**that neither the behavior in condition C nor in condition revisit A is similar to monkeys AB**
 **and ZZ. Condition reverse A was not acquired in XW. The authors seem to deal with this by**
 **excluding XW from the main analysis. It seems that a better characterization of the overall**
 **behavior would be to include XW in the pool.”**

 **Reply R2-1.1:** We appreciate the reviewer's suggestion. Understanding the reviewer's concern regarding the
 behavioral inconsistency observed in Monkey XW relative to the other subjects, we address this issue by
 first explaining the source of these behavioral differences and subsequently providing cross-subject statistical
 analyses pooling data from all three animals:

*(1) Pre-training-induced behavioral differences*

Regarding the behavioral differences of Monkey XW compared with Monkeys AB and ZZ, which the
 reviewer refers to in Fig. 1c (possibly Fig. 2d), we discussed this issue in Supplementary Note 1.1. This
 stems from differences in pre-training history across the three animals. Before the recording, Monkeys AB
 and ZZ were trained using fixed visual stimulus pairs to familiarize them with the basic task structure (visual
 stimulus → button press). Then, along with the start of the electrophysiological recordings, the sequential
 learning paradigm, i.e., learning of multiple new stimulus pairs followed by reversal learning (A, B, Revisit-
 170 A, Reverse-A), was introduced to them. In contrast, Monkey XW received additional training on sequential
 learning of new visual stimulus–action pairs (A, B, C, Revisit A) prior to the start of electrophysiological
 recordings. These pre-trained experiences did influence baseline learning speed: behaviorally, Monkeys AB
 and ZZ generally required more trials to learn new stimulus-action mappings compared to Monkey XW. On
 average, monkeys required the following number of trials to reach the learning criterion for task A (mean ±
 SD): Monkey AB: 95.6 ± 38.2 trials per session; Monkey ZZ: 116.5 ± 37.25 trials per session; Monkey XW:
 25.13 ± 8.79 trials per session. Therefore, as the reviewer noted, Monkey XW did not show statistically
 significant differences in the C and Revisit-A conditions because this monkey was already capable of
 learning new tasks (i.e., Task A) rapidly, which likely resulted in non-significant differences for the
 subsequent C and Revisit-A tasks. Nevertheless, the overall trend of accelerated learning (learning speed of
 task A vs B, A vs C, and A vs Revisit-A) was still present in XW.

*(2) Cross-subject statistics for all three monkeys*

Monkey XW was previously excluded from the cross-animal behavior analysis due to paradigmatic
 differences as the reviewer mentioned. We agree with the reviewer that it would be helpful to include all
 three monkeys and present the overall results. Accordingly, since only Monkey XW performed task C, we
 grouped tasks B and C together as both represent subsequent learning of new, similar stimulus–motor
 mappings. The revised analysis (in new Fig. 2c) confirmed our original findings, that is, later learned pairs
 exhibited significantly increased efficiency, while the reverse learning task was significantly slowed (pooled
 $n = 14$ sessions for each condition, i.e., 5 from Monkey AB, 5 from Monkey XW and 4 from Monkey ZZ;
 A vs. B/C: $p = 2.30E-04$, $g = 1.47$; A vs. Revisit-A: $p = 1.52E-06$, $g = 2.34$; A vs. Reverse-A: $p = 1.20E-03$,
 $g = -1.59$; B/C vs. Reverse-A: $p = 2.07E-07$, $g = -2.82$; Revisit-A vs. Reverse-A: $p = 1.02E-07$, $g = -3.37$).

**New Fig. 2.** (c) Proportion of trials required to reach performance criterion (90% correct over 15 consecutive trials) relative to the total number of
 trials in a session across three monkeys. Bar plots show the mean across 14 sessions (Monkey AB: 5 sessions; Monkey ZZ: 4 sessions; Monkey XW:
 5 sessions), with error bars indicating standard deviation across sessions. Colored scatter dots represent individual sessions, with each color
 corresponding to one monkey. *, $P < 0.05$, **, $P < 0.01$, ***, $P < 0.001$, t-test.

**Comment 1.2:** “Furthermore, it would be important to show that the neural results across
 all animals are not biased towards the idiosyncratic behavior of XW given that this animal
 contributed substantially more data.”

**Reply R2-1.2:** We thank the reviewer for raising the concern that monkey XW's substantial neural data
 contribution might cause bias. We address the reviewer's concerns in the following aspects:

(1) *Session-level analysis design prevents data pooling bias*

Our neural analyses were conducted at the session level (session-by-session, animal-by-animal analysis)
 rather than pooling all neural data together. This ensures that XW's large dataset does not dominate statistical
 comparisons, as each animal contributes equally to cross-animal statistics ($n = 3$ sessions per animal). We
 have supplemented this content in the Methods section (P13, Line 420) to avoid misunderstanding.

(2) *Main results include both individual animal statistics and cross-animal consistency validation*

Our main results demonstrate robustness through both individual animal statistics and cross-animal
 consistency validation, addressing concerns about potential bias from XW's larger data contribution. Below,
 we present the statistical significance at both individual monkey and cross-subject levels for each main neural
 finding:

- Decision-related decoders exhibit stable cross-task transfer while stimulus-related decoders do not. This
 finding is supported by individual animal analyses (Supplementary Fig. 4a) and confirmed through cross-
 animal consistency analysis (Fig. 3c).

- The degree of manifold reuse correlates with learning efficiency. This analysis involved two animals (AB
 and ZZ) since Monkey XW did not perform the reversal task. Both individual monkey statistics (Fig. 4) and
 cross-animal consistency analysis (Supplementary Fig. 12) validate this relationship.

- Decision-related and stimulus-related subspaces exhibit near-orthogonal relationships. While Monkey XW
 was initially excluded due to paradigmatic differences, we have now updated new Fig. 5a and new

Supplementary Fig. 15 to include comprehensive results both individual animal analyses and cross-animal
 validation.

**New Fig. 5a. Near-orthogonal relationship between decision- and stimulus-related subspace.** (a) The angle between the stimulus- and the
 decision-related subspace was compared with angles obtained after shuffling the decision and stimulus labels. Data are shown as box-and-whisker
 plots. The line within each box indicates the median; box edges represent the first (Q1) and third (Q3) quartiles; whiskers show the full data range.
 Colored scatter dots represent individual sessions, with each color corresponding to one monkey. (Original group: n = 3 per monkey; Shuffled group:
 n = 30 per monkey). *, P < 0.05, **, P < 0.01, ***, P < 0.001, Watson–Williams test.

**New Supplementary Fig. 15.** Angle between the stimulus- and decision-related subspaces combined across three monkeys, with black asterisks
 indicating comparisons with angles after shuffling decision and stimulus labels. Data are shown as box-and-whisker plots. The line within each box
 indicates the median; box edges represent the first (Q1) and third (Q3) quartiles; whiskers show the full data range. Colored scatter dots represent
 individual sessions, with each color corresponding to one monkey. *, P < 0.05, **, P < 0.01, ***, P < 0.001, NS, not significant, Watson–Williams
 test.

**2. Comment 2.1: “There are several inconsistencies regarding the stimulus**
 **representation in PMd. First, it seems that the results from the decoding analyses and**
 **the dPCA analyses are somewhat contradictory. The decoding analyses suggest that**
 **there is no “stable” representation of the stimuli, to the extent that the same stimuli**
 **cannot be decoded with the same classifier twice (A and revisit A). This is surprising,**
 **given that the authors themselves write that “PMd encodes the identity features of visual**
 **stimuli” (p11, line 314), and should be clarified.”**

**Reply R2-2.1:** We thank the reviewer for raising this important question regarding the contradiction between
 our decoding and dPCA analyses results. We appreciate the opportunity to address this issue.

First, concerning the statement "PMd encodes the identity features of visual stimuli" (p11, line 314) in our
 manuscript, we agree that PMd does not maintain fixed sensory identifiers, but rather dynamically remaps
 stimulus representations based on current behavioral context. We have therefore revised this expression as
 "It is conceivable that the PMd encodes visual stimulus-related information." (P11, Line 324).

**Comment 2.2: "On the other hand, the dPCA analyses seem to suggest that stimulus**
 **representations are well separable, according to the authors (p9, line 265). How can the**
 **representations be well separable but unstable?"**

**Reply R2-2.2:** We thank the reviewer for this question about how stimulus-related representations can be
 separable but unstable. This indeed warrants careful discussion as it involves understanding two dimensions of
 neural representations and the underlying key mechanisms.

*(1) Conceptual definition of separability and stability with neural data evidence*

First, we would like to further clarify the meaning of separability and stability in neural activity mentioned by the
 reviewer. Separability here refers to the ability to distinguish different conditions, reflected in distinct patterns of
 neural activity. Stability refers to the consistency of neural representations for the same condition across different
 265 times (e.g., A vs. Revisit A). These two concepts are not contradictory, as separability operates within conditions
 while stability operates across conditions.

Our decoding analyses provide multiple lines of evidence for this phenomenon. The left bar in Fig. 3b
 demonstrates that decoders can effectively decode a series of visual stimuli (stimuli in tasks A, B, and C),
 confirming that stimulus-related dimensions exhibit good separability within their respective conditions. The right
 270 bar in Fig. 3b further demonstrates that decoders trained on A/B/C tasks cannot effectively work in the revisit-A
 271 task, indicating cross-task instability of visual stimulus representations. The dPCA result in the bottom-right panel
 of Fig. 3d also shows inconsistency in manifold representations between tasks A and revisit-A. These lines of
 evidence from decoding analyses and manifold visualization confirm that stimulus-related subspaces exhibit both
 separability and instability.

*(2) Toy model for understanding dynamic encoding*

In our case, the separable but unstable representations may reflect context-dependent dynamic remapping of
 stimulus information in PMd, as discussed in Supplementary Notes 1.9. Visual stimuli in tasks A, B, and C can
 be distinguished within their contexts, but following new learning experiences and contextual influences, the same
 visual stimuli become difficult to distinguish and unstable between tasks A and Revisit-A. This suggests that
 stimulus-related information representations are influenced by contextual factors, leading to dynamic
 reorganization of the encoding space while preserving within-context discriminability.

To better understand this separable but unstable dynamic coding approach, we constructed a linear toy model of
 dual-context classification tasks to elucidate dynamic encoding, demonstrating the possible mechanisms
 underlying the coexistence of separability and instability. We defined two stimulus templates $V_1 = [1.0, 0]^T$ and
 $V_2 = [-0.6, 0.4]^T$, and implemented context-dependent encoding through linear transformation matrices: identity

transformation $T_0 = I$ (context 0) and rotation transformation $T_1 = R(\theta)$ (context 1), where $R(\theta) =$
 $[\cos\theta \ \sin\theta; -\sin\theta \ \cos\theta]$, $\theta = 10^\circ$. The data generation process was: $X_{c,i} = T_c V_i + \varepsilon$, where $\varepsilon \sim N(0, \sigma^2 I)$,
 $\sigma = 0.25$. As shown in new Supplementary Fig. 17, the left panel displays the representational patterns of two
 stimulus categories in context 0, with dark blue circles and dark red triangles representing the response
 distributions of the two stimulus types, exhibiting clear separability. The right panel shows the same stimulus
 categories in context 1, marked with light colors, which maintain within-class clustering and between-class
 separation after 10° rotation transformation.

This toy model demonstrates the coexistence of separability and instability: (1) Within-context separability: within
 each context, the two stimulus categories can be effectively distinguished; (2) Cross-context instability: classifiers
 trained in context 0 show degraded performance in context 1, reflecting dynamic reorganization of neural
 representations. The above content demonstrates that characteristics that are both separable and unstable can exist,
 and stimulus-related subspaces exhibit precisely such features. We have added discussion of this content in new
 Supplementary Notes 1.9.

**New Supplementary Fig. 17. Linear toy model demonstrating separable but unstable dynamic encoding.** Left: Two stimulus categories (blue
 circles and red triangles) are linearly separable in Context 0 under identity transformation. Right: The same stimulus categories remain separable in
 Context 1 after 10° rotation.

**Comment 2.3: “Here, the authors should also double check which figures they are referring**
 **to. At the moment, the reader is asked to look at Figure 4b to illustrate the separability, but**
 **this figure shows the decisionspace, not the stimulus space.”**

**Reply R2-2.3:** Thank you for pointing this out, and we apologize for this figure reference error. Figure 4b here
 should be Figure 5b. We have corrected this reference in the revised manuscript and systematically checked the
 entire manuscript for other similar errors to ensure all figure references are accurate.

**Comment 2.4: “The scatter plot in 5b does not show good separability in the stimulus**
 **space.”**

**Reply R2-2.4:** We sincerely thank the reviewer for raising the important concern about the lack of good
 separability in the scatter plot of Fig. 5b. This observation provides an opportunity to clarify the nature of stimulus-
 related subspace. We respond from three aspects: first explaining why the distinction is not very obvious on the

one-dimensional axis scatter plot, second improving figure readability and providing statistical analysis of
 separability through updated visualizations, and finally demonstrating enhanced figure visualization to show the
 advantages of near-orthogonal encoding.

(1) Feature compression limitations of stimulus-related dimensions

Fig. 5b showed only the first component of the stimulus-related subspace. However, as shown in new
 Supplementary Fig. 13, which presents the variance proportion of each dimension after dPCA including results
 from Monkey XW, we can demonstrate that the variance contribution of each axis component in stimulus-related
 dimensions is relatively small and dispersed. Therefore, single-dimensional visualization may underestimate the
 true discriminative capacity of the stimulus-related subspace.

**New Supplementary Fig. 13.** Cumulative variance explained by each dimension in the decision and stimulus subspaces. For each monkey, the
 cumulative variance explained by individual dimensions is shown for the decision (orange) and stimulus (blue) subspaces. Data are averaged across
 3 recording sessions. Error bars denote standard deviation across three recording sessions.

(2) Added statistical results

Nevertheless, new added statistical results and new Supplementary Fig. 16 (since plotting four distributions
 together makes it difficult to observe their separability, we conducted pairwise plotting analysis) show that
 stimulus categories in the stimulus-related dimension exhibit significant differences within the same direction,
 enabling discrimination of different visual stimuli (St A1 vs St B1: $p = 0.0018$, $\eta^2 = 0.0084$; St A2 vs St B2: $p <$
 0.001 , $\eta^2 = 0.0082$; St A1 vs St A2: $p = 0.5337$, $\eta^2 = 0.0081$; St A1 vs St B2: $p < 0.001$, $\eta^2 = 0.0086$; St A2 vs St
 B1: $p = 0.0003$, $\eta^2 = 0.0080$; St B1 vs St B2: $p = 0.15$, $\eta^2 = 0.0085$), while those within the same direction show
 no significant differences in the decision dimension (St A1 vs St B1: $P = 0.53$, $\eta^2 = 0.0084$; St A2 vs St B2: $p =$
 0.46 , $\eta^2 = 0.0082$; St A1 vs St A2: $p = 0.00$, $\eta^2 = 0.0081$; St A1 vs St B2: $p < 0.001$, $\eta^2 = 0.0086$; St A2 vs St B1:
 $p < 0.001$, $\eta^2 = 0.0080$; St B1 vs St B2: $p < 0.001$, $\eta^2 = 0.0085$). In the decision dimension, directions show
 significant statistical differences ($p < 0.001$, $\eta^2 = 0.0041$), while in the stimulus-related dimension, there is no
 statistical significance ($p = 0.49$, $\eta^2 = 0.0041$). These statistical results further validate the near-orthogonal
 organizational advantages demonstrated in Fig. 5b, confirming the ability to achieve relatively independent
 encoding of stimulus and decision information with minimal cross-dimensional interference.

**Supplementary Fig. 16. Pairwise analysis of stimulus separability in stimulus-related dimension.** Each panel shows scatter plots and distributions
of pairwise visual stimuli in both stimulus-related and decision dimensions.

*(3) Enhanced figure visualization to demonstrate near-orthogonal encoding advantages*
The scatter plot in Figure 5b aims to illustrate the advantages of near-orthogonal encoding, that is, along the
stimulus-related dimension, different visual stimuli show relatively good discrimination while motor decisions do
not; conversely, in the decision dimension, motor decisions are clearly different while visual stimuli are primarily
separated by movement direction. In the original Figure 5b, we could observe that green and yellow stimuli
(representing stimuli for the same button-press direction in tasks A and B) can be well distinguished on the
stimulus-related axis, but cannot be distinguished in the decision dimension.

**New Fig. 5b. Near-orthogonal relationship between decision- and stimulus-related subspace.** (b) Distribution of data for individual trials of
monkey AB on the stimulus-decision plane.

To more clearly demonstrate the advantages of near-orthogonal encoding in providing relatively independent
encoding of different dimensional information with reduced interference, and to address the reviewer's concerns,
we updated new Fig. 5b by adding decision distributions plotted on the stimulus-related dimension and stimulus
distributions plotted on the decision dimension. We can observe that decision separability is greatly reduced in

the stimulus-related dimension, while stimulus separability in the decision dimension mainly depends on the same
 movement direction (e.g., blue-red represents the same movement direction in two tasks, red-green represents
 another direction), failing to well distinguish different visual stimuli of the same direction.

Overall, we updated Fig. 5b in the revised manuscript with enhanced statistical analysis to demonstrate the
 advantages of near-orthogonal encoding in providing relatively independent encoding of different
 dimensional information while minimizing cross-dimensional interference, directly addressing the
 reviewer's concerns. Additionally, we included new Supplementary Fig. 16 to provide comprehensive
 pairwise comparisons.

**3. Comment 3.1: “The key finding is that stimulus and decision space are orthogonal, but**
 **this finding is currently not well supported by statistical analyses and bears some**
 **conceptual issues: The authors should not only test the angular difference between original**
 **and shuffled data, but also whether the original data is significantly different from 90 deg;**
 **only this would somewhat support their main claim about orthogonality.”**

Reply R2-3.1: We thank the reviewer for raising this important question. In the previous revision, we had
 already noted this issue. Our key finding is that stimulus-related and decision-related subspaces exhibit near-
 orthogonal rather than perfectly orthogonal relationships. Since orthogonality implies zero correlation, which
 is quite difficult to achieve in associative learning tasks where both subspaces participate in the learning
 process, we adopted the term "near-orthogonal" to avoid misunderstanding. However, we found that some
 expressions in the abstract were still misleading regarding the orthogonal relationship. We apologize for this
 and have now revised all related statements throughout the manuscript (P1 line32-35, P8 line250, P9 line277,
 P11 line335, P11 line337).

Following the reviewer’s recommendation, we conducted additional tests to examine whether the original
 data significantly differs from 90°. We first examined individual monkey results, showing that each
 monkey's subspace angles do not significantly differ from 90°, while shuffled results show significant
 deviation from 90° (Original data: Monkey ZZ: $p = 0.19$, $g = -1.28$; Monkey AB: $p = 0.37$, $g = -0.75$; Monkey
 XW: $p = 0.36$, $g = -0.77$; Shuffled data: Monkey ZZ: $p = 0.002$, $g = 3.42$; Monkey AB: $P = 0.017$, $g = 2.51$;
 Monkey XW: $p = 0.002$, $g = 3.54$, Watson-Williams test). Cross-subject statistics across the three monkeys
 also showed similar results, with the angle between the two subspaces showing no significant difference
 from 90°, while shuffled data significantly differed from 90° (Original data: $P = 0.17$, $g = -1.43$; Shuffled
 data: $P = 0.003$, $g = 3.03$).

However, we thought this result needed to be interpreted within the biological context of the task. In
 associative learning, stimuli-related and decisions cannot be completely independent, as stimuli guide
 decision formation to some extent, making perfect orthogonality theoretically unlikely to achieve. The
 observed non-significant difference from 90° may reflect a tendency toward near-orthogonal organization.
 We hope to clarify that the primary goal of our study is not to prove absolute orthogonality, but rather to
 demonstrate how the neural system achieves effective separation of stimulus-related and decision
 information through near-orthogonal encoding. This structure provides computational advantages over
 random encoding by minimizing cross-dimensional interference while maintaining necessary information

interactions. Therefore, our analysis in the paper focuses on understanding the functional benefits of this
 encoding strategy, rather than verifying strict geometric orthogonality. We appreciate your thoughtful
 statistical suggestions and believe this perspective provides a more comprehensive interpretation of our
 findings. We have added expanded discussion of the near-orthogonal concept to Supplementary Notes 1.12.

**Comment 3.2: “The Watson-Williams test is actually not significant ($p=0.06$) in monkey ZZ**
 **according to standard practice. Interestingly, the authors use a different convention to**
 **indicate significance in figure 5A than in all their other main figures: in 5A, * is $p<0.1$,**
 **whereas elsewhere, * is $p<0.05$. The conventions should be the same for all plots and the**
 **authors should acknowledge that the results are actually not the same across the animals**
 **in this main analysis. Perhaps the abovementioned test will remedy the overall situation,**
 **but currently, I do not see clear evidence across animals that supports the main claim of**
 **the paper.”**

Reply R2-3.2: We sincerely apologize for the inconsistent annotations of statistical results in Fig. 5a, which could
 be misleading. We have now adopted uniform statistical criteria (*, $p<0.05$) throughout the manuscript.
 Additionally, we discovered a computational error in analyzing Monkey ZZ's data, where one session's data was
 omitted, leading to abnormally small standard deviations. We have corrected this error and reanalyzed all data,
 presenting the corrected results in new Fig. 5a in Reply R2-1.2. The results show that all three monkeys' results
 are significantly higher than shuffle levels (Monkey ZZ: $p = 0.040$, $g = 1.27$; Monkey AB: $p = 0.005$, $g = 1.77$;
 Monkey XW: $p = 0.002$, $g = -2.02$, Watson-Williams test). Cross-subject statistics across the three monkeys also
 showed similar results ($p < 0.001$, $g = 1.66$).

To ensure complete transparency regarding our computational process, we have provided detailed data
 spreadsheet Excel Response appendix in the response supplementary material, containing both the original
 erroneous data and the corrected results.

**Comment 3.3: “The authors should explain how an instable subspace (stimulus) can be**
 **orthogonal to another, stable subspace (decision). It rather seems that the stimulus space**
 **is just noise.”**

Reply R2-3.3: We appreciate the reviewer for raising this question. We respectfully address this from the
 following two aspects:

*(1) Geometric mechanism of near-orthogonal relationship between unstable and stable subspaces*

To illustrate how an unstable subspace can maintain near-orthogonality with a stable subspace, we propose a
 geometric analogy. The stable subspace exists within a stable manifold field in the xy-plane of 3D space, while
 the unstable subspace exists along the z-axis without a fixed manifold field, allowing arbitrary representation.
 Regardless of how the activity along the z-axis transforms, it does not affect the orthogonality with the xy-plane
 (see new Supplementary Fig. 18). This demonstrates that instability within a subspace does not preclude
 orthogonality with another subspace—the key is that unstable transformations occur within a constrained
 dimensional space that maintains orthogonality with the stable subspace (see new Supplementary Notes 1.10).

**New Supplementary Fig. 18. Example of near-orthogonality between stable and unstable subspaces.** The stable subspace (yellow plane, xy-
 plane) maintains a fixed manifold structure, while the unstable subspace (blue plane, z-axis) can undergo arbitrary transformations. Despite dynamic
 changes in the unstable dimension, orthogonality with the stable subspace is preserved, demonstrating that instability within one subspace does not
 compromise its orthogonal relationship with another subspace.

*(2) Stimulus-related subspace contains information rather than random noise*

The left bar of the classifier results in Fig. 3b demonstrates that the stimulus space contains meaningful stimulus-
 related information, achieving significantly above-chance classification accuracy. Furthermore, if this were
 merely noise (which typically distributes uniformly across all dimensions), it would be unlikely that compression
 to a few dimensions via dPCA could produce meaningful information.

**4. “It is unclear to me why the authors suggest that the behavioral deficit in reversal A is**
 **explained by a schema and not by simple reversal learning of specific sensorimotor**
 **mappings previously learned in A. An abstract schema of the task would entail that one**
 **stimulus is mapped to an upward response and another stimulus is mapped to a downward**
 **response. Such a schema can explain the transfer between tasks, but it does not seem that**
 **this very abstract schema would predict interference in reversal A, since there is still a**
 **stimulus associated with up and another stimulus associated with down.”**

**Reply R2-4:** We sincerely thank the reviewer for raising the question about why reversal tasks cannot be
 explained as simple mapping flips but require schema-based explanations. This is an important distinction
 that deserves careful consideration. First, we will examine whether the behavioral deficits observed in reversal
 A reflect schema interference or simple sensorimotor remapping. Second, we will discuss what type of schema
 the monkeys might have learned.

*(1) Neural evidence indicates that reversal learning involves new schema construction rather than simple*
 *mapping flip*

To directly address the reviewer's concern, we utilized neural evidence from our previous response (old
 Rebuttal Fig. 3) and included it in new supplementary Fig. 19 to further clarify this issue. Taking Monkey ZZ,
 who successfully learned the reversal task, as an example, we examined whether decision manifolds were
 reused across different learning stages. Using a representative recording session, we compared the decision
 manifolds formed during early reversal learning (first 20 trials) and late stage (after reaching criterion

performance) with the manifold from Task A (new Supplementary Fig. 19). We found that both the early and
 post-learning reversal manifolds differed from the Task A manifold. If the reversal task simply involved
 relearning sensorimotor mappings while utilizing the same abstract schema, we would expect the learned
 reversal manifold to resemble the original Task A manifold. However, our results show no evidence of
 manifold reuse across learning stages; rather, reversal learning involved the formation of a new decision
 manifold, distinct from those supporting continuous learning tasks.

 **New Supplementary Fig. 19. The decision representations in the reverse task. Neural manifolds in the principal component of the decision**
 **subspace during Task A and Reverse-A.** (a) Decision subspace manifolds between Task A and the first 20 trials of Reverse-A (not learned stage);
 (b) Decision subspace manifolds between Task A and Reverse-A after reaching the learning criterion, $\geq 90\%$ correct over 15 consecutive trials (learned
 stage).

(2) Potential limitations in schema reuse capabilities of non-human primates

The reviewer's question raises an important discussion about what type of schema the monkeys might have
 learned. The cognitive strategies monkeys use to learn this task may differ from humans. Humans can easily
 understand the abstract rules of associative learning tasks and transfer them to reversal tasks, but the two
 monkeys we observed may not understand it this way, as new supplementary Fig. 19 also demonstrates this
 phenomenon. We speculate that this may reflect individual or species-specific cognitive differences.
 Behavioral performance in reversal tasks varies across different animals. For example, some studies indicate
 that reversal learning can be very difficult (Bobrowicz & Greiff, 2022), whereas others show that extensive
 overtraining can facilitate reversal tasks (Dhawan et al., 2019).

Therefore, our results suggest that the deficits in reversal A are more likely explained by the need to form new
 schemas rather than reusing the same learned schemas while learning reversal sensorimotor mappings. The
 schema we observed in our study represents NCS that can generalize across similar tasks, but in reversal tasks,
 it may be affected by other factors, preventing reuse of the NCS and requiring the construction of new NCS.
 We added new Supplementary Notes 1.11 and new supplementary Fig. 19 to provide a clearer understanding
 of what type of schema the monkeys learned in our study.

Reference:

Bobrowicz, K. & Greiff, S. Executive functions in birds. *Birds* 3, 184–220 (2022).

Dhawan, S. S., Tait, D. S. & Brown, V. J. More rapid reversal learning following overtraining in the rat is
evidence that behavioural and cognitive flexibility are dissociable. *Behav. Brain Res.* 363, 45–52 (2019).

**Minor Questions:**

**5. “The authors should provide p-values for the decoding accuracies versus chance, e.g.,**
**in Figure 3c.”**

**Reply R2-5:** We thank the reviewer for this important suggestion. We have now added statistical tests
comparing decoding accuracies to chance levels in Figure 3c and throughout the manuscript where decoding
results are presented. In Fig. 3c (Monkey XW: A&B&C: Stimulus $P = 0.021$, $g = 2.274$; Decision $P = 0.003$,
$g = 6.360$; Revisit A: Stimulus $p = 0.300$, $g = 0.463$; Decision $p = 0.008$, $g = 3.690$). In Supplementary Fig.
4c (Monkey AB: A&B: Stimulus $p = 0.017$, $g = 2.470$; Decision $p = 0.009$, $g = 3.450$; Revisit-A: Stimulus
$P = 0.012$, $g = 2.992$; Decision $p = 0.011$, $g = 3.063$) and in Supplementary Fig. 4c (Monkey ZZ: A&B:
Stimulus $p = 0.001$, $g = 9.387$; Decision $p = 0.007$, $g = 3.960$; Revisit-A: Stimulus $p = 0.040$, $g = 1.600$;
Decision $p = 0.003$, $g = 5.591$).

**6. “P11, line 323 - "richer" than what?”**

**Reply R2-6:** Thank you for pointing out this ambiguity. We have revised this sentence as follows: Flesch et
al. show that task identity, stimulus–response, and decision variables are encoded in orthogonal low-
dimensional manifolds rather than within a single shared subspace, in both human prefrontal and posterior
parietal cortices [44, 45] (P11 line333-336).

**Reviewer #3**

**“The authors significantly improved the manuscript and addressed my concerns.”**

**Reply R3-0:** We sincerely thank the reviewer for the positive feedback.

**Manuscript NCOMMS-24-73204C**

**Point-by-point responses to the reviewers (Reviewers' comments in bold Arial font, our**
**responses in regular Times New Roman font):**

**Reviewer #1:**

“

**The authors have further improved the manuscript in this revision. The updated**
**visualizations are clearer and help convey the main findings more intuitively. The additional**
**statistical analyses and clarifications regarding the conceptual framework also contribute**
**to a more coherent presentation of the work. Overall, I think the manuscript has reached a**
**strong and publishable state and adds valuable insight into how neural subspaces support**
**flexible learning.”**

**Reply R1-0:** We sincerely thank the reviewer for the positive evaluation and recognition of our revised
manuscript. We also appreciate your additional constructive suggestions, which we have carefully addressed
below through further revisions.

**“Below are a few minor points for the authors' discretion:**

**1. “A very minor point for the authors' discretion: in the updated Fig. 2c & 2d, the individual**
**session markers are somewhat small and the color scheme makes some markers difficult**
**to distinguish, especially for readers with red–green color weakness. Increasing the marker**
**size and choosing a more distinct palette could improve readability.”**

**Reply R1-1:** We thank the reviewer for this suggestion. We have increased the marker size and adopted a
colorblind-friendly color palette in the revised Fig. 2c & 2d.

New Fig. 2. Experimental paradigm and behavioral analysis. (a) Schematic of the visuomotor mapping tasks. (b) Workflow for learning a series of visuomotor mapping tasks within a single session. Left: Monkeys were trained on new visuomotor mappings each session, with visual stimulus pairs not seen before. Monkeys progressed to the next task only after successfully learning the previous one. Right: Task sequences completed within a session. Monkeys AB and ZZ followed an A–B–Revisit A–Reverse A training sequence; monkey XW followed an A–B–C–Revisit A training sequence. (c) Proportion of trials required to reach performance criterion (90% correct over 15 consecutive trials) relative to the total number of trials in a session. Data pooled across three monkeys. Bar plots show the mean across 14 sessions (Monkey AB: 5 sessions; Monkey ZZ: 4 sessions; Monkey XW: 5 sessions), with error bars indicating standard deviation across sessions. Data analyzed by two-tailed paired t-tests (A vs. B/C: $P = 2.30E-04$, $g = 1.47$; A vs. Revisit-A: $P = 1.52E-06$, $g = 2.34$; A vs. Reverse-A: $P = 1.20E-03$, $g = -1.59$; B/C vs. Reverse-A: $P = 2.07E-07$, $g = -2.82$; Revisit-A vs. Reverse-A: $P = 1.02E-07$, $g = -3.37$). (d) Comparison of task learning efficiency across monkeys within-session. For each monkey, the proportion of trials needed to reach criterion is plotted, with error bars representing standard deviation across recording sessions (Monkey AB: $n = 5$ sessions, A vs. Revisit-A: $P = 0.006$, $g = 2.36$, A vs. Reverse-A: $n = 5$ sessions, $P = 0.022$, $g = -1.80$; B vs. Reverse-A: $P = 0.013$, $g = -2.00$; Revisit-A vs. Reverse-A: $P = 0.001$, $g = -3.05$; Monkey ZZ: $n = 4$ sessions, A vs. B: $P = 0.004$, $g = 3.22$, A vs. Revisit-A: $P < 0.001$, $g = 4.57$, B vs. Reverse-A: $n = 4$ sessions, $P = 0.011$, $g = -2.57$; Revisit-A vs. Reverse-A: $P = 0.005$, $g = -3.06$; Monkey XW: $n = 5$ sessions, A vs. B: $P = 0.017$, $g = 1.90$, B vs. C: $P = 0.01$, $g = -2.04$). Data are presented as mean \pm SD. Colored scatter dots represent individual sessions, with each color corresponding to one monkey. *, $P < 0.05$, **, $P < 0.01$, ***, $P < 0.001$, Data analyzed by two-tailed paired t-tests.